# Nucleosome fibre topology guides transcription factor binding to enhancers

Michael R. O'Dwyer[1,2], Meir Azagury[3,7], Katharine Furlong[1,2,4,7], Amani Alsheikh[1,2,5], Elisa Hall-Ponsele[1,2], Hugo Pinto[6], Dmitry V. Fyodorov[6], Mohammad Jaber[3], Eleni Papachristoforou[1], Hana Benchetrit[3], James Ashmore[1], Kirill Makedonski[3], Moran Rahamim[3], Marta Hanzevacki[1,2], Hazar Yassen[3], Samuel Skoda[1,2], Adi Levy[3], Steven M. Pollard[1,4], Arthur I. Skoultchi[6], Yosef Buganim[3✉] & Abdenour Soufi[1,2,4✉]

Cellular identity requires the concerted action of multiple transcription factors (TFs) bound together to enhancers of cell-type-specific genes. Despite TFs recognizing specific DNA motifs within accessible chromatin, this information is insufficient to explain how TFs select enhancers[1]. Here we compared four different TF combinations that induce different cell states, analysing TF genome occupancy, chromatin accessibility, nucleosome positioning and 3D genome organization at the nucleosome resolution. We show that motif recognition on mononucleosomes can decipher only the individual binding of TFs. When bound together, TFs act cooperatively or competitively to target nucleosome arrays with defined 3D organization, displaying motifs in particular patterns. In one combination, motif directionality funnels TF combinatorial binding along chromatin loops, before infiltrating laterally to adjacent enhancers. In other combinations, TFs assemble on motif-dense and highly interconnected loop junctions, and subsequently translocate to nearby lineage-specific sites. We propose a guided-search model in which motif grammar on nucleosome fibres acts as signpost elements, directing TF combinatorial binding to enhancers.

The assembly of TF combinations on gene *cis*-regulatory elements such as enhancers is pivotal in establishing cell-type-specific gene expression[2]. The combinatorial function of TFs has been exploited in cellular reprogramming where defined TF sets convert cells from one type to another[3]. For example, OCT4, SOX2, KLF4 and MYC (hereafter, OSKM) can reprogram somatic cells into induced pluripotent stem (iPS) cells, which resemble embryonic stem (ES) cells[4]. Likewise, GATA3, EOMES, TFAP2C and MYC (hereafter, GETM) can convert fibroblasts into induced trophoblast stem (iTS) cells, which are like trophoblast stem (TS) cells[5,6]. Adding ESRRB to GETM (hereafter, GETMR) can result in either iPS cells or iTS cells, depending on the culture conditions[7]. How a small group of TFs select enhancers to control cellular identity continues to be an important and unresolved question.

Most reprogramming cocktails contain pioneer TFs that can target silent genes within inaccessible chromatin for subsequent activation[8–11]. Pioneer TFs can individually access closed chromatin by recognizing their motifs on mononucleosomes, enabling the entry of other non-pioneer TFs[9,12–18]. Here we demonstrate that, in combination, pioneer TFs recognize multi-motif patterns displayed by nucleosome arrays with specific 3D organization, guiding their binding to cell-type-specific enhancers.

## Diverse TF binding during reprogramming

To investigate the combinatorial function of TFs, we overexpressed four distinct TF combinations in mouse embryonic fibroblasts (MEFs),

leading to four cell fates. This included overexpression of OSKM in iPS cells; GETM in iTS cells; and GETMR in both iPS cells and iTS cells, representing two embryonic stem cell states of the epiblast and trophectoderm, respectively (Fig. 1a). The fourth combination, containing BRN2, SOX9, GATA4 and MYC (hereafter, $BS_9G_4M$), which displays structural similarities to OSKM (BRN2 is a POU factor like OCT4, SOX9 is an HMG factor like SOX2 and GATA4 is a zinc-finger TF like KLF4), did not reprogram MEFs, despite their ability to convert cellular identity in other combinations[19–22].

First, we confirmed the expression of OSKM, GETM, GETMR and $BS_9G_4M$ in the vast majority of MEFs (Extended Data Fig. 1a–c). We mapped the occupancy of all TFs 48 h after ectopic induction (OSKM-48h, GETM-48h, GETMR-48h and $BS_9G_4M$-48h cells), and after reprogramming completion using chromatin immunoprecipitation followed by sequencing (ChIP–seq) (Extended Data Fig. 1a). We used TF-specific antibodies and equivalent chromatin fragmentation and sequencing depth for appropriate comparison (Extended Data Fig. 1d,e). The sites enriched for OSKM, GETM and GETMR in early reprogramming showed limited overlap with fully reprogrammed cells, consistent with initial off-target binding to the genome[9,23] (Extended Data Fig. 1f). Importantly, SOX2, MYC and ESRRB, which are endogenously expressed in both iPS cells and iTS cells, displayed cell-type-specific genome occupancy after reprogramming (Extended Data Fig. 1g). Furthermore, BRN2 sites in $BS_9G_4M$-48h showed limited overlap with neural progenitor cells where it is endogenously

[1]Institute of Regeneration and Repair, Centre for Regenerative Medicine, University of Edinburgh, Edinburgh, UK. [2]Institute of Stem Cell Research, School of Biological Sciences, University of Edinburgh, Edinburgh, UK. [3]Department of Developmental Biology and Cancer Research, Institute for Medical Research Israel-Canada, The Hebrew University-Hadassah Medical School, Jerusalem, Israel. [4]Cancer Research UK Scotland Centre, University of Edinburgh, Edinburgh, UK. [5]Health Sector, King Abdulaziz City for Science and Technology, Riyadh, Saudi Arabia. [6]Department of Cell Biology, Albert Einstein College of Medicine, New York, NY, USA. [7]These authors contributed equally: Meir Azagury, Katharine Furlong. ✉e-mail: yossib@ekmd.huji.ac.il; Abdenour.Soufi@ed.ac.uk

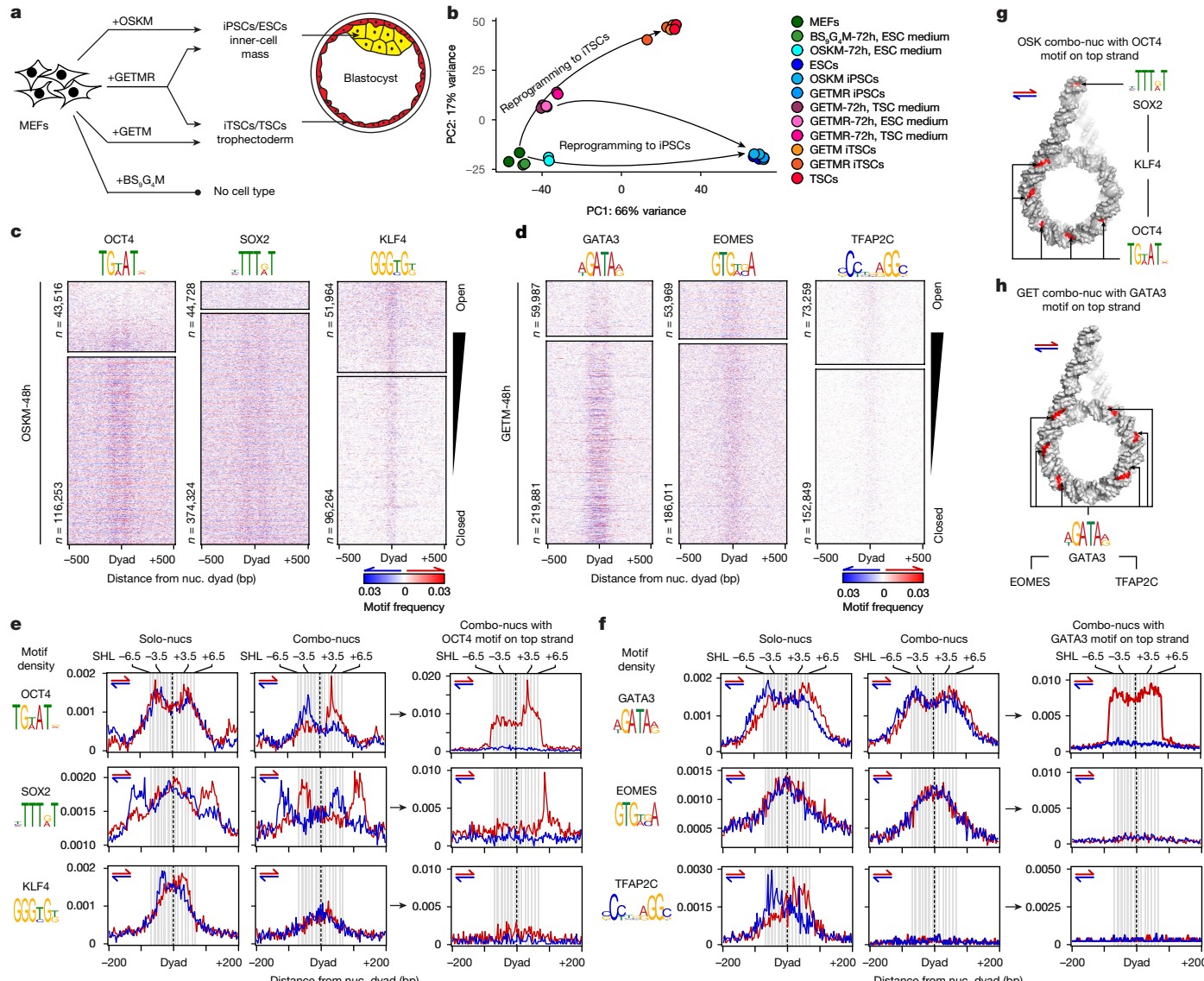

**Fig. 1 | Motif readout on mononucleosomes can explain only TF solo binding.**
**a**, Schematic of preimplantation blastocysts recapitulated by the different reprogramming cocktails used in this study. **b**, Principal component analysis of RNA-seq data in the early and final reprogramming contexts, showing a bifurcated trajectory (arrows) to iPS cells (iPSCs) and iTS cells (iTSCs) driven by GETMR. The reprogramming trajectory to iPS cells by OSKM is also indicated. **c**, Density heat maps of de novo motifs (logos on top) around nucleosome (nuc.) dyads (±500 bp) targeted by OSK during early reprogramming within open (top) and closed (bottom) chromatin. Motif density is scored on both DNA strands (red and blue) according to the colour gradient scale shown at the bottom. The number ($n$) of nucleosomes closest to each TF peak summit is indicated. **d**, The same as in **c**, but for GET during early reprogramming. **e**, Average profile plots of motif density scores on the two DNA strands (red

and blue) around nucleosome dyads (±200 bp) targeted by OSK individually (solo-nucs) or in combination (combo-nucs) during early reprogramming. Nucleosomes with dyads within ±80 bp from ChIP–seq peak summits are considered to be OSK targets. Nucleosomes targeted by all possible OSK combinations are considered to be combo-nucs. OSK combo-nucs with OCT4 motifs on the top strand ±80 bp from the dyad are shown on the right. Weighed frequency values were generated using kernel smoothing in 3 bp windows. DNA 10 bp twists are shown in grey–white stripes, indicating nucleosome SHL positions on top. **f**, The same as in **e**, but for GET during early reprogramming. **g**, Cartoon representation of OSK combo-nucs DNA (grey) containing an OCT4 motif on the top strand (red), highlighting possible SOX2 and KLF4 motif positions (red). **h**, The same as in **g**, but for GET combo-nucs with a GATA3 motif on the top strand. ESC, embryonic stem cell.

expressed, in contrast to BRN2 on-target binding observed during neuronal reprogramming when combined with ASCL1 and MYT1L[23,24] (Extended Data Fig. 1h). Thus, off-target binding to the somatic genome is a general feature of early reprogramming.

We next characterized the reprogramming process using RNA-sequencing (RNA-seq) analysis, confirming that both the iPS cells and iTS cells generated are like ES cells and TS cells, respectively (Fig. 1b and Extended Data Fig. 1i). Immunostaining of pluripotency markers in iPS cells and trophoblast markers in iTS cells, as well as the silencing of the exogenous factors, corroborate the completion of reprogramming

(Extended Data Fig. 1j–m). Moreover, BS9G4M-72h cell gene expression remained like uninfected MEFs, indicating that the control TF combinations did not change the fibroblast identity (Fig. 1b). Notably, GETMR reprogramming to iTS cells and iPS cells follows a bifurcated trajectory starting from a very similar transcriptional state to GETM but not OSKM at the 72 h timepoint (Fig. 1b and Extended Data Fig. 1n,o). Thus, reprogramming of MEFs to iPS cells using GETMR and OSKM follows divergent trajectories.

We measured chromatin accessibility using the assay for transposase-accessible chromatin followed by sequencing (ATAC–seq) in MEFs,

during early reprogramming (72 h after TF induction) and at the end of the process. Most TFs targeted predominantly closed chromatin (around 70% of sites) individually or when bound in combination, acting as pioneer TFs during early reprogramming (Extended Data Fig. 2a–e). Notably, SOX9 acted as a non-pioneer factor like MYC and was mostly associated with open chromatin, indicating that the HMG DNA-binding domain (DBD) is not always sufficient for implementing the pioneer activity (Extended Data Fig. 2d–f). After the completion of reprogramming, all pioneer TFs relocated to cell-type-specific *cis*-regulatory elements in open chromatin (Extended Data Fig. 2e–h). Moreover, the more of these factors that bind together, the more opening of closed chromatin and changes in gene expression is observed during early reprogramming (Extended Data Fig. 2i,j). Thus, while TF pioneering activity is inherent to individual TFs, chromatin opening and changes in gene expression are driven by TF combinatorial binding.

## Motif grammar on mononucleosomes

Considering that pioneer TFs engage closed chromatin by recognizing their cognate sites on nucleosomes, we hypothesized that the arrangement of multiple motifs on a single or mononucleosome would be sufficient to drive combinatorial TF binding. We mapped nucleosome positioning in MEFs, iPS and ES cells, and iTS and TS cells using micrococcal nuclease digestion with deep sequencing (MNase–seq). An exponential titration series of MNase was used to preserve 'fragile' nucleosomes[25,26] (Extended Data Fig. 3a). Intact mononucleosomes were identified as approximately 160 bp fragments in all cell types, and evidence of subnucleosomes (<150 bp) that diminished at high MNase concentrations was also observed in MEFs and iTS cells, consistent with a fragile nucleosome state (Extended Data Fig. 3b). Generally, open-chromatin sites targeted by TFs are enriched for fragile nucleosomes, while the closed sites are predominantly enriched for intact nucleosomes (Extended Data Fig. 3c–e). We measured motif enrichment around nucleosome dyads bound by each TF. Notably, motifs targeted by pioneer TFs like OCT4, BRN2, GATA3 and GATA4 are particularly enriched around nucleosomes in closed chromatin (Fig. 1c,d and Extended Data Fig. 4a–d), suggesting different motif readout on fragile and intact nucleosomes.

To define the motif grammar that may dictate whether TFs bind alone or together to nucleosomes, we identified nucleosomes bound by TFs individually (solo-nucs) and in combination (combo-nucs) based on the presence of ChIP–seq summits within ±80 bp from the dyad, considering only intact nucleosomes within closed chromatin (Extended Data Fig. 4e). Each of the OCT4, SOX2 and KLF4 factors (hereafter, OSK) display a distinct motif readout on nucleosomes (Fig. 1e and Extended Data Fig. 4f). Motif distribution on solo-nucs was markedly different from combo-nucs bound by OSK (compare the left to middle panels in Fig. 1e). In solo-nucs, OCT4 motifs positioned mainly between nucleosome superhelix location (SHL) 3.5 and 6.5 in both orientations, contrasting with the combo-nucs, which displayed an orientation-specific distribution (Fig. 1e). SOX2 motifs were enriched near the dyad of the solo-nucs in both directions, and outside the nucleosome core particle (linker DNA), in an orientation-specific manner (Fig. 1e). Conversely, SOX2 motifs were orientationally distributed between SHL 3.5 and 6.5 in combo-nucs and their linker DNA (Fig. 1e). KLF4 motifs were largely located around SHL 2.5 on solo-nucs with clear DNA strand preference, as opposed to combo-nucs, which showed relatively low motif enrichment (Fig. 1e and Extended Data Fig. 4f).

Likewise, GATA3 and TFAP2C motifs were different in solo-nucs and combo-nucs, showing orientation-specific preference mainly on solo-nucs (Fig. 1f and Extended Data Fig. 4g). However, EOMES displayed similar motif readout on solo-nucs and combo-nucs (Fig. 1f and Extended Data Fig. 4g). In BS$_9$G$_4$M-48h cells, BRN2 motifs were predominantly enriched at the extremity of solo-nucs in both DNA directions (beyond SHL 6.5), which is similar but not identical to OCT4 with homologous DBD (Extended Data Fig. 4h). GATA4 also showed different motif readout from GATA3 despite belonging to the same DBD family (Extended Data Fig. 4h). There were limited combo-nucs bound by the control TF combination, mainly enriched for GATA4 motifs (Extended Data Fig. 4h). In summary, motif grammar on nucleosomes can differentiate between solo and combinatorial TF binding, which may contribute to cell-type-specific enhancer selectivity.

As the average enrichment of different motifs on combo-nucs does not necessarily represent their co-occurrence on the same nucleosomes, we assessed the interdependence of motif co-occurrence after fixing one motif arrangement criteria. Notably, OSK combo-nucs containing at least one OCT4 motif on the top strand are depleted of OCT4 motifs on the bottom strand and any SOX2 or KLF4 motifs, apart from SOX2 motifs located in the linker DNA, beyond the ±80 bp distance threshold (Fig. 1e (right) and Extended Data Fig. 4f (right)). Searching all possible OSK combo-nucs also resulted in no particular OSK motif arrangement on the same nucleosomes (Supplementary Table 1). Importantly, the observed frequency of OSK motif co-occurrence on the combo-nucs is almost identical to their expected independent probabilities ($P$), that is, $P(\text{OSK}) = P(\text{O})P(\text{S})P(\text{K})$ (Extended Data Fig. 4i). This suggests that OSK combinatorial binding and motif co-occurrence on mononucleosomes are independent events (Fig. 1g). Similarly, GATA3, EOMES and TFAP2C (hereafter, GET) combinatorial binding could not be explained by motif co-occurrence, as GET combo-nucs that contain GATA3 motif on the top strand are not enriched for EOMES and TFAP2C motifs (Fig. 1f,h and Extended Data Fig. 4g,j). In conclusion, motif recognition on mononucleosomes can explain only TF solitary binding, indicating that TFs may be co-assembled at a higher-order nucleosome structure.

## Motif grammar on nucleosome arrays

We hypothesized that pioneer TFs engage chromatin in combination by recognizing multiple nucleosomes at the chromatin fibre level. We mapped broad domains enriched for multiple TFs and defined their nucleosome borders (Methods). This revealed extensive OSK colocalization across large genomic regions (up to ~7 kb) containing six nucleosomes on average in array arrangements and covering a total region of around 97 Mb (Fig. 2a). When bound individually, OSK engaged much smaller sites, containing one nucleosome on average (Extended Data Fig. 5a). When *Pou5f1* (encoding OCT4) is expressed alone in MEFs (O-48h), its sites significantly overlapped with OCT4 solo sites in OSKM-48h cells, but not with OSK nucleosome arrays[13] (Extended Data Fig. 5b). Gel electrophoresis mobility shift assays (EMSA) confirmed that OCT4 and SOX2 in OSKM-48h cells can form a complex on specific DNA sites, in contrast to when *Pou5f1* is expressed individually in MEFs (Extended Data Fig. 5c,d). Thus, OSK broad peaks represent their combinatorial binding to nucleosome arrays rather than disparate binding events.

Mapping the OSK motif arrangement across the nucleosome arrays revealed a notable orientation-specific distribution both within and beyond the borders of nucleosome arrays in OSKM-48h cells (Fig. 2b). Within the arrays, OCT4 and SOX2 motifs were arranged in orientation-specific clusters, while KLF4 motifs were concentrated on both directions at the centre of the arrays (Fig. 2b). Outside the arrays, all OSK motifs showed orientation-specific cluster distribution with two additional KLF4 motif peaks with opposing directions around ±5 kb away from the array centre, which we designate as the far border (Fig. 2b (dashed line)). On average, OCT4 and SOX2 showed more motif spreading compared with KLF4 motifs (Extended Data Fig. 5e). Thus, OSK co-localization on nucleosome arrays may be driven by specific motif arrangement expanding beyond the bound sites, covering around 187 Mb in total.

To decode the motif grammar within OSK nucleosome arrays, we isolated arrays containing SOX2 motifs on the same orientation at

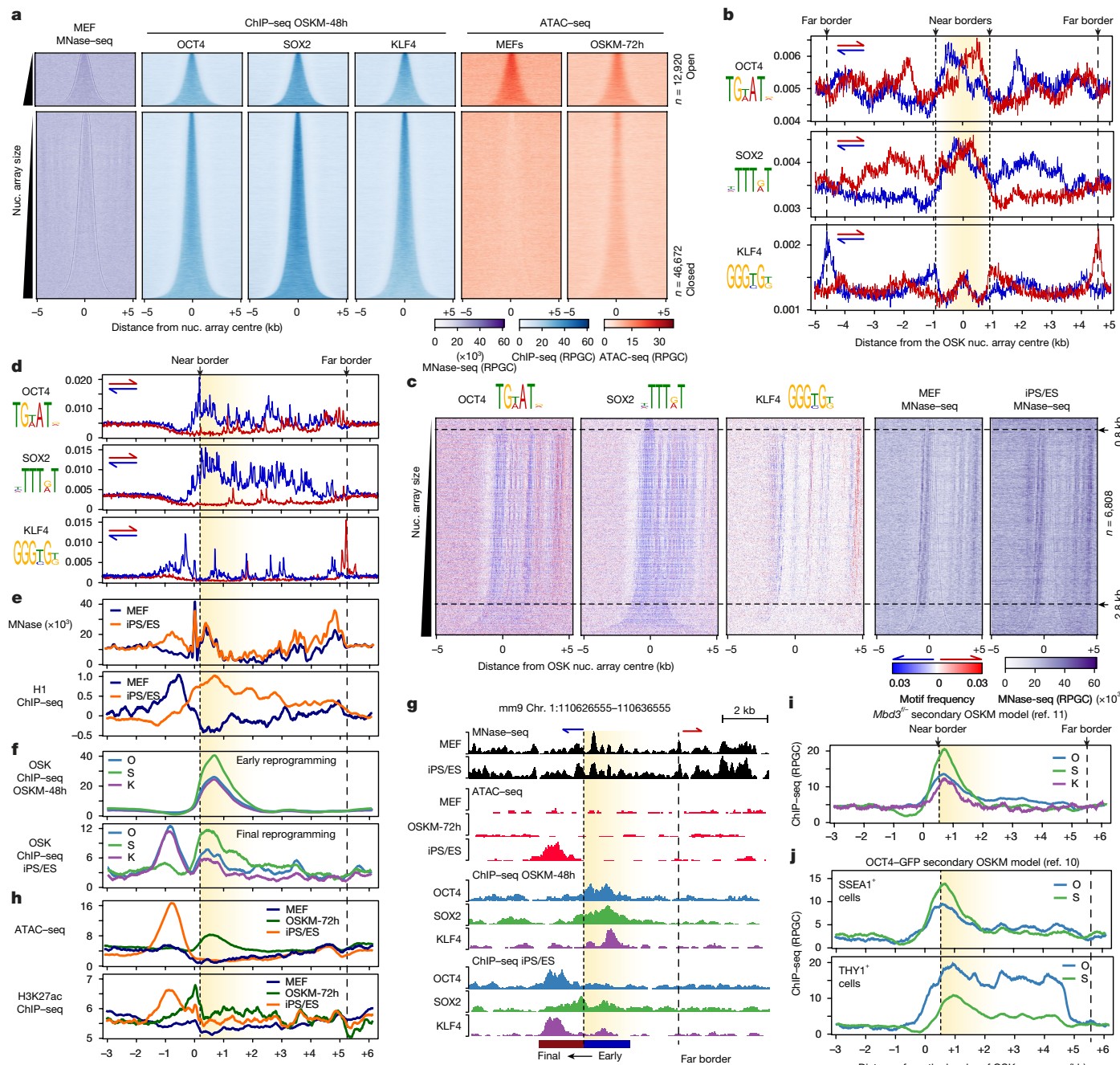

**Fig. 2 | Motif readout on nucleosome arrays deciphers OSK combinatorial binding. a**, Density heat maps showing the MNase–seq (purple), OSK ChIP–seq (blue) and ATAC–seq (red) signal, spanning ±5 kb around OSK nucleosome arrays during early reprogramming. The arrays within open (top) and closed (bottom) chromatin were separated according to ATAC–seq in MEFs and rank ordered based on size. The number of nucleosome arrays (*n*) is indicated. **b**, Profile plots of motif enrichment on both DNA strands (red and blue) around OSK nucleosome arrays (±5 kb) during early reprogramming. The average array size is highlighted in yellow. The dashed lines indicate near and far borders. **c**, Density heat maps showing OSK motif distribution (logos on top) around the OSK nucleosome array midpoints (±5 kb) during early reprogramming within closed chromatin and containing ≥7 SOX2 motifs per kb on the bottom strand. The motif density is scored on the top (red) and bottom (blue) strands, as indicated by the colour gradient scale shown at the bottom. MNase–seq read

density heat maps (purple) are also shown. The arrays were rank ordered based on size, and those within 0.8–2.8 kb are indicated by arrowheads, dashed lines and number (*n*). **d**, Profile plots of OSK motifs centred around the near border (dashed line) of OSK nucleosome arrays (0.8–2.8 kb in size) as shown in **c**. The average array size is highlighted in yellow. **e**, The same as in **d**, but showing MNase–seq (top) and H1 ChIP–seq (bottom) in MEFs (blue) and ES cells (orange). **f**, The same as **d**, but showing the OSK occupancy (ChIP–seq). **g**, Genome browser screenshot around an exemplar OSK nucleosome array targeted in early and final reprogramming, showing MNase–seq, ATAC–seq and OSK ChIP–seq. The near and far borders are indicated by dashed lines, with the direction of KLF4 motifs on top. **h**, The same as **d**, but showing ATAC–seq (top) and H3K27ac ChIP–seq (bottom). **i,j**, The same as in **d**, but showing OSK ChIP–seq data from two independent studies ((**i**)[11] and (**j**)[10]). RPGC, reads per genome coverage.

different frequencies. We focused on SOX2 motifs, as they were the most prevalent within and outside the OSK arrays. Almost all (~90%) of OSK nucleosomes arrays contained four or more SOX2 motifs per kb,

arranged in a unique direction, enabling us to split the arrays into two distinct groups, with limited overlap, based on the strandedness of SOX2 motifs (Extended Data Fig. 5f,g). Sorting OSK nucleosome arrays

by SOX2 directionality revealed high interdependence of OSK motif co-occurrence with marked parallelism in their orientation (Fig. 2c,d and Extended Data Fig. 5h). KLF4 motifs displayed a streaky appearance flanked by two stripes: one upstream of the near border and the other at the far border in the opposite direction (Fig. 2d). This OSK motif arrangement was most evident in nucleosome arrays ranging from 0.8 to 2.8 kb in size, and thereby containing 4–12 nucleosomes (Fig. 2c). Moreover, pronounced nucleosome phasing was observed at the near and far borders of OSK arrays, indicating nucleosome stacking against fixed barriers (Fig. 2c,e). In conclusion, specific motif grammar at the scale of chromatin fibre may direct OSK to accumulate near one border.

## Nucleosome fibres as signpost elements

In fully reprogrammed cells, OSK were colocalized on nucleosome arrays (enhancers) containing no OCT4 or SOX2 motif enrichment over the background but were enriched for the KLF4 motif with no apparent directionality (Extended Data Fig. 6a). The nucleosome arrays within pluripotency enhancers were also smaller in size than during early reprogramming, spanning ~5 Mb only (Extended Data Fig. 6b). We therefore examined the positional relationship between OSK binding in early (off targets) and final reprogramming (pluripotency enhancers). Transitioning from early to fully reprogrammed cells was concurrent with a lateral shift of OSK binding across the near border of the initial OSK nucleosome arrays to the enhancers (Fig. 2f,g). Overall, OSK nucleosome arrays were in the vicinity (~500 bp) of pluripotency enhancers compared with random genomic regions (Extended Data Fig. 6c). The shift in OSK binding to enhancers was also associated with an increase in chromatin accessibility and histone H3 Lys27 acetylation (H3K27ac) (Fig. 2h (orange lines)). However, during early reprogramming (OSKM-72h), H3K27ac was deposited mainly at the near border of OSK arrays before spreading to the enhancers, in contrast to chromatin accessibility, which followed OSK binding (Fig. 2h (green lines)). Nucleosome enrichment also spread across the near border of the OSK arrays in fully reprogrammed cells (Fig. 2e,g (MNase)). Notably, the linker histone H1 enrichment and OSK binding were mutually exclusive, moving in opposite direction during reprogramming (Fig. 2e). Along with H1, OSK arrays became enriched for the repressive histone marks H3K9me1/2/3 as well as HP1 and SUV39H1/2 displaying distinctive patterns and depleted from histone marks and co-factors usually associated with open chromatin (Extended Data Fig. 6d,e). Thus, OSK initially target nucleosome arrays adjacent to pluripotency enhancers.

Considering that the lateral movement of OSK binding mirrors the directionality of OSK motifs, we hypothesized that this motif distribution funnels OSK binding along nucleosome fibres to adjacent enhancers (hereafter, signpost elements). Using the secondary OSKM-MEF-*Mbd3*[f/−] systems[11], we confirmed that OSK were also colocalized on nucleosome arrays (Fig. 2i). In another secondary system, in which a subpopulation of MEFs poised to become iPS cells (SSEA1[+]) was isolated from cells that resisted reprogramming (THY1[+])[10], OCT4 and SOX2 were enriched at OSK nucleosome arrays near the border in SSEA1[+] cells but spread across the entire nucleosome fibre (from the far to near borders) in THY1[+] cells (Fig. 2j). Thus, during successful reprogramming, OSK binding is effectively guided to accumulate next to enhancers.

To functionally validate the directionality of OSK motifs in signpost elements, we selected the pluripotency gene *Nanog*, where OSK were initially bound next to the enhancer (Fig. 3a). We constructed a piggyBac plasmid with dual fluorescence reporter cassettes (Methods). The first contains the intact *Nanog* promoter–signpost–enhancer element (~5 kb) driving eGFP expression, while the second enables tdTomato expression under the intact *Nanog* promoter and enhancer but separated by a flipped signpost element, thereby reversing the directionality of OSK motifs (Fig. 3b). We inserted an insulator between the two reporter cassettes to eliminate transcriptional interference,

and flanked two insulators at both ends to minimize integration position effects from the neighbouring chromosomal environment[27] (Fig. 3b). PiggyBac-targeted ES cells expressed both eGFP and tdTomato at a similar efficiency (Fig. 3c,d). We injected the sorted dual eGFP/tdTomato[+] ES cells into host blastocysts and then isolated chimeric mouse embryos at E13.5 (Fig. 3c). Both eGFP and tdTomato reporters were equally silenced in all tissues apart from in the gonad, reflecting the precise expression of *Nanog* at this embryonic stage[28] (Fig. 3e).

We next investigated whether the direction of the signpost element has any effect on reactivating the silenced eGFP/tdTomato reporters in MEFs from chimeric embryonic day 13.5 (E13.5) embryos during reprogramming by OSKM (Fig. 3c). In accordance with our hypothesis, eGFP[+] cells gradually increased starting from day 9 after OSKM induction, whereas tdTomato[+] cells did not appear until after the completion of reprogramming and stability of iPS cells (Fig. 3f and Extended Data Fig. 6f). Moreover, fully reprogrammed tdTomato[+] cells were always eGFP[+], suggesting that eGFP was already activated in these cells (Fig. 3g). The reactivation of both eGFP and tdTomato continued to increase in individual iPS cell clones with extended passaging (Extended Data Fig. 6g–i). Motif orientation within signpost elements is therefore crucial for reactivating pluripotency enhancers during reprogramming.

## OSK signpost elements are within loops

To characterize the chromatin organization of the signpost elements, we used Micro-C to map the 3D chromatin architecture at single-nucleosome resolution[29–31] (~100–200 bp). Two different MNase concentrations were used, which resulted in efficient proximity-induced nucleosome ligation (Extended Data Fig. 7a,b). Micro-C consistently recovered fine-scale internucleosome contacts in specific orientations within arrays of up to six nucleosomes[31,32] (Extended Data Fig. 7c). Measuring the average Micro-C junction density showed a markedly diminished internucleosome interaction within OSK arrays in cells during early reprogramming compared with in fully reprogrammed cells (Fig. 4a). Micro-C pileup analysis at fine-scale resolution (bin, 100 bp) revealed two intense interaction points across the borders of OSK arrays during early reprogramming, which deconvoluted into a single anchor point after arranging the arrays by SOX2 motif directionality (Fig. 4b and Extended Data Fig. 7d). This is consistent with a loop conformation linking the near and far borders of OSK arrays, and thereby aligning the two opposing KLF4 motifs towards one direction (Fig. 4c,d). Importantly, the Micro-C junction intensity at these loop anchors was significantly reduced in final reprogramming (arrowheads in Fig. 4b,c), suggesting that chromatin opening is associated with the dismantling of loop intersections. At long-range resolution (bin, 20 kb), OSK nucleosome arrays were entrenched in a highly interactive environment during early reprogramming consistent with closed chromatin but becoming more connected in a loosely connected environment in iPS cells, indicative of open chromatin (Extended Data Fig. 7e,f). Motif directionality may therefore guide OSK binding along chromatin loops to accumulate near the borders, before infiltrating laterally to H1-enriched nucleosomes inside loop junctions where enhancers are located (Fig. 4d). The loop fusion from the outside-in may be initiated by depositing H3K27ac at the near-border and evicting H1, which could be mediated by other factors that bind preferentially to H3K27ac nucleosomes[33].

To examine the role of H1 on chromatin fibre conformation and OSK binding, we identified H1.3/1.4 as the major H1 variants expressed in MEFs using high-performance liquid chromatography (HPLC) analysis of acid-extracted chromatin (Extended Data Fig. 7g,h). We were able to reduce chromatin-associated H1 protein levels by knocking down H1.4 with shRNA (H1.4 KD) and elevate H1 levels by overexpressing H1.4 (H1.4 OE), as validated using liquid chromatography-mass spectrometry (LC–MS) and western blotting, despite the compensatory effects of the remaining H1 variants[34,35] (Extended Data Fig. 7g–j).

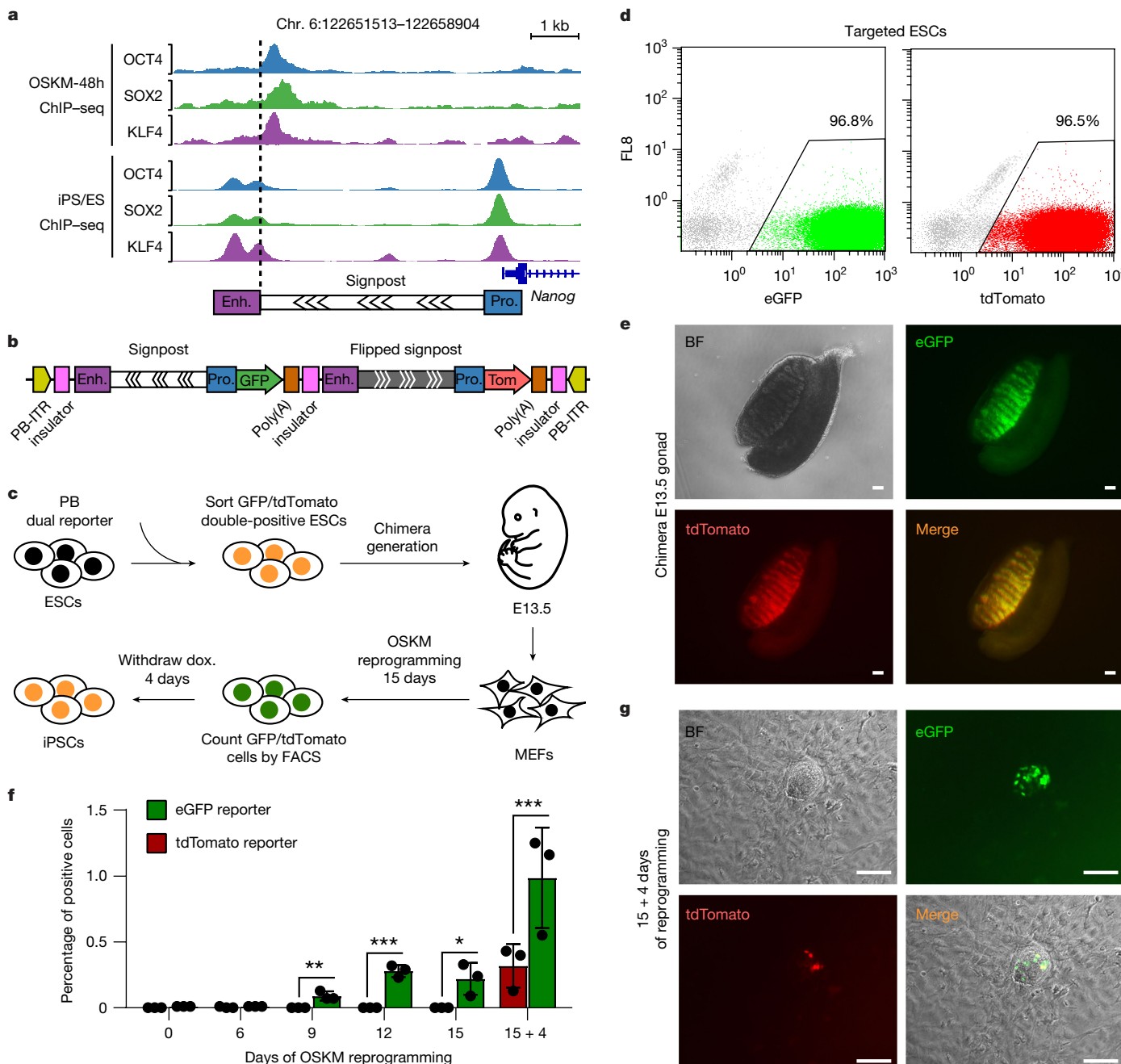

**Fig. 3 | Signpost elements guide OSK binding to pluripotency enhancers during reprogramming. a**, Genome browser screenshot of the *Nanog* locus bound by OSK (ChIP–seq) in early (top) and final reprogramming (bottom). A schematic of the *Nanog* promoter (pro.) and enhancer (enh.) separated by a signpost element, with the directionality of OSK motifs indicated by chevrons, is shown below. **b**, Schematic of the PiggyBac (PB) construct containing the dual eGFP/tdTomato reporter cassettes. eGFP is driven by the WT *Nanog* promoter–signpost–enhancer shown in **a**, and tdTomato is driven by the same promoter–enhancer but separated by a flipped signpost element. ITR, inverted terminal repeat. **c**, Experimental flow chart illustrating PB construct integration into ES cells, which contributed to chimeric embryos from which MEFs were derived; these cells were then used for iPS reprogramming to examine the reactivation of the dual reporters. **d**, The expression of eGFP and tdTomato in ES cells targeted by the PB construct was measured using flow cytometry and the percentages of eGFP⁺ and tdTomato⁺ cells are indicated. FL8, fluorescence channel 8 (non-specific channel). **e**, Expression of eGFP and tdTomato in the male gonad isolated from chimeric embryos at E13.5, reflecting *Nanog* expression. Representative image from *n* = 3 biological replicates. Scale bar, 100 μm. **f**, Motif directionality in the signpost element leads to more efficient eGFP activation during reprogramming. Quantification of eGFP⁺ and tdTomato⁺ cells during reprogramming is shown, as measured using flow cytometry. Statistical significance was determined using two-sided paired *t*-tests; *$P$ = 0.03, **$P$ = 0.01, ***$P$ < 0.001. Data are mean ± s.d. from three biological replicates (*n* = 3). **g**, eGFP expression precedes tdTomato in reprogramming. Fluorescence images of an iPS cell colony showing expression of eGFP and tdTomato at day 15 followed by 4 days without doxycycline (dox.). Bright-field (BF) and merged images are also shown. Representative image from *n* = 3 biological replicates. Scale bar, 100 μm.

We then performed Micro-C analysis of MEFs after H1.4 KD and H1.4 OE and probed for distinctive nucleosome array conformations by measuring the abundance of internucleosomal contacts[30]. In MEFs,

contacts between nucleosomes *n* and *n* + 2 (*n*–*n* + 2) are almost identical to *n*–*n* + 4, and *n*–*n* + 3 is like *n*–*n* + 5, supporting the folding of chromatin fibre into a two-start zig-zag helix with tetranucleosomal

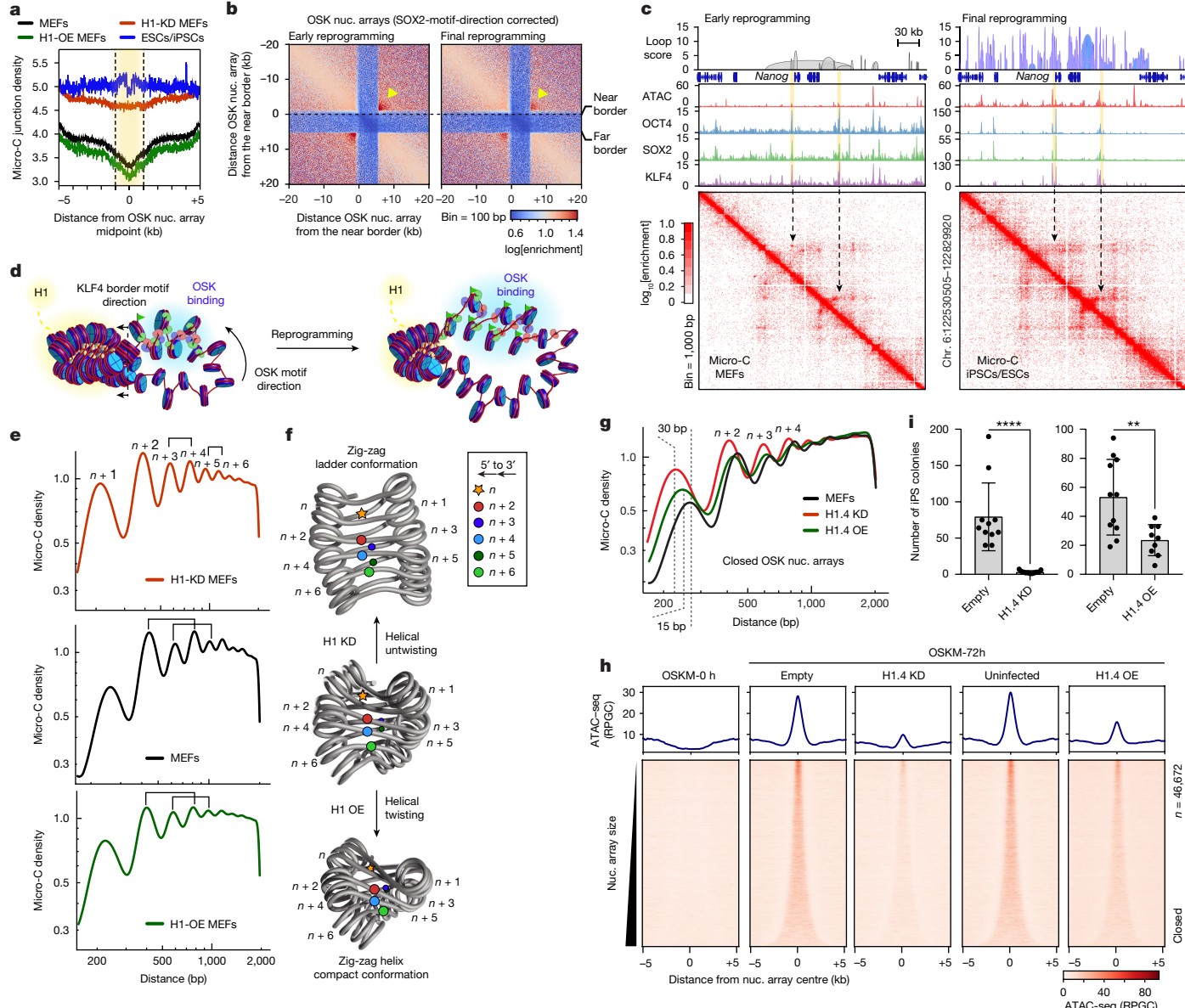

**Fig. 4 | OSK target chromatin loops with diminished linker histone. a,** Profile plots of Micro-C ligation junctions around OSK nucleosome arrays (±5 kb) in early (black) and final (blue) reprogramming, H1-KD MEFs (red) and H1-OE MEFs (green). **b,** Micro-C pileup heat maps of OSK nucleosome arrays (SOX2-motif-direction corrected) in early (right) and final (left) reprogramming. Maps are plotted using bin = 100 bp at log scale. The arrowheads indicate strong interactions across chromatin loops. **c,** Micro-C contact matrices (bottom), showing 1-kb-resolution interactions around the *Nanog* locus in early (left) and final (right) reprogramming. Contacts around exemplar OSK nucleosome arrays are indicated by arrows with the corresponding genome browser tracks of ATAC–seq and ChIP–seq shown above and highlighted in yellow. Associated loops called by FitHiChIP (*q* < 0.01) are shown at the top. **d,** Schematic of the nucleosome array organization within chromatin loops, illustrating OSK co-binding in early (left) and final (right) reprogramming. OSK binding is highlighted in blue, H1-enrichment in yellow and H3K27 acetylation is indicated

by green flags. **e,** Micro-C decay curves showing internucleosomal contacts in H1-KD MEFs (red), MEFs (black) and H1-OE MEFs (green). Interactions between nucleosome *n* and *n* + *x* in 5′-to-3′ orientation and similar abundance are linked by brackets. **f,** Cartoon representations of the two-start zig-zag nucleosome fibre that comply with the internucleosomal *n* and *n* + *x* contacts shown in **e.** The coloured circles indicate the ligated partners between *n* (star) to *n* + *x* (coloured circles) in 5′-to-3′ orientation. **g,** Micro-C decay curves as in **e** for OSK nucleosome arrays. Nucleosome repeat-length changes are indicated by dashed lines. **h,** Profile plots (top) and heat maps (bottom) of ATAC–seq around OSK nucleosome arrays of TNG-KOSM-MEFs (OSKM-0h) or after OSKM induction (OSKM-72h) infected with empty, H1.4-KD or H1.4-OE vectors. **i,** Quantification of iPS cells (NANOG⁺) generated from TNG-KOSM-MEFs that were infected with empty, H1.4-KD and H1.4-OE vectors. Statistical significance was determined using two-sided unpaired *t*-tests; ****$P$ < 0.0001, **$P$ = 0.005. Data are mean ± s.d. from biological replicates (*n* = 11).

repeating units, as seen previously in mouse ES cells[30] (Fig. 4e,f (middle)). Micro-C analysis of H1.4-KD MEFs showed a distinctive pattern whereby the *n*–*n* + 3 ligation frequency was similar to *n*–*n* + 4, and *n*–*n* + 5 is similar to *n*–*n* + 6, which is consistent with an untwisted zig-zag ladder conformation (Fig. 4e,f (top)). The chromatin fibre in H1.4-OE MEFs folds into a more-twisted (condensed) zig-zag helix, where nucleosomes *n* + 2, *n* + 3, *n* + 4 and *n* + 5 become closer to

each other, therefore resulting in similar ligation frequencies with nucleosome *n* (Fig. 4e,f (bottom)). Such twisted and untwisted zig-zag helices are consistent with the structures of condensed and relaxed nucleosome arrays bound to linker histone H1 under different ionic conditions[36,37]. Thus, changing H1 levels in mammalian cells can substantially change nucleosome organization by twisting and untwisting the chromatin fibre.

Within OSK nucleosome arrays, Micro-C junction density in H1.4-KD MEFs become almost flat like the flanking regions (Fig. 4a (red line)). In H1.4-OE MEFs, the Micro-C junction density remained depleted within OSK arrays but slightly less than that in MEFs (green line in Fig. 4a). Internucleosomal contacts within OSK arrays in MEFs support loose zig-zag folding, where H1.4 KD decreased the nucleosome repeat length by around 30 bp and H1.4 OE by around 15 bp (Fig. 4g). This indicates that nucleosomes are stacked closer together after changing H1 levels, which is known to greatly affect chromatin fibre folding[35,38,39]. We next used ATAC–seq to investigate whether H1 levels affect chromatin accessibility within OSK arrays in MEFs. The closed OSK arrays remained inaccessible in both H1.4-KD MEFs and H1.4-OE MEFs, suggesting that H1 levels can change chromatin fibre conformation without affecting chromatin accessibility (Extended Data Fig. 7k). We therefore investigated whether H1 levels affect OSK binding during reprogramming, using the secondary system TNG-MKOS-MEFs[40]. Using chromatin accessibility as a proxy for OSK binding, we performed ATAC–seq in TNG-MKOS-MEFs after H1.4 KD or H1.4 OE and inducing OSKM for 72 h (Extended Data Fig. 7l). As seen with primary MEFs, reprogramming TNG-MKOS-MEFs resulted in the opening of OSK arrays, which remained almost inaccessible after H1.4-KD and only marginal accessible in H1.4-OE, indicative of diminished OSK binding (Fig. 4h). Importantly, both H1.4 KD and H1.4 OE significantly inhibited reprogramming to iPS cells (Fig. 4i). In conclusion, H1 levels affect OSK binding to nucleosome arrays by changing chromatin conformation, not the overall accessibility, supporting the role of chromatin fibre topology in TF combinatorial binding.

## GET bind highly connected signpost elements

We examined GET combinatorial binding in GETM-48h cells; GET also targeted larger genomic regions compared to when bound individually, although GET co-assembled on 3–5 nucleosomes on average, relatively smaller than OSK (Fig. 5a and Extended Data Fig. 8a). GET nucleosome arrays were enriched for GET motifs without any directionality (Extended Data Fig. 8b,c). However, TFAP2C motifs, which are palindromic sequences, were positioned either upstream (left) or downstream (right) of the border nucleosome, polarizing GET arrays to two distinct groups (Fig. 5b,c and Extended Data Fig. 8d,e). After the completion of reprogramming, GET remained partly bound within the initial nucleosome arrays and partly spread to other arrays (enhancers) containing less GATA3 and EOMES motifs but more TFAP2C motifs (Fig. 5d and Extended Data Fig. 8b). While chromatin accessibility mirrored GET binding during reprogramming, H3K27ac was predeposited at the border nucleosomes (with TFAP2C motifs) before reprogramming, which then spread to GET arrays during reprogramming (Fig. 5e). Thus, GET bind to chromatin fibres following specific motif grammar before finding their enhancers.

In contrast to OSK, GET nucleosome arrays were enriched for internucleosome contacts as well as H1 (Fig. 5f,g and Extended Data Fig. 8f). At the fine scale, nucleosome contacts within GET arrays were almost diminished after reprogramming (compare the yellow arrows in Fig. 5h). However, the long-range interactions mediated by GET were significantly enhanced in final reprogramming (Extended Data Fig. 8g). Moreover, GET nucleosome arrays form stripes at the boundaries of topologically associated domains that became diffused in final reprogramming (Fig. 5i (black arrows) and Extended Data Fig. 8h). Such stripe patterns suggest that GET nucleosome arrays spatially segregate into nucleosome array assemblies, facilitating the translocation of GET across chromatin to find their enhancers. Indeed, GET arrays are significantly more linked by loops to TS cell enhancers than random sequences (Fig. 5j and Extended Data Fig. 8i). Furthermore, GET arrays are depleted from the cohesin subunit RAD21 and CTCF, suggesting that GET may translocate to enhancers by chromatin guided translocation rather than the loop excursion model, unless these factors are involved later during the process[41,42] (Extended Data Fig. 8j). Notably, H1.4 KD

and, to a lesser extent, H1.4 OE increased Micro-C junction density within GET arrays, re-enforcing the important role of H1 on chromatin organization (Fig. 5g). Furthermore, H1.4 KD and H1.4 OE both blocked iTS cell reprogramming (Fig. 5k). In summary, GET recognize highly interconnected signpost elements located at H1-enriched loop junctions, guiding their translocation to enhancers by fusing the loops from the inside out (Fig. 5l).

## MYC follows different motif grammar

Although MYC does not act as a pioneer factor, it can access closed chromatin by co-binding with OSK and more extensively with GET but there is negligible co-binding with the control factors (Extended Data Fig. 9a). Importantly, the combinatorial binding with MYC resulted in substantially more chromatin opening in early reprogramming (Extended Data Fig. 9b), consistent with its ability to recruit histone acetyltransferases[43]. Mapping the MYC motif (E box) enrichment across OSKM nucleosome arrays revealed a marked central depletion in the arrays, but an orientation-specific enrichment at the borders, which continued outside the arrays, reminiscent of OSK motif distribution (Extended Data Fig. 9c,d). However, MYC binding with GET was completely E-box independent (Extended Data Fig. 9e). Notably, AlphaFold-Multimer predicted that MYC and its obligate heterodimer MAX can directly interact with TFAP2C homodimer[44] (Extended Data Fig. 9f). Indeed, EMSA and immunoprecipitation confirmed that MYC can directly interact with TFAP2C, suggesting that MYC binding with GET is driven by MYC–TFAP2C protein–protein interactions (Extended Data Fig. 9g,h). Thus, MYC combinatorial binding with GET and OSK follows distinct motif grammar on nucleosome arrays.

## Competitive TF binding on nucleosome fibres

To examine how ESRRB can expand GETM reprogramming, we compared GETM ChIP–seq in the presence and absence of ESRRB. Notably, the enrichment of TFAP2C and its partner MYC were markedly lower in GETMR-48h compared with in GETM-48h cells (Extended Data Fig. 10a,b). TFAP2C remained mainly bound to the sites that are co-occupied by GEM as well as ESRRB (Extended Data Fig. 10b–d). Moreover, ESRRB nucleosomes that are co-bound by TFAP2C were enriched only for TFAP2C motifs, in contrast to the other ESRRB nucleosomes, which were enriched only for ESRRB motifs, suggesting that their co-binding may occur at the nucleosome array level (Extended Data Fig. 10e). We have therefore identified nucleosome arrays that contained TFAP2C-retained or TFAP2C-lost sites in GETMR-48h cells. Micro-C pileup analysis revealed that TFAP2C binding was retained in arrays with more internucleosome contacts, mediating ESRRB–TFAP2C combinatorial binding (compare the yellow arrowheads in Extended Data Fig. 10f). Thus, the addition of ESRRB restricts GETM combinatorial binding by retaining TFAP2C in nucleosome fibres with discrete topology.

As the relative stoichiometry between EOMES and ESRRB has been shown to influence GETMR reprogramming[7], we hypothesized that ESRRB binding with TFAP2C occurs in competition with EOMES and GATA3. Indeed, increasing ESRRB levels reduced the amounts of the co-immunoprecipitated TFAP2C with EOMES, and the reverse is true (Extended Data Fig. 10g). Accordingly, removing EOMES and GATA3 from GTEMR (TMR) was sufficient to reprogram MEFs to iPS cells that are morphologically and functionally like mouse ES cells (Extended Data Fig. 10h–k). Thus, ESRRB expands GETM reprogramming capacity by competing with EOMES and GATA3 to bind with TFAP2C and MYC.

## Discussion

During cellular reprogramming, the prevailing view is that TFs sample the genome randomly aided by low-affinity sites to select

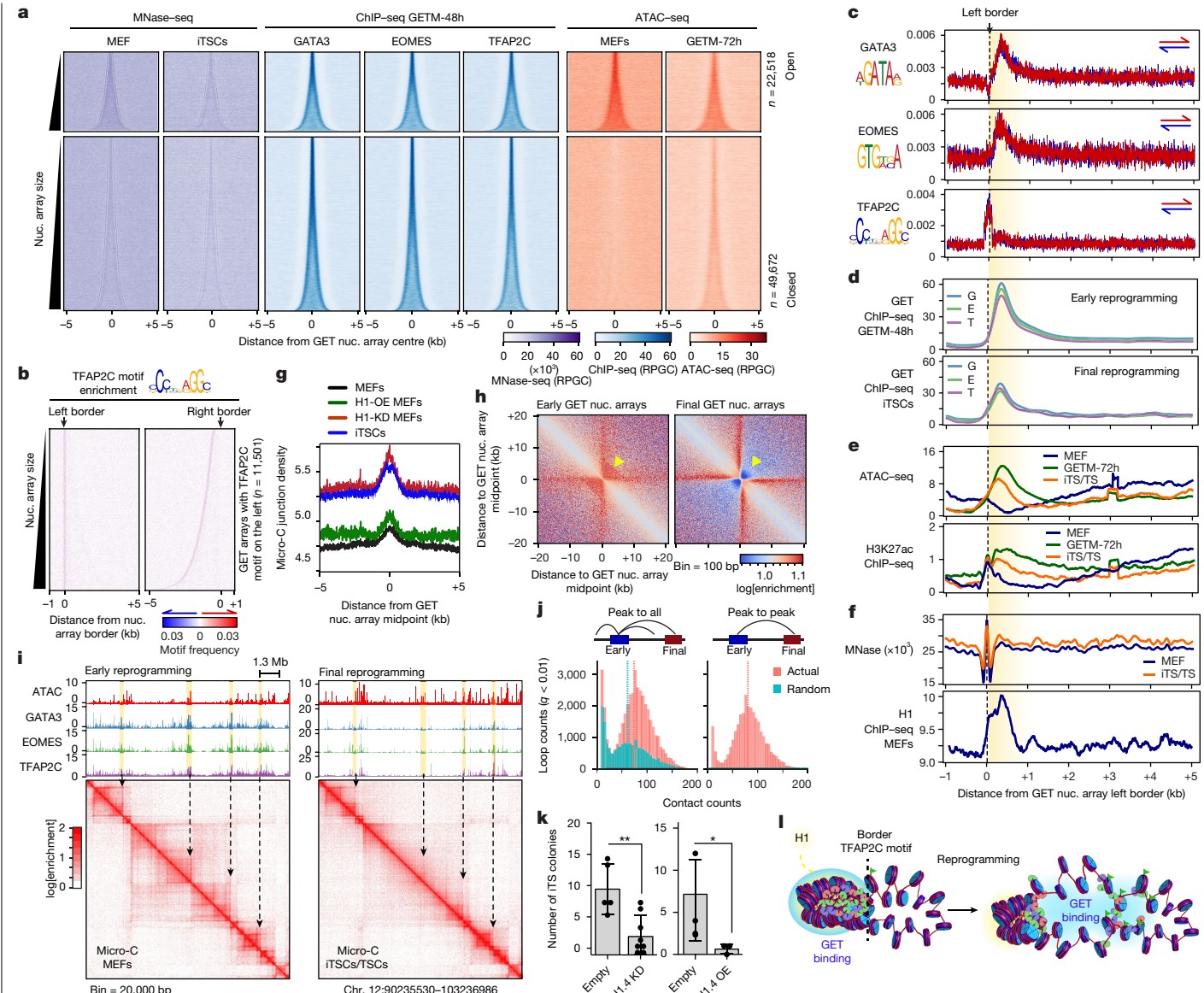

**Fig. 5 | GET target loop junctions enriched for linker histone. a**, Density heat maps showing the MNase–seq (purple), GET ChIP–seq (blue) and ATAC–seq (red) signal around GET nucleosome arrays during early reprogramming. The arrays were grouped by ATAC–seq in MEFs and ranked by size. The number of nucleosome arrays (*n*) is indicated. **b**, Motif density heat maps on DNA strands (red and blue) around the left and right borders of GET nucleosome arrays containing the TFAP2C motif on the left border. The arrays were rank-ordered by size and motifs were scored by colour gradient scale (bottom). **c**, Profile plots of GET motifs centred around the GET array left border (dashed line). The average array size is highlighted in yellow. **d**–**f**, The same as in **c**, but for GET ChIP–seq (**d**), ATAC–seq and H3K27ac ChIP–seq (**e**), and MNase–seq and H1 ChIP–seq (**f**). **g**, Profile plots of Micro-C density around GET nucleosome arrays in early (black) and final (blue) reprogramming, H1-KD MEFs (red) and H1-OE MEFs (green). **h**, Micro-C pileup heat maps of GET nucleosome arrays during

early reprogramming (left) and in fully reprogrammed cells (right). The arrowheads indicate interactions within GET nucleosome arrays diminished after reprogramming. **i**, Micro-C contact matrices highlighting stripe contacts (arrows) at topologically associated domain borders where GET binding is strongest, as indicated by the genome tracks above. **j**, Chromatin loops linking the actual GET nucleosome arrays to all regions (left) or enhancers (right) in iTS cells compared with randomized sequences, as shown in the inset. **k**, The number of iTS cell colonies (CDX2⁺) of H1.4-KD MEFs and H1.4-OE MEFs compared with MEFs infected with an empty vector. Statistical significance was determined using two-sided unpaired *t*-tests; *P = 0.02 and **P = 0.001. Data are mean ± s.d. from *n* = 6 (H1.4-KD) and *n* = 3 (H1.4-OE) biological replicates. **l**, Schematic of chromatin loop junctions targeted by GET in early (left) and final (right) reprogramming. GET binding is shown in blue, H1 enrichment is shown in yellow and H3K27 acetylation is indicated by green flags.

cell-type-specific enhancers[45,46]. However, the dynamics by which lineage-specific TFs explore chromatin to reach enhancers is not consistent with random sampling[47]. For example, OCT4 and SOX2 find their target sites in fewer than 100 binding attempts, suggesting that they sample only a miniscule fraction of the genome (<1%)[48]. Here we propose that, instead of being random, TF combinations recognize motifs exhibited by nucleosome assemblies with defined 3D organization (signpost elements), which guide their binding to enhancers

in accordance with a 'guided search' model (Extended Data Fig. 10l). Motif readout on higher-order chromatin structures would therefore reduce the dimensionality of the genome to be explored by TFs for an optimal search process. An unexpected aspect of this model is that motifs are used as guides for TFs, not their final destiny, suggesting that enhancer functionality has a key role in trapping TF combinatorial binding. However, signpost elements can also act as rheostats to fine-tune or synchronize enhancer activity. A challenge in the future is

to resolve the difference in the kinetics of TF binding, which is usually measured in timescales of seconds to minutes, and enhancer activity and cell fate changes that require longer timescales (days to weeks).

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

# Methods

## Inclusion and ethics

All animal experiments for the iPS cell and iTS cell generation from MEFs were approved by the University of Edinburgh Animal Welfare and Ethical Review Body, performed at the University of Edinburgh, and carried out according to regulations specified by the Home Office and Project License. All reprogramming experiments have been approved by the University of Edinburgh SBS ethics committee (asoufi-0001). This research was performed in compliance with the joint ethics committee (IACUC) of the Hebrew University and Hadassah Medical Center and the National ethic committee (Israel health ministry) and NIH, which approved the study protocol for animal welfare. The Hebrew University is an AAALAC international accredited institute.

## MEF isolation

Primary MEFs were generated from 129 and 129/C57BL/6 mouse embryos at E12.5–13.5 after removing internal organs and heads. The remaining body of each embryo was incubated in 200 µl of trypsin-EDTA (0.25%, Gibco) for 15 min at 37 °C. The trypsin was then inactivated by adding 800 µl MEF medium and the embryos were quickly dissociated with an 18 gauge needle fixed to 1 ml syringe. The embryo suspension was passed through the syringe several times (4 to 6) until becoming homogeneously cloudy and then transferred, drop-wise, to a 15 ml falcon tube containing 9 ml of warm MEF medium (GMEM (Sigma-Aldrich, G5154), 10% FCS, 1 mM sodium pyruvate, 1 mM L-glutamine, 1× non-essential amino acids (Thermo Fisher Scientific, 11140035)). The suspension was sedimented by gravity until forming a cell debris pellet. The majority of the supernatant (10 ml), containing single cells, was gently removed and plated onto a 10 cm dish containing warm MEF medium. The cells were monitored daily under a microscope and, if not confluent after 2 days, the cells were discarded. The confluent cells (passage 0) were collected by trypsin digestion and cryopreserved or used immediately.

## Chimeric embryo

Blastocyst injections were performed using CB6F1 host embryos. After priming with PMSG (M.I.P. Veterinary) and hCG (Merck) hormones and mating with CB6F1 males, embryos were obtained at 3.5 days post-coitum (blastocyst stage), and then injected with 10–20 PB-integrated ES cells, tdTomato-marked ES cells or TMR-iPS cells with a flat tip microinjection pipette with an internal diameter of 16 mm (Origio) in a drop of FHM medium (Zenith Biotech, ZEHP-050) covered by mineral oil. Shortly after injection, blastocysts were transferred to 2.5 days post-coitum pseudopregnant CD1/ICR females (10–15 blastocysts per female). Chimeric embryos and placentas were isolated at E13.5 and observed under the fluorescence microscope (Nikon Eclipse T!). Gonads were excised from chimeric embryos at E13.5 and observed by fluorescence microscope (Nikon Eclipse T!).

## Cell culture

MEFs were maintained in MEF medium (GMEM (Sigma-Aldrich, G5154), 10% FCS, 1 mM sodium pyruvate, 1 mM L-glutamine, 1× non-essential amino acids (Thermo Fisher Scientific, 11140035), 0.1 mM β-mercaptoethanol, penicillin (50 U ml$^{-1}$) and streptomycin (50 µg ml$^{-1}$)) at 37 °C and 5% $CO_2$. Human embryonic kidney 293T (HEK293T) cells (Lenti-X TAKARA, 632180) were maintained in HEK medium (GMEM, 10% FCS, 1 mM sodium pyruvate, 1 mM L-glutamine) at 37 °C and 5% $CO_2$. Mouse ES cells were grown on 0.2% gelatine and maintained in ES cell medium (GMEM, 10% FCS, 1 mM L-glutamine, 0.1 mM β-mercaptoethanol, 1× non-essential amino acids, 100 U ml$^{-1}$ leukaemia inhibitory factor (LIF)) at 37 °C and 5% $CO_2$. TS cells were maintained on γ-irradiated feeder MEFs on 0.2% gelatin in TS cell medium (RPMI-1640 (Thermo Fisher Scientific, 21875034), 20% FCS, 0.1 mM β-mercaptoethanol, 1× non-essential amino acids, penicillin–streptomycin (100 µg ml$^{-1}$), 25 ng ml$^{-1}$ hFGF4 (R&D, 235-F4-025), 1 µg ml$^{-1}$ heparin (Sigma-Aldrich, H3149)). TS cells cultured without feeders were maintained on Matrigel-coated plates (Corning) in TX medium (DMEM/F12 without HEPES and L-glutamine (Life Technologies), 64 mg l$^{-1}$ L-ascorbic acid-2-phosphate magnesium, 14 µg l$^{-1}$ sodium selenite, 19.4 mg l$^{-1}$ insulin, 543 mg l$^{-1}$ NaHCO$_3$, 10.7 mg l$^{-1}$ holo-transferrin (all Sigma-Aldrich), 2 mM L-glutamine, 1% penicillin and streptomycin), freshly supplemented with 25 ng ml$^{-1}$ hFGF4, 2 ng ml$^{-1}$ hTGF-ß1 (PeproTech) and 1 µg ml$^{-1}$ heparin (Sigma-Aldrich)[49]. All ChIP–seq, ATAC–seq, RNA-seq, Micro-C and MNase–seq experiments were performed under feeder-free conditions.

## Reprogramming to iPS cells and iTS cells

Reprogramming MEFs to iPS cells and iTS cells was performed as previously described[5,7]. All infections were performed on MEFs (passage 0 or 1) that were seeded at 60–80% confluency 2 days before the first infection. For infection, replication-incompetent lentivirus expressing vectors encoding for reprogramming TFs and ratios (GETM, 3:3:3:1; TMR, 4.5:1:4.5; GETMR, 2:2.5:2:1:2.5; OSKM, 3:3:3:1; and BS$_9$G$_4$M, 3:3:3:1) were packaged with a lentiviral packaging mix (5.1 µg psPAX2 and 2.4 µg pMD2.G) in 10 cm dishes containing HEK293T cells and collected at 48 h after transfection. The supernatants were filtered through a 0.45 µm filter, supplemented with 8 µg ml$^{-1}$ of polybrene (Sigma-Aldrich), and then used to infect MEFs. Then, 24 h after the infection, the medium was replaced with fresh GMEM (Thermo Fisher Scientific) containing 10% FBS. To initiate reprogramming, 2 µg ml$^{-1}$ doxycycline was added to the culture medium (GMEM containing 10% FBS) for the first 48 h before switching into the relevant reprogramming medium. For iPS cell reprogramming, the medium was replaced to ES cell medium supplemented with LIF at a final concentration of 200 U ml$^{-1}$ and 2 µg ml$^{-1}$ doxycycline for a further 12 days before withdrawing doxycycline. For iTS cell reprogramming, medium was replaced to TS cell reprogramming medium and 2 µg ml$^{-1}$ doxycycline. For reprogramming to iTS cells or iPS cells with GETMR, reprogramming medium was replaced every other day for 20 days with doxycycline, followed by 10 days culture without doxycycline. The plates were monitored for primary iPS cell and iTS cell colonies. For iPS cell clone isolation, single-iPS-cell colonies were trypsinized (0.25%), and individually plated in separate wells in a six-well plate on feeder cells. The morphology of the isolated colonies was monitored under the microscope and medium was replaced every other day for five to ten passages, until stable iPS cell colonies were developed.

## Early reprogramming

Owing to the large chromatin amounts required to carry out ChIP–seq in early reprogramming, large-scale concentrated lentiviruses encoding for each TF were generated. First, HEK293T cells were seeded at a density of $2 × 10^6$ cells per 15 cm plate and grown in 30 ml HEK medium for 24 h, before being transfected with the relevant lentivirus plasmids. Each virus was prepared in a separate dish. For transfection, 2.4 µg pMD.G, 5.1 µg psPAX2 and 7.5 µg of the corresponding FUW-tet-O-TF vector were dissolved in 1,710 µl Opti-MEM medium (Thermo Fisher Scientific, 31985062) and 90 µl Fugene 6 reagent (Promega, E2692), thoroughly mixed by vortexing and incubated for 15 min at room temperature, before adding to the 15 cm plate containing HEK293T cells, which were incubated for 16 h. The transfection medium was replaced with fresh HEK medium and the transfected cells were cultured for a further 60 h. The lentiviruses were collected by collecting the 30 ml supernatant, which was passed through a 0.45 µm polyethersulfone filter-fitted syringe and incubated for 16 h at 4 °C with 10 ml Lenti-X reagent (Clontech, 631232). The virus was then pelleted by centrifugation at 1,500g for 1 h at 4 °C. The supernatants were removed, and the viral pellet was dissolved in 200 µl GMEM overnight at 4 °C and then aliquoted and stored at −80 °C. On average, the titre of each virus was identified as around $7 × 10^8$ infection units per ml.

For early-reprogramming ChIP–seq analysis, $4.8 \times 10^6$ MEFs (passage 1) were cultured in MEF medium on a 15 cm dish for 16 h. The next morning, the cells were infected by replacing the medium to MEF medium containing the Tet-ON OSKM, GETM, GETMR or $BS_9G_4M$ lentiviruses at a multiplicity of infection (MOI) of 5 for each TF plus 5 MOI of rtTA2M2 lentivirus and 8 µg ml$^{-1}$ polybrene. After 24 h, the medium was changed to MEF medium without polybrene. The next day, the infected cells reached around 90% confluency and were split 1 in 2 and incubated for a further 16 h, before TF induction by adding 2 µg ml$^{-1}$ doxycycline to the medium and incubating for 48 h. The cells were then cross-linked to collect the chromatin (see the 'ChIP-seq' section).

## ChIP-seq

Chromatin fragments were prepared from approximately $1.5 \times 10^7$ cells per TF. For cell cross-linking, 3 ml of formaldehyde cross-linking buffer (50 mM HEPES-KOH, pH 7.5, 100 mM NaCl, 1 mM EDTA, 0.5 mM EGTA, 11% formaldehyde) was added to 15 cm dishes (Corning, 430599) containing 30 ml medium and incubated at room temperature for 10 min with swirling every 2 min. Cross-linking was blocked by adding 1.65 ml 2.5 M glycine and incubating for 5 min with swirling at room temperature. Cells were collected in their medium using a silicon scraper (Thermo Fisher Scientific, 08100240) and centrifuged for 5 min at 1,350 rcf at 4 °C. The cross-linked pellet was washed three times with 10 ml ice cold PBS by resuspension and subsequent centrifugation for 5 min at 1,350 rcf at 4 °C. Five 15 cm dishes of ES cells or iTS cells and seven 15 cm dishes of infected MEFs were combined into single pellets for processing. The pellets were subsequently flash-frozen with liquid nitrogen and stored at −80 °C.

For efficient lysis, MEF samples were flash-frozen in liquid nitrogen and thawed in ice three times before thawing on ice for 1 h. Cell pellets were resuspended in 10 ml lysis buffer 1 (50 mM HEPES-KOH pH 7.5, 140 mM NaCl, 1 mM EDTA, 10% glycerol, 0.5% NP-40 substitute (Sigma-Aldrich, 74385), 0.25% Triton X-100 and cOmplete Ultra Protease Inhibitor (Roche, 5892970001)) with rotation for 10 min at 4 °C. Nuclei were extracted by passing the cell lysates through a tight 7 ml Dounce homogeniser with 40 strokes on ice. Nuclei were collected by centrifugation at 1,350 rcf for 5 min at 4 °C. The nuclei were washed in 10 ml lysis buffer 2 (10 mM Tris-HCl pH 8, 200 mM NaCl, 1 mM EDTA, 0.5 mM EGTA and cOmplete Ultra Protease Inhibitor) for 10 min with rotation at room temperature. The nuclei were then collected by centrifugation at 1,350 rcf for 5 min at 4 °C and resuspended in 5 ml lysis buffer 3 (10 mM Tris-HCl pH 8, 100 mM NaCl, 1 mM EDTA, 0.5 mM EGTA 0.1% Na-deoxycholate, 0.5% N-lauroylsarcosine and cOmplete Ultra Protease Inhibitor).

The resuspended nuclei were split into five aliquots in prechilled 1 ml millitubes containing AFA Fibre (Covaris, 520130) and sonicated using the Covaris M220 focused ultrasonicator (Covaris) (peak power, 75 W; duty factor, 10; cycles per burst, 200; minimum temperature, 5 °C; set temperature, 7 °C; maximum temperature, 9 °C). The millitubes were each sonicated for 10 min intervals sequentially and kept on ice. Sonicated chromatin was transferred to Protein Lobind tubes (Eppendorf). Then, 100 µl of 10% Triton X-100 was added to each 1 ml sonicated chromatin to increase chromatin solubility. Chromatin samples were then centrifuged (20,000g at 4 °C for 10 min) and the supernatants transferred into fresh tubes. The optimum sonication time was determined by taking 50 µl aliquots in 10 min intervals and checking DNA fragment size distribution by agarose gel electrophoresis until predominantly generating a 150–350 bp band. Early reprogramming samples were sonicated for 60–70 min, ES cells samples for 30–40 min and iTS cells for 50–60 min. Another 50 µl aliquot from the final sonication was retained to be used as an input DNA control for ChIP analysis. The sonicated chromatin and the input DNA samples were snap-frozen in liquid nitrogen and stored at −80 °C.

For each ChIP replicate, 30 µl of Protein G Dynabeads (Thermo Fisher Scientific, 10004D) was washed three times in blocking solution (PBS, 0.5% (w/v) BSA). The beads were saturated with 10 µg antibody raised against the appropriate TF (Supplementary Table 2) diluted in 200 µl blocking solution by rotating for 6 h at 4 °C. The beads were then washed three more times in blocking solution. ChIP was performed by incubating the beads with 40 µg of chromatin (based on DNA content) on a rotator for 20 h at 4 °C. The beads were then transferred to a fresh prechilled tube, washed five times with RIPA wash buffer (50 mM HEPES-KOH pH 7.5, 500 mM LiCl, 1 mM EDTA, 1% NP-40 substitute, 0.7% Na-deoxycholate) and once with TE NaCl (10 mM Tris-HCl pH 8, 1 mM EDTA, 50 mM NaCl). Bound chromatin was eluted by resuspending the beads in 200 µl ChIP elution buffer (50 mM Tris-HCl pH 8, 10 mM EDTA, 1% SDS) and shaking at 65 °C for 30 min before transferring the supernatant to a fresh tube. Cross-linking was reversed by incubating for 16 h at 65 °C with shaking. The samples were diluted with 200 µl TE (10 mM Tris-HCl pH 8, 1 mM EDTA) and then incubated with 0.2 mg ml$^{-1}$ RNase A (Sigma-Aldrich, R4642) for 2 h at 37 °C. Proteins were then digested by incubating with 0.2 mg ml$^{-1}$ proteinase K (Ambion, AM2546) for 2 h at 55 °C. The DNA was then purified by phenol–chloroform extraction followed by ethanol precipitation. Precipitated DNA was eluted in 20 µl of 10 mM Tris-HCl pH 8.5 for library generation or qPCR analysis. ChIP reactions were quantified by Qubit 2.0 using the HS dsDNA quantification kit (Thermo Fisher Scientific, Q32854).

ChIP–seq DNA libraries were prepared using the NEBNext Ultra II Library Preparation Kit (NEB, E7645S) with dual-index primers (NEB, E7600S). For each TF, libraries were prepared using 5–20 ng ChIP DNA corresponding to a pool of at least three ChIP replicates. Input libraries were generated using 20 ng of sonicated DNA. Size selection (200 bp) was performed according to the manufacturer's instructions. PCR amplification during library preparation was limited such that samples with 5–10 ng of ChIP DNA underwent 11 cycles of PCR amplification and samples with 10–20 ng of ChIP DNA underwent 10 cycles. Input libraries were generated using 20 ng of DNA starting material. PCR clean-up was performed with 45 µl Seramag Speedbeads in 10% PEG-8000 solution. Libraries were quantified using a Qubit 2.0 device with a high-sensitivity dsDNA kit (Thermo Fisher Scientific, Q32854) and fragment size was determined using an Agilent 2200 Tapestation with D1000 HS reagents (Agilent, 5067-5584, 5067-5585). The samples were sequenced by Edinburgh Genomics on either an Illumina HiSeq 4000 using 75 bp paired-end settings or on an Illumina NovaSeq using 50 bp paired end settings.

## RNA-seq

Total RNA was isolated using the Qiagen RNeasy kit. All mRNA libraries were prepared using the SENSE mRNA-Seq library prep kit V2 (Lexogen), and pooled libraries were sequenced on the Illumina NextSeq 500 platform to generate 75 bp single-end reads.

## ATAC–seq

ATAC–seq library preparation was performed as previously described[5,7,50]. In brief, 100,000 cells per replicate (two biological replicates per line) were incubated with 0.1% NP-40 to isolate nuclei. Nuclei were then transposed for 30 min at 37 °C with adaptor-loaded Nextera Tn5 (Illumina, Fc-121-1030). Transposed fragments were directly PCR amplified and sequenced on the Illumina NextSeq 500 platform to generate $2 \times 36$ bp paired-end reads.

For H1 OE and H1 KD, 400,000 cells per sample were incubated with 0.1% NP-40, 0.1% Tween-20 and 0.01% digitonin (Calbiochem, 300410) to isolate nuclei. Nuclei were then split into four replicates of 100,000 cells each for transposition for 30 min at 37 °C using the Illumina Tagment DNA Enzyme and Buffer small kit (20034210). Transposed fragments were directly PCR amplified and sequenced on the NovaSeq 600 system to generate 50 bp paired-end reads.

## MNase–seq

MNase samples were prepared from approximately $1.5 \times 10^7$ cells per digestion condition. For cross-linking, 1.1 ml of cross-linking buffer

(Dulbecco's PBS with 11% formaldehyde) was added to 10 ml medium and incubated at room temperature for 10 min with swirling on a 10 cm cell culture plate (Corning, 430167). Cross-linking was blocked by adding 0.55 ml 2.5 M glycine and incubating for 5 min with swirling at room temperature. The medium was aspirated from the cross-linked cells and the cells were washed twice with 10 ml ice cold SST (150 mM NaCl, 0.5 M trisodium citrate, 10 mM Tris-HCl pH 7.5). Cells were scraped into 5 ml ice cold RSB (10 mM Tris-HCl pH 7.5, 10 mM NaCl, 3 mM MgCl$_2$ and 10 mM sodium butyrate with cOmplete Ultra EDTA-Free Protease Inhibitor (Roche, 5892953001) supplemented with 0.5% NP40 substitute (Roche, 11332473001). Cells were then pelleted at 1,000 rpm for 3 min at 4 °C in a Rotina 380R centrifuge (Hettich) with a swinging-bucket rotor (Hettich, 1754). The supernatants were discarded, and cells were resuspended in 1 ml RSB with NP40 substitute and incubated for 1 min on ice. The cells were disrupted by passing through a tight 2 ml Dounce homogeniser with 20 strokes on ice. 4 ml RSB with NP40 substitute was added to the sample and the sample was centrifuged for 7 min at 4 °C at 1,400 rpm. The supernatants were discarded, and the nuclei were resuspended in 10 ml cold RSB with NP40 substitute. Nuclei were pelleted by centrifugation at 900 rpm for 10 min. The supernatants were discarded, and nuclei were resuspended in 600 µl cold RSB. A 2 µl aliquot was taken and mixed with 98 µl 1 N NaOH, and the optical density at 260 nm (OD$_{260}$) was measured using the Eppendorf BioPhotometer Kinetic system. The dilution-corrected OD$_{260}$ value of the nuclei was adjusted to 1 using RSB.

For a reaction of MNase, 5 ml of OD$_{260}$ = 1 nuclei was transferred to a 15 ml tube. One tube was processed at a time. Then, 150 µl 100 mM CaCl$_2$ was added to a final concentration of 3 mM, and the sample was incubated for 90 s in a 37 °C water bath. Micrococcal nuclease (Worthington Biochemicals, LS004797) was added to a final concentration of 0, 1, 4, 16 or 64 U ml$^{-1}$ and chromatin was digested for 2 min in a 37 °C water bath. To inactivate the MNase, 5.2 ml 2× room temperature TNESK (20 mM Tris-HCl pH 7.5, 200 mM NaCl, 2 mM EDTA, 2% SDS and 0.2 mg ml$^{-1}$ proteinase K) was then added and the sample was mixed vigorously. The sample was placed at 37 °C for at least 2 h and then placed at 65 °C overnight to reverse cross-linking. The samples were purified by phenol–chloroform extraction followed by ethanol precipitation. RNase A was then added to a concentration of 0.2 mg ml$^{-1}$ and the sample was incubated for 2 h at 37 °C. The DNA sample was subsequently purified by phenol–chloroform extraction and ethanol precipitation. Then, 7.5 µg of this sample was run on a 1.3% agarose gel to check the digestion pattern.

To purify digested samples, digested DNA was run on a 6% polyacrylamide TBE gel for 3 h and 30 min at 90 V on a 20 × 20 cm vertical electrophoresis system. The gel was post-stained with ethidium bromide and a band was excised corresponding to around 90 to 210 bp. This excised gel was broken up by centrifuging through a 0.5 ml tube that had been pierced with a needle into a 1.5 ml tube. Two gel volumes of diffusion buffer (500 mM ammonium acetate, 10 mM magnesium acetate, 1 mM EDTA, 0.1% SDS) were added and the sample was shaken overnight at 37 °C. The sample was then rotated for 2 h on a wheel at room temperature. The sample was then centrifuged for 10 min at 20,000g and the supernatant was transferred to new tube. This was centrifuged for a further 10 min at 20,000g to remove gel fragments and the supernatant was transferred to a new tube. The DNA was then purified by ethanol precipitation followed by further purification using the Monarch PCR and DNA cleanup kit (NEB, T1030). DNA was quantified using the Qubit 2.0 Flourometer (Thermo Fisher Scientific) using the HS dsDNA quantification kit (Thermo Fisher Scientific, Q32854).

MNase–seq libraries were prepared using 30 ng, 300 ng, 500 ng or 1 µg DNA for 1 U ml$^{-1}$, 4 U ml$^{-1}$, 16 U ml$^{-1}$ or 64 U ml$^{-1}$, respectively, using the NEBNext Ultra II Library Preparation Kit (NEB, E7645S) with dual-index primers (NEB, E7600S). The manufacturer's instructions were followed for library preparation, apart from deviations in bead-based size-selection and PCR clean-up. A modified size-selection protocol was carried out before PCR cycling, the volumes of size-selection beads

for 200 bp libraries were changed to 42 µl and 37.5 µl for the first and second size-selection bead additions. The size-selection beads used here were Seramag SpeedBeads carboxyl-coated particles (GE healthcare, GE65152105050250), prepared with a 1 in 50 dilution in a solution of 18% (w/v) PEG-8000 solution (10 mM Tris-HCl pH 8, 1 mM EDTA, 1 M NaCl, 0.05% Tween-20). This deviation from the manufacturer's protocol was to avoid small-fragment loss. 1 U ml$^{-1}$ MNase samples underwent ten cycles of PCR amplification, and all of the other samples underwent seven cycles of PCR amplification. PCR clean-up was done with 45 µl of Seramag Speedbeads prepared in 18% PEG-8000. Library fragment size was determined using an Agilent 2200 Tapestation with D1000 HS reagents (Agilent, 5067-5584, 5067-5585). MNase–seq libraries were sequenced by Edinburgh Genomics on the Illumina NovaSeq platform using an SII flow cell with 50 bp paired-end settings to a depth of approximately 160 million reads pairs per library.

### Micro-C

To prepare cross-linking samples for Micro-C, cells were grown to a confluency of approximately 80% on 15 cm cell culture dishes. Before starting, a 15 cm or 10 cm plate, which was prepared in parallel, was trypsinized and used to obtain approximate cell numbers by counting on a haemocytometer. Micro-C samples were then allowed to come to room temperature before the culture medium was aspirated and the samples were washed twice with 30 ml DPBS. For cross-linking, 3.3 ml of cross-linking buffer (DPBS with 11% formaldehyde) was added to the plate, containing 30 ml DPBS and incubated at room temperature for 10 min with swirling. Cross-linking was blocked by adding 1.65 ml 2.5 M glycine and incubating for 5 min with swirling at room temperature. The samples were then transferred onto ice and incubated for 15 min. Cells were then scraped on ice and transferred to 50 ml conical tubes. Cells were then pelleted at 1,000 rpm for 5 min at 4 °C in a Rotina 380R centrifuge (Hettich) with a swinging-bucket rotor (Hettich, 1754). Subsequently, pellets of the same type were combined in a single 15 ml conical tube and resuspended in 10 ml cold DPBS before being pelleted at 1,000 rpm for 5 min at 4 °C. Cell pellets were then resuspended at 4 million cells per ml in DPBS with 3 mM DSG and rotated for 40 min at room temperature (DSG stock was initially prepared by making up a 300 mM stock in DMSO and diluting into DPBS). DSG was quenched by adding glycine to a final concentration of 400 mM and incubating at room temperature for 5 min before transferring to ice for 15 min. Cells were then washed twice with DPBS 0.5% BSA and snap-frozen in pellets containing 5 million cells using liquid nitrogen before being stored at −80 °C. One cell pellet was used per Micro-C library.

To prepare MNase digestions, two cell pellets were resuspended in 600 µl PBS with 0.1 mg ml$^{-1}$ BSA (NEB, B9000S) and incubated on ice for 20 min. Cells were then collected by centrifugation by spinning at 5,000g for 5 min at 4 °C. The cell pellet was then washed with MB1 (10 mM Tris-HCl, pH 7.5, 50 mM NaCl, 5 mM MgCl$_2$, 1 mM CaCl$_2$, 0.2% NP-40, 1× Roche cOmplete EDTA-free (Roche Diagnostics, 04693132001)), collected by centrifugation (5,000g for 5 min at 4 °C) and then resuspended into 225 µl MB1 per 1 million cells (1,125 µl per sample). The pellet was then split into five 200 µl digestion aliquots (with 100 µl taken as a no-digestion control). To one set of five 200 µl aliquots, 15 U of MNase was added and, to the other five 200 µl aliquots, 20 U of MNase was added by adding 15 or 20 µl of 1 U µl$^{-1}$ MNase, respectively, and the samples were incubated for 10 min in 37 °C water bath. A 20 min digestion was used for the H1-OE MEF samples. To stop the digestion, 2 µl of 0.5 M EGTA was then added and the samples were incubated at 65 °C for 10 min to inactivate the MNase.

MNase digestion samples of the same MNase concentration were then recombined into a single tube, and 100 µl was taken as a no-ligation control. The remaining recombined samples were then split across two tubes and the cells were collected by centrifugation (5,000g for 5 min at 4 °C). Cells were then washed with 500 µl 1× NEB buffer 2.1 (NEB, B7202), pelleted by centrifuging at 5,000g for 5 min at 4 °C and were

then resuspended in 45 μl 1× NEB buffer 2.1. Then, 5 μl rShrimp alkaline phosphatase (NEB, M0203) was added and the samples were incubated at 37 °C for 45 min to dephosphorylate DNA ends. The reaction was then stopped by incubating at 65 °C for 5 min. Next, 40 μl Klenow pre-mix buffer (5 μl 10× NEB buffer 2.1, 2 μl 100 mM ATP (Thermo Fisher Scientific, R0441), 3 μl 100 mM DTT, 30 μl water), 8 μl large Klenow fragment (NEB, M0210L) and 2 μl T4 PNK (NEB, M0201L) were added, in that order. 5′ DNA overhangs were then generated by incubating at 37 °C for 15 min. 5′ overhangs were then filled in with biotinylated nucleotides by adding 100 μl biotin pre-mix (10 μl 1 mM biotin-14-dATP (Jena Biosciences, NU-835-BIO14-L), 10 μl 1 mM biotin-14-dCTP (Jena Biosciences, NU-956-BIO14-L), 1 μl of 10 mM dGTP and 10 mM dTTP (NEB, N0446), 10 μl 10× T4 ligase buffer (NEB, B0202S), 0.5 μl 200× BSA (NEB, B9000S), 67.5 μl water) and incubating for 45 min at 25 °C. Then, 12 μl of 0.5 M EDTA was added and the sample was incubated for 20 min at 65 °C to stop the reaction.

The samples were pelleted at 10,000g for 5 min at 4 °C, the supernatant was removed and the pellet was resuspended in 500 μl 1× T4 ligase buffer with 50 mM NaCl. The samples were centrifuged at 10,000g for 5 min at 4 °C and the was supernatant removed. The samples were then resuspended in 500 μl ligation pre-mix (5 μl 200,000 U ml$^{-1}$ T4 ligase (NEB, M0202M), 1.25 μl 200× BSA, 50 μl 10× T4 ligase buffer, 443.75 μl water) and incubated for 2.5 h at room temperature. Next, 5 μl of 5 M NaCl was then added, the sample was centrifuged at 16,000g and 4 °C and the supernatant was discarded.

To remove biotin nucleotides from unligated DNA ends, pellets were resuspended in exonuclease mix (20 μl 10× NEB buffer 1 (NEB, B7001S), 170 μl water and 10 μl 100,000 U ml$^{-1}$ exonuclease III (NEB, M0206L)) and incubated at 37 °C for 15 min with agitation. Subsequently, 1.25 μl 20 mg ml$^{-1}$ RNase A, 10 μl 20 mg ml$^{-1}$ proteinase K and 25 μl 10% SDS were added. At this point, no-ligation control samples were also processed by diluting to 200 μl with 100 μl water and RNase A, proteinase K and 10% SDS were added as above. The samples were incubated at 65 °C overnight to lyse cells. The DNA was then purified by phenol–chloroform extraction followed by ethanol precipitation and eluted in 100 μl 10 mM Tris-HCl pH 8.5. A further round of DNA purification was carried out using the Zymo Research DNA Clean & Concentrator-5 kit (Zymo Research, D4013) and eluted in 6.5 μl of 10 mM Tris-HCl pH 8.5. The ligation efficiency was tested by comparing the no-ligation control and unligated samples on the Agilent 2200 Tapestation using HS D1000 reagents. At this point, individual replicates of ligation samples were pooled (that is, 2 replicates of 2.5 million cells generated by splitting the MNase digest across two tubes).

To purify dinucleosome-sized ligated fragments, a 1.5% gel prepared with either NuSieve GTG low-melting-point agarose (Lonza, 50081) or TopVision low-melting-point agarose (Thermo Fisher Scientific, R0801). TAE running buffer was prechilled to 4 °C and ligation samples were run at 60 V for 2.5 h on ice. A band was excised corresponding to around 250–400 bp. DNA was purified from this using the Zymoclean Gel DNA Recovery Kit (Zymo Research, D4001T) using 31 μl 10 mM Tris-HCl pH 8.5 as the elution buffer. The DNA concentration was determined using the Qubit 2.0 system and high-sensitivity dsDNA reagents.

To prepare Micro-C libraries, 2.5–10 μl Dynabeads MyOne Streptavidin C1 beads (Invitrogen, 65001) were prepared depending on the amount of DNA present in the Micro-C sample relative to the binding capacity of beads as specified by the manufacturer. These beads were washed once with 300 μl 1× TBW (5 mM Tris-HCl, pH 7.5, 0.5 mM EDTA, 1 M NaCl, 0.05% Tween-20) and suspended in 150 μl 2× BB (10 mM Tris-HCl, pH 7.5, 1 mM EDTA, 2 M NaCl). Micro-C samples were diluted to 150 μl final volume by adding 120 μl nuclease-free water and then added to the bead suspension. The samples were incubated for 20 min at room temperature with agitation. The beads were washed twice with 300 μl 1× TBW by incubating at 55 °C for 5 min with agitation. The beads were then suspended in 35 μl 10 mM Tris-HCl pH 8.5, 3.5 μl end prep reaction buffer and 1.5 μl end prep enzyme mix (from the NEBNext Ultra II DNA

library prep kit) was added. The samples were then incubated for 30 min at 20 °C with agitation and then for 30 min at 65 °C with agitation. Then, 0.5 μl of NEBNext Adapter for Illumina, 15 μl NEBNext ligation master mix and 0.5 μl NEBNext ligation enhancer were added and the samples were incubated for 30 min at 20 °C with agitation. Next, 1.5 μl NEBNext USER enzyme was then added, and the samples were incubated for 30 min at 37 °C with agitation. The beads were then washed once with 100 μl 1× TBW by incubating at 55 °C for 5 min with agitation. The beads were then washed once with 100 μl 10 mM Tris pH 7.5 and then suspended in 20 μl 10 mM Tris pH 7.5. Then, 2 μl of bead suspension was then taken as a test quantitative PCR (qPCR) reaction to find a suitable number of PCR cycles for library generation. The beads were then split into nine PCR tubes (to reduce the number of beads settling in individual PCR tubes during PCR cycling), and 10 μl NEBNext Ultra II Q5 Master Mix, 2 μl 10 μM NEBNext i5 primer, 2 μl 10 μM NEBNext i7 primer (NEB, E7600S) and 4 μl water were added. PCR was then performed according to the NEBNext Ultra II Library kit cycling conditions with 9 or 10 PCR cycles typically being used. The supernatants from each separate PCR reaction when then combined into a single tube for each library and DNA was purified using a 0.9× ratio of NEB sample-purification beads (NEB, E7103S). The library fragment size was determined using the Agilent 2200 Tapestation with D1000 HS reagents. Micro-C libraries were sequenced by Edinburgh Genomics on the Illumina NovaSeq platform using an SI or SP flow cell to a depth of approximately 1 billion read pairs per cell type using 50 bp paired-end settings.

### Immunostaining

Cells were fixed in PBS containing 4% paraformaldehyde for 10 min at room temperature. Fixed cells were permeabilized with 0.1% Triton X-100 for 10 min at room temperature and blocked with 4% donkey or goat serum (Sigma-Aldrich) in PBS for at 60 min at room temperature, or overnight at 4 °C. Blocked cells were incubated overnight in blocking buffer (4% serum in PBS) containing an appropriate concentration of antibodies (Supplementary Table 2). Antibody-stained cells were washed three times with TBST (20 mM Tris-HCl, pH 7.4, 0.15 NaCl, 0.05% Tween-20) before being incubated with the appropriate secondary antibodies in blocking solution for 2 h at room temperature. Nuclei were stained with 3 mg ml$^{-1}$ 4,6-diamidino-2-phenylindole (DAPI) (Invitrogen, Thermo Fisher Scientific) for 10 min at room temperature. Fluorescence images were taken using the IRIS Digital Cell Imaging System (Logos Biosystems) and visualized using ImageJ[51]. Infection efficiency quantification was performed by counting TF-positive nuclei as the percentage of DAPI-positive nuclei across multiple images.

For CDX2-positive iTS cell colonies, cells were fixed in 4% paraformaldehyde in PBS for 20 min, rinsed three times with PBS, blocked for 1 h with PBS containing 0.1% Triton X-100 and 5% FBS, and incubated overnight in PBS containing 0.1% Triton X-100 and 1% FBS with anti-CDX2 (Biogenex, CDX2-88, 1:500). The cells were then washed three times with PBS, incubated in PBS containing 0.1% Triton X-100 and 1% FBS with the relevant (Alexa) secondary antibody (1:500 dilution) for 1 h. DAPI (1:1,000) was added for the last 10 min of incubation. The cells were washed three times with PBS and visualized under a fluorescence microscope (Nikon eclipse Ti).

### Co-IP

Reprogrammed cells at 48 h were lysed with lysis buffer (100 mM Tris-HCl, 300 mM NaCl, 2% Triton X-100, 0.2% sodium deoxycholate, 10 mM CaCl$_2$) supplemented with EDTA-free protease inhibitor (Roche, 11873580001) for 20 min on ice. The lysate were then centrifuged for 20 min at 14,000 rpm to get rid of the cell debris, then the supernatant containing the proteins was precleared by adding Dynabeads (A and G mix) (Invitrogen, 10004D/10002D) and incubating at 4 °C for 1 h on a shaker. The precleared supernatant was then incubated overnight with pre-bound Dynabeads (A and G mix) using anti-TFAP2C (Abcam, ab110635), anti-ESRRB (Perseus Proteomics, PP-H6705-00), anti-EOMES

(Abcam, ab3345) or anti-IgG (Santa Cruz, sc-2025, sc-2027). The samples were then washed twice with ice-cold lysis buffer, the Dynabeads with the protein complexes were resuspended with sample buffer and boiled for or 10 min at 100 °C and subjected to western blot analysis. Blots were probed with the following primary antibodies: anti-TFAP2C (Abcam, ab110635) and anti-MYC (Abcam, ab32072) and the appropriate IgG-HRP secondary antibody (1:10,000) and visualized using the ECL detection kit.

## Western blotting

Whole-cell extracts were prepared from doxycycline-induced and uninduced MEFs using RIPA extraction buffer (25 mM Tris HCl pH 7.6, 150 mM NaCl, 1% Na-deoxycholate, 1% NP-40, 0.1% SDS) supplemented with cOmplete ultra protease inhibitor and Pierce phosphatase inhibitor cocktail (Thermo Fisher Scientific, A32957).

The protein concentrations of the lysates were quantified using the Pierce BCA Protein Assay Kit according to the manufacturer's instructions (Thermo Fisher Scientific). Proteins resolved by SDS-polyacrylamide gel electrophoresis were electroblotted onto a PVDF membrane. Membranes were blocked overnight in PBST with milk (0.1% Tween-20, 10% non-fat dry milk overnight) at 4 °C with rocking. Membranes were washed three times for 5 min with PBST on a rocker at room temperature. The primary antibody incubations were performed for 4 h at room temperature diluted into PBST 5% BSA (Supplementary Table 2). Membranes were washed three times for 5 min with PBST on a rocker at room temperature. Secondary antibody incubations were carried out PBST 10% non-fat dry milk for 2 h at room temperature with rocking followed by three washes with PBST. Blots were visualized by using SuperSignal West Pico Chemiluminescent Substrate (Thermo Fisher Scientific) using Amersham Hyperfilm ECL (GE Healthcare) developed in Mi5 Processor (Jet X-Ray).

Histone proteins were isolated from MEF129 cells and TNG-MKOS-MEFs, after 72 h of doxycycline induction, and uninfected cell line control, or 144 h of H1.4 shRNA infection, and empty vector infected cell line control, by extraction with 0.2 N sulfuric acid, as previously described[52,53]. In brief, cells were resuspended in a 0.3 M sucrose buffer and nuclei were obtained using a Dounce homogeniser. Nuclei were lysed using a high-salt buffer containing 0.35 M KCl, and then histones were dissolved using 0.2 N sulfuric acid, subsequently precipitated with ethanol and finally resuspended in nuclease-free water.

The protein concentrations of the acid extracted histones were quantified using the Pierce BCA Protein Assay Kit according to the manufacturer's instructions (Thermo Fisher Scientific). Proteins resolved by SDS–polyacrylamide gel electrophoresis were electroblotted onto a PVDF membrane at 200 mA for 2.5 h. Membranes were blocked for 4 h in PBST with milk (0.1% Tween-20, 10% non-fat dry milk) at room temperature with rocking. Membranes were washed once with PBST on a rocker at room temperature. The primary antibodies against H1, and the H3 loading control were diluted into PBST 5% BSA (Supplementary Table 2) and incubated overnight at 4 °C. Membranes were washed six times for 5 min, and once for 10 min with PBST on a rocker at room temperature. Secondary antibody incubations were carried out in PBST 5% BSA for 1 h at room temperature with rocking followed by six washes for 5 min and one wash for 10 min with PBST. Blots were visualized by using SuperSignal West Pico Chemiluminescent Substrate (Thermo Fisher Scientific) and a BioRad ChemiDoc imager on the white tray using the chemiluminescent setting. A list of antibodies used in this study is provided in Supplementary Table 2.

## EMSA

To prepare the cell lysates, MEFs (WT 129) were infected with doxycycline-inducible lentiviruses encoding the TF of interest and overexpressed in for 48 h by doxycycline treatment (see the lentivirus protocol above). In total, 10 million cells were collected for each preparation. Cells were then lysed in buffer A (10 mM HEPES pH 7.5,

1.5 mM $MgCl_2$, 10 mM KCl, 0.5 mM DTT) on ice for 10 min and dounced 40× (tight dounce). The cells were then pelleted and resuspended in 100 µl of buffer B (20 mM HEPES pH 7.5, 30% glycerol, 420 mM NaCl, 1.5 mM $MgCl_2$, 0.2 mM EDTA, 0.5 mM DTT) per each 10 million cells, and incubated for 30 min at 4 °C. After spinning, the supernatant was dialysed for 2 h at 4 °C in dialysis buffer (20 mM HEPES pH 7.5, 30% glycerol, 100 mM KCl, 0.83 mM EDTA pH 8, 1.66 mM DTT, 0.2 mM PMSF). The cell lysates were aliquoted and stored in −80 °C after flash freezing in liquid nitrogen until use for EMSA.

For EMSA, Cy5-end-labelled oligonucleotide duplexes (50 nM) were prepared as described previously[12]. The Cy5-end-labelled oligonucleotide duplexes were mixed with the increasing cell lysates (0.5 µl to 4 µl) and non-specific competing poly(G-C) oligonucleotides (1 µg) in binding buffer (50 mM Tris HCl pH 7.5, 5 mM $MgCl_2$, 50 µM $ZnCl_2$, 50 mM KCl, 5 mM DTT, 25% (v/v) glycerol, 2.5 mg ml⁻¹ BSA) to a final volume of 10 µl and incubated in the dark at 21 °C for 1 h. For the EMSA-supershifts, 5 µg of antibody or 20× of non-labelled oligonucleotide competitor was mixed with TF–DNA mixture in binding buffer and incubated for 20 min at room temperature. The full volume was run on a 5% polyacrylamide gel at 90 V and 100 mA for 4 h in 0.5× TBE (45 mM Tris-borate, 1 mM EDTA) and imaged detecting Cy5 fluorescence using the Bio-Rad Chemi-Doc MP (Bio-Rad).

## Flow cytometry

For flow cytometry analysis, cells were first trypsinized and then neutralized with medium containing 10% fetal bovine serum (FBS). The cells were next centrifuged and washed twice with PBS to ensure the removal of any residual trypsin and medium. The washed cells were then resuspended in PBS for subsequent analysis.

The fluorescence markers eGFP and tdTomato were used to identify and quantify specific cell populations. Flow cytometry analysis was performed using the Beckman Coulter (Gallios) flow cytometer. Data acquisition and analysis were conducted using the Kaluza Software (v.1.0.14029.14028).

To remove dead cells, all of the samples were initially gated using the FSC-A/SSC-A gating to identify the live-cell population (below 200 FS Area). To remove cell doublets, single cells were selected by gating forward scatter height versus area. The positively fluorescent cells were gated based on the fluorescence intensity of positive control cells. Examples of the gating strategy for eGFP and tdTomato are shown in Supplementary Fig. 2.

## DNA constructs

The plasmids constructed in this study are as follows:

The pFUW-TetO-hEsrrb plasmid was generated by PCR amplifying human ESRRB from pPB-PGK-hEsrrb (Addgene, 60434)[54] and inserting the amplified fragment into an EcoRI digested FUW-tet-O-hOct4 plasmid (Addgene, 20726) backbone using an IN-Fusion HD Cloning Plus kit according to the manufacturer's instructions (Takara Clontech) and the following primers: 5′-GCCTCCGCGGCCCCGAATTCGCCAC CATGTCCTCGGACGACA-3′; and 5′-ATAAGCTTGATATCGAATTCT TATTACATGGTGAGCCAGAGATGCTT-3′.

The H1.4 cDNA was generated synthetically by Twist Bioscience and inserted into the pET-28a(+) bacterial plasmid. H1.4 cDNA was then amplified by PCR using the pET28-H1.4 construct as a template and primers containing EcoRI site and Kozak fragment (forward) and XbaI restriction site (reverse) (5′-CCCCGAATTCGCCACCATGTCCGAGACT GCGCCT-3′ and 5′-TATCTCTAGACCTACTTTTTCTTGGCTGCCGCC-3′). The PCR product was digested with EcoR1 and XbaI and ligated into a linear FUW-TetO plasmid digested with the same enzymes.

The dual PiggyBac reporter (PB-TAP-InsX3-Nanog_enh-eGFP-Nanog_flip-tdTomato) plasmid was constructed according to the following steps:

(1) The PB-TAP-InsX3-eGFP- ccdB plasmid was constructed by first removing the Tet-ON-CMV promoter and AttR1 from a

PB-TAP-InsX2-Tet-ON-ccdB plasmid (provided by K. Kaji) using XhoI and NotI digestion. Then, eGFP-poly(A), chicken β-globin insulator and AttR1 PCR products were inserted in that order into the linear PB-TAP-InsX2-ccdB plasmid by Gibson assembly using the IN-Fusion cloning kit (Takara). The resulting construct was used to transform One Shot ccdB Survival 2 T1R chemically competent cells (Invitrogen). PB-TAP-InsX3-eGFP-ccdB was purified and validated by restriction enzyme digestion and Sanger sequencing. The eGFP-poly(A) gene was amplified from pConditional-pac-eGFP plasmid (K. Kaji laboratory), introducing XhoI into the forward primer and AatII into the reverse primer. Chicken β-globin insulator, and AttR1 were amplified from the PB-TAP-InsX2-Tet-ON-ccdB plasmid.

(2) The PB-TAP-InsX3-Nanog_enh-eGFP-ccdB plasmid was then constructed by inserting the *Nanog* enhancer–signpost–promoter fragment (-5 kb) upstream of the *eGFP* gene. The *Nanog* enhancer–signpost–promoter was isolated from the pNanog_enh-Luc plasmid (I. Chambers laboratory)[55], by restriction enzyme digestion using SpeI and XhoI. The ligated construct was used to transform One Shot ccdB Survival 2 T1R chemically competent cells (Invitrogen). PB-TAP-InsX3-Nanog_enh-eGFP-ccdB was purified from selected clones using colony PCR. The correct construct was validated by restriction enzyme digestion and Sanger sequencing.

(3) PENTR-Nanog_flip-part1 was constructed by inserting the *Nanog* enhancer, fliped-signpost-1 and fliped-signpost-2 PCR products in that order by Gibson assembly using the In-Fusion kit (Takara) into the Gateway pENTR 2B2 (Thermo Fisher Scientific). The pENTR vector was first linearized with KpnI and NotI to remove the *ccdB* gene insert. The IN-Fusion reaction was used to transform Stellar Competent Cells (Takara), and positive clones were selected by restriction digestion of mini-preps (Qiagen). pENTR-Nanog_flip-part1 with the correct insert was validated by Sanger sequencing.

(4) PENTR-Nanog_flip-part2 was constructed by inserting the fliped-signpost-3, fliped-signpost-4 and *Nanog* promoter PCR products in that order by Gibson assembly using In-Fusion kit (Takara) into PENTR-Nanog_flip-part1 linearized with XhoI (downstream of the *Nanog* enhancer). The In-Fusion reaction was used to transform Stellar Competent Cells (Takara), and positive clones were selected by restriction digestion of mini-preps (Qiagen). pENTR-Nanog_flip-part2 with the correct insert was validated by Sanger sequencing.

(5) PENTR-Nanog_flip-tdTomato was constructed by inserting the tdTomato PCR products into PENTR-Nanog_flip-part2 linearized with EcoRV (downstream of the *Nanog* enhancer–fliped_signpost-promoter element). The IN-Fusion reaction was used to transform Stellar Competent Cells (Takara), and positive clones were selected by colony PCR. pENTR-Nanog_flip-tdTomato with the correct insert was validated by EcoRI/EcoRV digestion and Sanger sequencing. The tdTomato gene was amplified from the pPyCAG-tdTomato-i-puro plasmid (K. Kaji laboratory).

(6) Finally, the PB-TAP-InsX3-Nanog_enh-eGFP-Nanog_flip-tdTomato plasmid was constructed by Gateway technology (Invitrogen). Essentially, PENTR-Nanog_flip-tdTomato was used as the entry vector and PB-TAP-InsX3-Nanog_enh-eGFP-ccdB as the destination vector for the LR recombination using the LR Clonase II enzyme mix (Invitrogen, 11791-020). Successful insertion resulted in replacing the *ccdB* gene in the destination vector with Nanog_flip-tdTomato from the entry vector. The LR recombination reaction was used to transform One Shot Stbl3 chemically competent *Escherichia coli* (Thermo Fisher Scientific) and positive clones were selected by restriction digest of mini-preps (Qiagen). The final construct was validated by whole-plasmid sequencing using Oxford Nanopore technology (Source BioScience). As expected, the Nanog_enh-eGFP and Nanog_flip-tdTomato cassettes were separated by an insulator and flanked by two other insulators.

The rest of the plasmids were obtained from the following sources: the lentivirus plasmids FUW-TetO-hOct4 (Addgene, 20726), FUW-tet-O-hSox2 (Addgene, 20724), FUW-TetO-hKlf4 (Addgene, 20725), FUW-TetO-hMyc (Addgene, 20723) FUW-TetO-mSox9 and FUW-TetO-mGata4 (Addgene, 41084) were generated in the R. Jaenisch laboratory[56,57]. FUW-TetO-Gata3, FUW-TetO-Tfap2c and FUW-TetO-Eomes were generated as described previously[5]. The pWPT-rtTA2M2 vector was generated in the K. Zaret laboratory[58]. The pFUW-TetO-mBrn2 (Addgene, 27151) vector was generated by the Wernig laboratory[22]. A set of four hairpin shRNAs against the *Hist1h1e* gene (H1.4) in the pLKO.1 lentiviral vector were designed by The RNAi Consortium (TRC)[59], and obtained from Horizon/ Dharmacon. The empty pLKO.1 plasmid was obtained from Addgene (8453)[60]. The pCMV-hyPBase was obtained from the Kaji laboratory[61]. A list of all of the DNA constructs used in this study and their sources is provided in Supplementary Table 3.

## Integration of mES cells with piggyBac transposon vectors

Two days before nucleofection, a near-confluent (70–80%) mES cell culture was split at a 1:10 ratio. For each nucleofection, $2 \times 10^6$ mES cells were prepared. For each nucleofection, one 15 ml falcon tube with 9.5 ml warm medium was prepared. After washing with PBS, mES cells were treated with 0.25% trypsin EDTA and incubated for 2–3 min at 37 °C. Trypsin was inhibited by adding serum–medium and mES cells were collected by centrifugation for 3 min at 300 rcf. The cell pellet was washed with PBS and centrifuged for 3 min at 300 rcf. In a 1.5 ml tube, 1 μg of pBase and 1 μg of the PB vector were mixed (high-quality plasmids with concentrations between 0.5–2 μg μl$^{-1}$ to keep volumes below 10 μl are required). The nucleofection mixture was prepared by adding 90 μl nucleofector solution and 20 μl supplement 1 to the plasmid mix (pBase and PB vector) using the Mouse Embryonic Stem Cell Nucleofector Kit (Lonza, VAPH-1001). The mES cell pellet ($2 \times 10^6$ cells) was resuspended quickly in nucleofection mix. The cell suspension was then transferred into a cuvette without introducing bubbles (bubbles will short the electric current and negatively affects cell viability). The cuvette was placed into the Nucleofector machine (Amaxa Biosystems) and pulsed with the program A-023. The cuvette was quickly brought to the tissue culture hood and 500 μl prewarmed media was added. The cell suspension was removed from cuvette using the Lonza Pasteur pipettes and transferred to the prepared 15 ml falcon tube with 9.5 ml warm medium. The cell suspension was plated onto a gelatine-coated 10 cm dish. The cells were incubated at 37 °C and culture medium was changed every 2 days. Green (eGFP) and red (tdTomato) fluorescence was checked under the microscope and cells were sorted by fluorescence-activate cell sorting as explained below.

## Transgene silencing analysis

For transgene expression analysis, total RNA from the indicated samples was extracted using the Macherey-Nagel kit (Ornat). Between 500 and 2,000 ng of total RNA was reverse transcribed using the iScript cDNA Synthesis kit (Bio-Rad). qPCR analysis was performed on three biological replicates ($n = 3$), using 1/100 of the reverse transcription reaction in a StepOnePlus Real-Time PCR System (Applied Biosystems) with the SYBR Green Fast qPCR Mix (Applied Biosystems).

Specific primers were used to exclusively detect transgene expression. For the genes *Gata3*, *Eomes*, *Tfap2c*, *Myc*, *Esrrb* and *Sox2*, primers targeting the last exon (forward primer) and the WPRE element of the FUW-TetO vector (reverse primer) were used. For the genes *Oct4* and *Klf4*, primers targeting the first exon (reverse primer) and the beginning of the viral vector (TetO, forward primer) were utilized. The amount of cDNA in each sample was normalized to the level of the housekeeping control gene *Gapdh*. A list of the primers used in this study is provided in Supplementary Table 4.

## H1 overexpression (OE)

For infection, HEK293T cells were seeded at a density of $2.4 \times 10^6$ cells per 15 cm plate and grown in 30 ml HEK medium for 24 h, before

being transfected with rtTA2 or H1.4 lentivirus plasmids. Each virus was prepared in a separate dish. For transfection, 2.4 µg pMD.G, 5.1 µg psPAX2 and 7.5 µg of the corresponding plasmid were dissolved in 1,710 µl Opti-MEM medium (Thermo Fisher Scientific, 31985062) and 90 µl Fugene 6 reagent (Promega, E2692), thoroughly mixed by vortexing and incubated for 15 min at room temperature, then added to the 15 cm plate containing HEK293T cells, which were incubated for 16 h. The transfection medium was replaced with fresh HEK medium, and the transfected cells were cultured for a further 60 h. The lentiviruses were collected by collecting the 30 ml supernatant, which was passed through a 0.45 µm polyethersulfone filter-fitted syringe. The virus was then pelleted by ultracentrifugation in Ultraclear 38.5 ml centrifuge tubes at 25,000 rpm (77,000$g$), using the Beckman Coulter OptiMAXPN-80 ultracentrifuge and the SW32-Ti swinging-bucket rotor (Beckman Coulter) for 2.5 h at 4 °C. The supernatants were removed, and the viral pellets were dissolved in 300 µl GMEM by swirling and then aliquoted the same day and stored at −80 °C. On average, the titre of each virus was determined to be around $5 \times 10^7$ infection units per ml.

MEF129 (passage 2) were seeded at 25,000 cells per cm$^2$ 24 h before infection, two 10 cm dishes were used (1.4 million cells per plate) for H1 infection for Omni-ATAC and western blotting, to confirm over-expression compared with uninfected cell line controls. Seven 15 cm dishes (3.6 million cells per plate) were used for Micro-C. The next morning, the medium was changed to MEF medium supplemented with 8 µg ml$^{-1}$ polybrene and pFUW-TetO-H1.4 and pWPT-rtTA2M2 viruses at a MOI of 2. Then, 24 h after infection, the medium on all plates was changed to fresh MEF medium. Next, 48 h after infection, the expression of H1.4 was induced by the addition of MEF medium containing doxycycline to a final concentration of 2 µg ml$^{-1}$ and the cells were incubated for 72 h. Next, 72 h after doxycycline induction, the infected and uninfected 10 cm plates were collected by trypsinization and counted using a haemocytometer. In total, 400,000 cells from the infected and uninfected samples were immediately subjected to the Omni ATAC protocol[50] (see the 'ATAC−seq' section (libraries and sequencing for H1 OE and KD experiments) below) while the remaining cells were acid extracted for HPLC quantification and western blot (see the 'Western blotting' section). The 15 cm plates were subjected to double cross-linking for Micro-C (see the 'Micro-C' section (MNase digestion and ligation)).

## H1 knockdown (KD)

For infection, HEK293T cells were seeded at a density of $2 \times 10^6$ cells per 15 cm plate and grown in 30 ml HEK medium for 24 h. Double the number of plates was seeded for each 15 cm plate of MEFs, due to two rounds of infection using viral supernatant (VSN), in total. Twenty two 15 cm plates of HEK cells were used for H1.4-targetting shRNA and two 15 cm plates of HEK cells were used for the empty vector control. For transfection, 2.4 µg pMD.G, 5.1 µg psPAX2 plus either a mixture of 1.875 µg each of the four H1.4- targeting shRNA plasmids (Horizon Discovery TRC-ID TRCN0000096935: TTTGGCCGCTTTAGG CTTTAC, TRCN0000096936: TTGACGGGTGTCTTCTCGGCG, TRCN0 000096937: TCTTAGCCTTAGTTGCCTTTG, TRCN0000096938: TAG CTGCCTTAGGCTTGGAGG) together or 7.5 µg of the empty pLKO plasmid were dissolved in 1,710 µl Opti-MEM medium (Thermo Fisher Scientific, 31985062) and 90 µl Fugene 6 reagent (Promega, E2692). The shRNA and empty transfection mixes were thoroughly mixed by vortexing, and incubated for 15 min at room temperature, before adding to the 15 cm plates containing HEK293T cells. After 16 h of incubation with the transfection mixes, the medium was replaced with fresh HEK medium, and the transfected cells were cultured for a further 60 h. The lentiviruses were collected by collecting the 30 ml supernatant, which was passed through a 0.45 µm PVDF filter unit (Stericup Millipore) and supplemented with 8 µg ml$^{-1}$ polybrene. Half of the VSN was flash-frozen in liquid nitrogen and stored at −80 °C.

At 24 h before the virus was collected, 2.8 million of MEF129 cells (passage 2) were seeded into two 10 cm dishes (density, 25,000 per cm$^2$) for the empty vector control to be used for Omni-ATAC and western blotting. Ten 15 cm dishes (3.6 million cells per plate) were infected with H1.4-targeting shRNAs for Micro-C, Omni-ATAC and western blotting. For infection, 25 ml of H1.4-targeting shRNA VSN was added per 15 cm plate, with an additional 15 ml of MEF medium, supplemented with 8 µg ml$^{-1}$ polybrene. A total of 8.5 ml of empty pLKO vector VSN was added per 10 cm plate, with an additional 5 ml of MEF medium, containing 8 µg ml$^{-1}$ polybrene. The remaining VSN, with 8 µg ml$^{-1}$ polybrene, was flash-frozen in liquid nitrogen and stored at −80 °C until a second round of infection after 72 h. At 24 h after infection, the medium was changed for MEF medium with 1 µg ml$^{-1}$ puromycin, to select for pLKO-vector-containing MEFs. Then, 72 h after the initial infection, puromycin selection was paused and a second round of infection was carried out as previously, using ice-thawed VSN and prewarmed at 37 °C. Then, 24 h later, the medium was changed for 1 µg ml$^{-1}$ puromycin-containing MEF medium. Next, 144 h after the initial infection, two of the 15 cm plates that were infected with H1.4-targetting shRNA VSN and both 10 cm plates infected with empty vector control VSN were collected by trypsinization and cells counted using a haemocytometer. About 400,000 cells from the infected and uninfected samples were immediately subjected to the Omni ATAC protocol (see the 'ATAC−seq' section (libraries and sequencing for H1 OE and KD experiments)), and the remaining cells were acid-extracted for HPLC quantification and western blotting (see the 'Western blotting' section). The remaining 15 cm plates, infected with H1.4 shRNA were subjected to double cross-linking for Micro-C (see the 'Micro-C' section (MNase digestion and ligation)).

## OSKM reprogramming with H1.4 KD and H1.4 OE

H1.4 OE and KD was performed in TNG-MKOS-MEFs[62] for ATAC−seq similarly to that for MEF129 WT cells (see the 'H1 overexpression' and 'H1 KD' sections). In brief, TNG-MKOS-MEFs (passage 3) were seeded at 27,000 cells per cm$^2$ 24 h before viral infection for H1.4 OE, (seven 10 cm plates for H1 overexpression and three 10 cm plates for uninfected control). Cells were infected with pFUW-TetO-H1.4 and pWPT-rtTA2M2 viruses at a MOI of 2. The medium was changed the next day and viral gene expression was achieved by administering doxycycline (2 µg ml$^{-1}$) 48 h after infection to infected and uninfected cells. ATAC was performed on samples at 0 h of induction and 72 h after induction, for the infected and uninfected samples. Western blotting was performed on histone extractions 72 h after doxycycline, to confirm successful overexpression, compared with the uninfected cell line control.

To achieve H1.4 KD, HEK293T cells were seeded, 72 h before MEFs, then, 24 h later, were transfected to make VSN of the H1.4 shRNA pool (see the 'H1 KD' section above). TNG-MKOS-MEFs (passage 3) were seeded at 21,500 cells per cm$^2$ on five 10 cm plates 24 h before infection. Three 10 cm plates were infected with 10 ml of H1.4 shRNA VSN, and two were infected with 10 ml of empty vector VSN; the remaining VSN was flash-frozen in liquid nitrogen and stored at −80 °C. All VSN was supplemented with 8 µg ml$^{-1}$ polybrene, and an additional 5 ml of fresh MEF medium with polybrene was added for all plates. The medium was changed for fresh MEF medium with 1 µg ml$^{-1}$ puromycin. Then, 48 h later, one plate infected with H1.4-shRNA-VSN and one plate infected with empty-vector-VSN were collected for the 0 h doxycycline timepoint of the ATAC experiment. The remaining plates were infected for a second time with VSN, as before, with the addition of doxycycline to a final concentration of 2 µg ml$^{-1}$ to induce the expression of MKOS. The medium was changed the next day to 2 µg ml$^{-1}$ doxycycline-containing MEF medium. Then, 72 h after beginning doxycycline induction, the remaining plates were collected for ATAC (see the 'ATAC−seq' section (libraries and sequencing for H1 OE and KD experiments)).

### GETM reprogramming with H1.4 KD and H1.4 OE

To downregulate H1.4 expression in fibroblasts, four different shRNA sequences targeting H1.4 (PLKO.1 vector) were incorporated into replication-incompetent lentiviruses. Lentiviruses were packaged using a mix of lentiviral packaging vectors (psPAX2 and pGDM.2, ratio: 1:1) and the four shRNAs (ratio, 1:1:1:1) at a ratio of 1:1. The packaging was performed in HEK293T cells, and the VSNs were collected at 48 h after transfection. The supernatants were filtered through a 0.45 μm filter, supplemented with 8 μg ml$^{-1}$ polybrene (Sigma-Aldrich), and used to infect MEFs. Then, 24 h after infection, the medium was replaced with fresh DMEM containing 10% FBS.

Next, 4 days after infection, replication-incompetent lentiviruses containing GETM factors (ratio, 1:1:1:0.3) were similarly packaged and used to infect the H1.4-downregulated cells. Twenty-four hours after this second infection, the medium was replaced with fresh DMEM containing 10% FBS and 2 μg ml$^{-1}$ doxycycline. Two weeks later, the medium was switched to TS cell reprogramming medium (RPMI supplemented with 20% FBS, 0.1 mM β-mercaptoethanol, 2 mM L-glutamine, 25 ng ml$^{-1}$ human recombinant FGF4 (PeproTech), 1 μg ml$^{-1}$ heparin (Sigma-Aldrich) and 2 μg ml$^{-1}$ doxycycline). After 1 week, the medium was replaced with TX medium without doxycycline. Then, 1 week later, the plates were fixed and stained for CDX2 to identify positive colonies.

Similarly, to overexpress H1.4, MEFs were infected with lentiviruses encoding H1.4 using the pFUW-TetO-H1.4 plasmid. The lentiviruses were packaged in HEK293T cells as described above. To initiate iTS cell reprogramming, H1.4 overexpression was induced along with GETM using doxycycline (2 μg ml$^{-1}$) as described above.

### H1 quantification by HPLC

Histone proteins were isolated by extraction with 0.2 N sulfuric acid, as previously described[52]. In brief, cells were resuspended in a 0.3 M sucrose buffer and nuclei were obtained using a Dounce homogeniser. Nuclei were lysed using a high-salt buffer containing 0.35 M KCl, and then histones were dissolved using 0.2 N sulfuric acid, subsequently precipitated with ethanol and finally resuspended in nuclease-free water. Acid-extracted histones were quantified using the Pierce BCA Protein Assay Kit according to the manufacturer's instructions (Thermo Fisher Scientific). Acid-extracted histones were analysed by reversed-phase high-pressure LC using the Waters 2695 system equipped with the Vydac 218TP C18 HPLC column. The effluent was monitored, and peaks were recorded using the Waters 996 Photodiode Array Detector at 214 nm. H1 peak integrations were performed using the Waters Empower Pro software (v.2) and normalized to H2B peaks.

### Mass spectroscopy (MS)

Acid extracts were reduced in 10 mM DTT, 0.02% NP-40 and 100 mM NH$_4$HCO$_3$ at 37 °C for 1 h. The samples were then alkylated with 30 mM IAA for 45 min at room temperature in the dark. The reactions were then desalted into 50 mM NH$_4$HCO$_3$ using ZebaSpin 7k columns (Thermo Fisher Scientific) and the eluates were supplemented with trypsin (0.1 mg ml$^{-1}$) and digested for 2 h at 37 °C. At the end of the 2 h, the samples were supplemented with additional trypsin and the digestions were allowed to proceed overnight. The digestions were quenched with 1% formic acid, dried in SpeedVac and then resuspended in 130 μl MS sample buffer (0.1% formic acid, 1% acetonitrile in water).

### MS instrument settings

LC–MS analyses were performed on the TripleTOF 5600+ mass spectrometer (AB SCIEX) coupled with the M5 MicroLC system (AB SCIEX/Eksigent) and PAL3 autosampler. LC separation was performed in a trap-elute configuration, which consists of a trap column (LUNA C18(2), 100 Å, 5 μm, 20 × 0.3 mm cartridge, Phenomenex) and an analytical column (Kinetex 2.6 μm XB-C18, 100 Å, 50 × 0.3 mm microflow column,

Phenomenex). The mobile phase (phase A) consisted of 0.1% formic acid in water, and phase B consisted of 0.1% formic acid in acetonitrile.

Peptides in MS sample buffer were injected into a 50 μl sample loop, trapped and cleaned on the trap column with 3% mobile phase B at a flow rate of 25 μl min$^{-1}$ for 4 min before being separated on the analytical column with a gradient elution at a flow rate of 5 μl min$^{-1}$. The gradient was set as follows: 0–24 min, 3% to 35% phase B; 24–27 min, 35% to 80% phase B; 27–32 min, 80% phase B; 32–33 min, 80% to 3% phase B; and 33–38 min at 3% phase B. An equal volume of each sample (30 μl) was injected four times, once for information-dependent acquisition (IDA), immediately followed by DIA/SWATH in triplicate. Acquisitions of distinct samples were separated by a blank injection (80 μl MS sample buffer) to prevent sample carryover. The mass spectrometer was operated in positive-ion mode with an EIS voltage at 5,200 V, source gas 1 at 30 psi, source gas 2 at 20 psi, curtain gas at 25 psi and the source temperature at 200 °C.

### IDA and data analyses

IDA was performed to generate reference spectral libraries for SWATH data quantification. The IDA method was set up with a 250 ms TOF-MS scan from 300 to 1,250 Da, followed by MS/MS scans in a high-sensitivity mode from 100 to 1,500 Da of the top 25 precursor ions above the 100 cps threshold (100 ms accumulation time, 100 ppm mass tolerance, rolling collision energy and dynamic accumulation) for charge states (z) from +2 to +5. IDA files were searched using ProteinPilot (v.5.0.2, ABSciex) with the default setting for tryptic digest and IAA alkylation against a protein sequence database.

The *Mus musculus* proteome FASTA file (54,910 protein entries, UniProt: UP000000589) augmented with sequences for common contaminants was used as a reference for the search. Up to two missed cleavage sites were allowed. Mass tolerance for precursor and fragment ions was set to 100 ppm. A false-discovery rate (FDR) of 5% was used as the cut-off for peptide identification.

### SWATH acquisitions and data analyses

For sequential window acquisition of all theoretical mass spectra (SWATH-MS) acquisitions[63], one 50 ms TOF-MS scan from 300 to 1,250 Da was performed, followed by MS/MS scans in a high-sensitivity mode from 100 to 1,500 Da (15 ms accumulation time, 100 ppm mass tolerance, +2 to +5 z, rolling collision energy) with a variable-width SWATH window[64]. DIA data were quantified using PeakView (v.2.2.0.11391, ABSciex) with SWATH Acquisition MicroApp (v.2.0.1.2133, ABSciex) against selected spectral libraries generated in Protein-Pilot. Retention times for individual SWATH acquisitions were calibrated using 23 peptides for core histone H4c1 (UniProt: P62806), which was highly representative in the IDA ion library and all SWATH acquisitions. The following software settings were used: up to 25 peptides per protein, 6 transitions per peptide, 95% peptide confidence threshold, 5% FDR for peptides, XIC extraction window 10 min and XIC width 100 ppm. In all SWATH files, the quantification data for core and linker histone proteins were manually curated to exclude from consideration the peptides that exhibited an aberrant retention time in at least one SWATH acquisition (>20% difference from that in the IDA/ion library or other SWATH acquisitions). Protein peak areas were exported as Excel files and processed as described below.

### Quantification of proteomics data

Quantification of individual H1 subtypes in MEFs was modelled using the combination of relative LC–MS determinations and absolute HPLC quantifications of known mouse embryonic stem cell standards as described previously[65].

### Bioinformatics

**Sequencing data processing and alignment.** Initial quality-control analysis was performed using the FastQC toolkit (https://github.com/s-andrews/FastQC). ES cell H1 ChIP–seq reads and their associated

inputs were trimmed to remove adapters and bases with a phred score of <30 using Cutadapt[66] (cutadapt -a AGATCGGAAGAGCACACGTCTGA ACTCCAGTCA -q 30). ChIP–seq, ATAC–seq and MNase–seq samples were aligned to mouse reference genome MGSCv37 (mm9) using Bowtie2[67] v.2.3.4.1, using a --very-sensitive call and paired-end settings (or single-end settings where appropriate). Aligned reads were sorted and subsequently converted to BAM format using the samtools suite[68]. RNA-seq samples were aligned using STAR (v.2.7)[69] with --outFilterMultimapNmax 1. Duplicated reads were eliminated using the Picard (https://github.com/broadinstitute/picard) function Mark-Duplicates, except for MNase–seq and RNA-seq, for which duplicates were retained. Sequencing replicates were merged using samtools merge. The sequencing coverage and the insert size distribution were measured from the resulting BAM files using Qualimap (v.2.2.1)[70].

Micro-C libraries were aligned to the mm9 reference genome and processed using the Nextflow (https://www.nextflow.io/) pipeline distiller-nf (https://github.com/open2c/distiller-nf) using the following configurations; make_pairsam = False, drop_readid = False, parsing_options: '--add-columns mapq --walks-policy mask', max_mismatch_bp = 1. Balanced multi-resolution cool (mcool) files were outputted with the following bin sizes: 10,000,000, 5,000,000, 2,500,000, 1,000,000, 500,000, 250,000, 100,000, 50,000, 25,000, 10,000, 5,000, 2,000, 1,000, 500, 100. 15U and 20U Micro-C libraries for each cell type were merged using pairtools merge (https://github.com/open2c/pairtools).

**ChIP–seq peak calling.** ChIP–seq narrowPeaks and summits showing significant enrichment over input DNA were called using MACS2 (v.2.1.1.20160309)[71], and were controlled to a *q*-value (minimum FDR) cut-off of 0.01. To identify broadPeaks of TF binding, peaks were called using MACS2 with the following flags: -B --broad-cutoff 0.1 --broad --nomodel --extsize 200. Regions that overlapped with the ENCODE blacklist[72] were removed using the bedtools[73] intersect function (flag --v).

**MNase peak calling.** To obtain a consensus list of nucleosome positions, the alignments for each MNase concentration were merged into a single BAM file using samtools merge. Nucleosome and nucleosome dyad positions were called using the DANPOS2[74] function dpos with a 1% FDR, paired-end settings and bin size of 1 bp to ensure dyad position accuracy. A file of nucleosome dyad positions was then generated by taking the summit position and adding 1 to create a bed file of 1 bp chromosome coordinates. The smoothened.wig file of MNase signal from DANPOS2 was converted to bigwig using wigToBigWig (http://hgdownload.cse.ucsc.edu/admin/exe/linux.x86_64/) and used for heat maps and profiles of MNase signal.

**Read density analysis.** The aligned reads (BAM files) were normalized for sequencing coverage to 1× genome depth (RPGC) using the bamCoverage tool from DeepTools2[75] with a bin size of 10 bp and extendReads parameter, chromosome X was ignored for normalization. The resulting bigwig files were converted to wig format using the UCSC bigWigToWig tool[76], and subsequently converted into a bed file using the wig2bed[77]. To sort peaks of individual TFs based on either ChIP–seq or ATAC–seq enrichment, 1 bp summits produced by MACS2 were extended by 150 bp on each side to produce a 301 bp peak using the bedtools slop function[73]. The tag density under these peaks was then quantified using the bedmap function of BEDOPS[77], against the RPGC-normalized bed file of either the ChIP or ATAC samples. Peaks were then sorted from highest-to-lowest enrichment using the UNIX command line sort function.

For ATAC–seq-sorted peaks, peaks were split based on RPGC to open (>20 RPGC) or closed chromatin (<20 RPGC), representing the value whereby no ATAC enrichment is observed within the central 301 bp peak over the flanking 350 bp either side (total region of 1 kb). As there is no input DNA for ATAC–seq, we compared the enrichment of ATAC–seq

within the peak to a 1 kb local region. By plotting ATAC–seq enrichment of TF sites as function of number of reads (sequence coverage normalised in RPGC or reads per genome coverage), we identified the baseline of 20 RPGC.

To generate the read density heat maps and line profiles, we first computed a density matrix using the DeepTools2 tool computeMatrix reference-point and the following parameters: --referencePoint center, --binsize 10, -b 1000 -a 1000, --sortregions keep, --missingDataAsZero and --averageTypeBins sum using the peak bed files as reference files (-R) and the normalized ChIP–seq and ATAC–seq bigwig files as score files (-S)[75]. The ENCODE blacklist was excluded. The resulting matrix was subsequently used to generate heat maps and profiles using Deeptools2 functions plotHeatmap and plotProfile, respectively[75].

Histone H1 ChIP–seq data were processed as described above except that the Deeptools2 function bigwigCompare[75] was used to subtract the RPGC-normalized input signal from the RPGC-normalized H1 ChIP–seq signal. ChIP–seq data for H1c and H1d[78] were merged for analysing H1 in ES cells to obtain maximum coverage of H1-bound regions in ES cells.

Profiles of Micro-C contact junctions around TF sites were produced by generating a bed file containing 1 bp coordinates for each junction in a '.pairs' contact file generated by the distiller-nf pipeline. This was then used to generate a genome coverage bedgraph using the bedtools[73] function genomecov before being subsequently converted to a bigwig file using bedGraphToBigWig (http://hgdownload.soe.ucsc.edu/admin/exe/linux.x86_64/). This bigwig file was then used as a sample file with Deeptools2[75].

**Genomic intervals.** To assess peak overlaps between conditions (but not co-bound sites), all peaks were considered as 301 bp centred round the summit. This is because the average peak size was identified by MACS to be ~300 bp, and one nucleotide was added to place the summit in the middle. Overlapping peaks between conditions were identified using the Intervene venn function with the flag --save-overlaps[79], such that regions would be called as overlapping based on a 1 bp or greater overlap. Bar plots were generated by counting the number of peaks in each list. For comparison of MYC peaks within closed and open chromatin across all reprogramming systems, intersection over union or the Jaccard index was measured using the bedtools jaccard function and ggplot2 was used to generate the resulting heat map[73]. Peaks were assigned to transcription start sites using the GREAT tool available online with mm9 association settings for 'Single Nearest Gene' with a maximum distance of 1,000 kb[80].

To quantify ATAC–seq on co-bound TF sites, 301 bp peaks for each TF were labelled with a single-letter identifier for each TF and combined into a single file using bedops --everything[77]. Bedtools merge was used to collapse each overlapping peak with a --distinct settings used for the single letter label column to label each peak with the letter code for each TF present (that is, 'OS' for OCT4 and SOX2), awk was then used to count the number of TFs present by counting the number of letters. RPGC-normalized ATAC–seq data on these merged peaks were then quantified using bedmap --echo and the value was scaled by dividing by the peak width, to account for variability in peak size. These values were then used to generate violin plots using the ggplot2 functions geom_violin() and geom_boxplot().

To generate lists of TF sites distal or proximal to MYC, closed chromatin peaks of all TFs within a combination were combined using bedops --everything without merging overlapping peaks. The master peak list was used as an input in the bedtools window with MYC peaks from that condition as the −b file, and a −w flag of 350 to ensure detection of nearby MYC peaks. Proximal or distal sites were then obtained with the --u or --v flags, respectively.

**Motif discovery.** De novo motif analysis was performed using the MEME suite installed on a local Linux server[81]. First, the DNA sequences (FASTA) were generated from the central 200 bp of the ChIP–seq peak regions

using bedtools getfasta[73]. To use as the background, DNA sequences (200 bp) were extracted from genomic regions located 1 kb upstream from the summit of each peak using bedtools shift[73]. All regions were filtered through the ENCODE blacklist. Finally, meme-chip was run using the Fasta sequence files and the corresponding Markov model and the following parameters: -nmeme 600, -meme-mod zoops, -meme-minw 6, -meme-maxw 18, -meme-maxsize 50000000, -dreme-e 0.00001, -dreme-m 20 using the JASPAR core motif database[82]. The most enriched de novo motifs discovered by MEME[83] and DREME[84] were analysed using CentriMO to confirm their central enrichment over the background sequences and compared to the canonical motifs using Tomtom.

**Gene expression analysis.** Gene expression quantification was performed using the featureCounts function of the R subRead package[85], using a gtf file containing the UCSC genes for mm9 with paired or single-end settings depending on the samples. Tables generated for the paired and single-end data were combined using cbind(). Differential gene expression analysis was performed using the package DESeq2 (v.1.22.2)[86] with DESeqDataSetFromMatrix() followed by DESeq2(). Genes with 0 counts in all of the conditions were excluded and the samples were normalized according to library size using sizeFactors(). Values then underwent regularized log-transformation with rlog() and counts were obtained using assay(). Pearson correlation analysis was performed using the top 500 most variable genes with cor() with method=c("pearson") followed by package pheatmap(). A PCA plot was generated using plotPCA() on the regularised log transformed matrix. Differentially expressed genes at 72 h in each of the early reprogramming systems were identified using the results() function in DESeq2 using a contrast versus MEFs, lfcThreshold=1, altHypothesis="greaterAbs" and alpha = 0.05.

To perform upset analysis of DEGs, unique DEG gene IDs were combined into a dataframe in R. This was then used as the input for the function upset() from the package upsetR. DEGs targeted by MYC were identified by taking the gene IDs from the output of the GREAT analysis of ChIP peaks and finding matching gene IDs in the DEG lists with join(). These were combined into a data.frame and plotted with upset().

To analyse TF enrichment at differentially expressed genes, the gene IDs of differentially expressed genes were combined with a list of coordinates of transcription start sites for the mm9 genome using UCSC refGene TSS mm9 coordinates of seqMINER[87] with join(), and a bed file was generated. Approximately 4–5% of genes per set did not have matching gene IDs due to release differences in the annotation and were excluded. The Deeptools function plotProfile was used to plot TF enrichment as described for the ChIP/ATAC–seq analysis.

To define whether differentially expressed genes were targets of a specific TF, the nearest gene from each TF summit was obtained using GREAT. This gene list was compared with the list of differentially expressed genes using join() such that each gene appeared once in a final list of genes that are both TF targets and differentially expressed. Overlaps were identified using the package UpsetR[88].

**MNase fragment-size maps.** Fragment-size enrichment heat maps were drawn using plot2DO[89] with ChIP–seq peak summits or TSS as a reference and the aligned raw BAM files as the sample. Only fragment sizes between 50 and 250 bp were considered. Heat-map scales were scaled to the same value between open and closed chromatin to allow for direct comparison of fragment enrichment.

**Identifying TF-bound nucleosomes.** Bound nucleosomes were identified by selecting the closest 1 bp nucleosome dyads to ChIP–seq summits using the closest features function from bedops[77] (closest-features --delim '\t' --dist --closest). Nucleosomes where the ChIP–seq summit was greater than 80 bp from the dyad were filtered out using awk and the remaining nucleosomes were labelled with a column containing a single letter identifier for that TF (that is, 'O' for

OCT4). Co-bound mononucleosomes were identified by combining the lists of bound nucleosome dyads for each individual TF and merging using bedtools merge[73], such that the single-letter TF label column would contain multiple identifiers if the same dyad was present in each list of bound nucleosomes. The number of TFs present on each nucleosome was counted by using awk to count the number of characters in this column.

**Motif position analysis on TF-bound nucleosomes.** Bound nucleosome dyad positions were used to generate a 1 bp GRanges object[90]. IRanges[90] was used to extend this object to 160 bp, representing our average nucleosome fragment size. Sequences were obtained for the positive strand using the BSgenomes function getSeq(). The position weight matrix (pwm) was obtained from the MEME-ChIP[91] and used to scan each strand separately for nucleosome sequence using the seqPattern[92] function motifScanHits() with 100% match score. To count motifs in the reverse orientation, the pwm was passed through Biostrings function reverseComplement() before scanning. The motif count for each strand was then assigned to the corresponding nucleosome dyad by counting the number of times that each sequence identifier appeared in the motifScanHits() output. Total motif counts were obtained by summing the values for the positive and negative strands.

To generate heat maps of motif density, nucleosome dyads were extended symmetrically by 500 bp in each direction using IRanges[90]. An image matrix for each strand was generated using the function PatternHeatmap() of the R package heatmaps[93] with the pwm and minimum score between 80 and 95% depending on the motif lengths (shorter motifs used a higher match score)[92]. To generate a matrix for the reverse orientation of a the motif, the sequences on the positive DNA strand was queried using the pwm reverse complemented using the Biostrings function reverseComplement(). Kernel smoothing was applied to the matrix using smoothHeatmap(). To plot both strands together, the matrix produced for motif reverse complement was multiplied by −1. Positive and negative matrices were converted to data frames and then combined using rbindlist() from the package data. table (https://github.com/Rdatatable/data.table) by alternating lines according to the row number in the data frame such that, for every line of positive-strand scores on a sequence, the next line is corresponding scores for the motif reverse complement on that same sequence. The combined data frame was then converted back to a matrix using data. matrix(). Heat maps were then plotted using the R package heatmaps functions Heatmap() and plotHeatmapList().

To generate density plots of motif position around the dyad, the positive and negative strands were considered independently. Dyad positions were extended symmetrically by 200 bp using IRanges[90], and sequences were obtained using getSeq(). The seqPattern function plotMotifOccurrenceAverage() was used with the MEME pwm and its reverse complement. A smoothing window of 3 bp was used for plotting. To increase the resolution of motif identification around the nucleosome dyad, only perfect motif matches were considered.

**Identifying TF-bound nucleosome arrays.** To identify TF-bound nucleosome arrays, a column containing a single letter label was added to the broadpeak file for each TF. These broadpeaks were then combined into a single file using bedops --everything. Bedtools merge was used to collapse each overlapping broadpeak with a --distinct settings used to for the single-letter label column to label each peak with the letter code for each TF present (that is, 'O' for OCT4), with awk being used to count the number of TFs present by counting the number of letters. The RPGC-normalized ATAC–seq signal was then quantified on these broadpeaks using bedmap --echo --sum --delim '\t', this value was then scaled by the broadpeak length in kb, and open and closed sites were separated using a read counts per kb cut-off value of 40. The positions of flanking nucleosome dyads were identified using

bedops closest-features (with flags --delim '\t' --dist --no-overlaps) and the array width was obtained by between by subtracting the first coordinate position of the upstream dyad from the downstream dyad using awk. This value was used for sorting on the basis of distance using the UNIX command line sort function. TF combinations were identified by selecting for different letter combinations using an awk equality. Oligonucleosomes were centred on their array midpoint by taking the coordinates of the upstream nucleosome dyad, and shifting them by half the array width using awk. Arrays were centred on the left or right edge by shifting to either the upstream or downstream dyad coordinate. Array width histograms were generated by passing the array width values to the geom_hist() function of ggplot2.

**Motif analysis on TF-bound nucleosome arrays.** Motif analysis for TF-bound nucleosome arrays was performed similarly to mononucleosomes. The seqPattern function plotMotifOccurrenceAverage() with the MEME pwm and its reverse complement were used to generate density plots. A smoothing window of 10 bp was used for plotting and percentage match cut-offs were set between 80 and 95% depending on the motif length. Motif heat maps were generated using the heat-map library as on mononucleosomes.

To identify motif occurrences within arrays, a GRanges object was object was created using the 1 bp array midpoint coordinates and adding metadata columns for the array width, half the array width, a left boundary of (5,000 − half array width) and a right boundary of (5,000 + half the array width). This object was then extended ±5 kb using promoters() and the sequences were obtained using getSeq(). This gives arrays a maximum array size for motif identification of 10 kb but prevents most sequences from extending off the chromosome boundaries. Arrays that were extended off the chromosome boundary were filtered using GenomicRanges:::get_out_of_bound_index() (which occurred for approximately 1 in every 15,000 arrays). This filtering was applied to the 10 kb extended sequences, the 1 bp midpoints and the left boundaries as applicable. An ID column was then generated for each array using seqalong() and added as metadata. To identify motifs occurring within an array (such as for SOX2) motifScanHits() was used to identify motif occurrences on each 10 kb sequence using a MEME pwm (with a 95% match score used for SOX2 motif). This produces a table of the motif positions in which each line contains two columns, sequence ID and the start position of a single motif on that sequence. A left_join() was used to match the motif table with the GRanges object of array midpoints by sequence ID. subset() was used to filter sequences for which the motif start position was outside the array edges (motif_position >= left_boundary and motif_position <= right_boundary). A frequency table was then made to count the occurrence of each sequence ID in the filtered motif table and these counts were appended to the Granges object of array midpoints to produce a column containing motif counts within each array. This process was repeated to add another column motif counts on the bottom strand by passing the motif pwm through reverseComplement(). Motif counts per kb were obtained by dividing the motif count within the array by the array width in kb (after setting the width values for any array >10 kb to 10 kb). This value was used to filter arrays based on motif counts per kb. Count histograms were generated by passing these motif counts per kb to the geom_hist() function of ggplot2. Motif heat maps and profiles were generated as described above for mononucleosomes, and the percentage matches for the motif pwm were typically set between 80 and 95% depending on the motif length and degeneracy.

All of the scripts with nucleosome array positions and motifs have been deposited at GitHub (https://git.ecdf.ed.ac.uk/soufi_lab/motif_nucleosome_arrays).

**Micro-C pileup analysis.** Micro-C pileup analysis was performed using the Coolpup.py package[94]. To generate pile-up heat maps, a bed file containing TF-bound sites and a Micro-C mcool file were used to generate a matrix Micro-C contacts using the --local settings and --ignore_diags set to 0. For 20 kb padding windows around the TF site, 100 bp mcool bins were used and, for 400 kb padding windows, 2 kb mcool bins were used.

**Micro-C loop calling and *cis*-interactions between ChIP–seq peaks.** Statistically significant loops connecting TF-binding sites were called using the FitHiChIP pipeline[95] (https://ay-lab.github.io/FitHiChIP/html/index.html). The following settings were used in the configuration file: COOL= path to.mcool files from the Nextflow pipeline (see above), PeakFile=path to.broadpeak files output from MACS2, BINSIZE=1000, IntType=5, LowDistThr=1000, UppDistThr=2000000, QVALUE=0.01, UseP2PBackgrnd = 0, BiasType=1 (coverage bias regression was used), MergeInt=1.

**Micro-C arc plots and contact heat maps.** Plots of Micro-C contacts at individual loci were generated using the cLoops2 package[96] (https://github.com/YaqiangCao/cLoops2). First, Micro-C.pairs files were preprocessed using cLoops2 pre --format pairs. Reasonable contact matrix resolution was estimated using the cLoop2 estRes function. cLoops2 plot was used to plot contact density on specified genomic coordinates with --m obs, -arch was specified to plot arch plots and --triu was specified to plot binned triangular contact matrices. A bin size of 20 kb was used for regions larger than 1 Mb, otherwise a bin size of 500 bp was used.

**Circular genome tracks.** Circular tracks of Micro-C contacts, MNase, ChIP–seq and ATAC–seq were prepared using the HOMER software package[97] in combination with Circos[98]. First, duplicate-filtered ChIP–seq and ATAC–seq BAM files, and merged MNase BAM files were converted into HOMER tag directories using makeTagDirectory with the flag --keepAll. To prepare Micro-C samples, .pairs files were converted to the .hicsummary format by rearranging the columns as follows: egrep -v "(^#.*|^$)" filename.pairs | awk 'BEGIN {OFS = "\t"} {print($1, $2, $3, $6, $4, $5, $7)}' - > file.hicsummary. This was then converted a HOMER tag directory with makeTagDirectory with the flag --format HiCsummary. Tracks were produced with the analyzeHiC command and the following parameters: -res 5000 -superRes 10000 --circos cirOutput -nomatrix -minDist 20000 -pvalue 0.000000000000001. Track scales, line thickness and colours were edited in the cirOutput.config file and replotted using circos --conf.

**Micro-C nucleosome orientation density profiles.** To define nucleosome orientation from Micro-C ligation, a .pairs file was split into three files according to the orientation of read pairs using awk. First, intrachromosomal ligation events were isolated, then the reads were filtered to obtain junctions between 200 bp and 2 kb of one another. Inward (IN–IN) pairs were defined by matching read pairs with the read orientations +/− as follows: egrep -v "(^#.*|^$)" filename.pairs | awk 'BEGIN {OFS = "\t"} {if ($2 == $4 & $5-$3 > = 200 & $5-$3 < = 2000 & $6 == "+" & $7 == "−") print}' -. Outward (OUT–OUT) pairs were defined by matching +/− read orientations as follows: egrep -v "(^#.*|^$)" filename.pairs | 'BEGIN {OFS = "\t"} {if ($2 == $4 & $5-$3 > = 200 & $5-$3 < = 2000 & $6 == "−" & $7 == "+") print}' -. Tandem (IN–OUT or OUT–IN) pairs were identified by match pairs with +/+ or −/− orientations as follows: egrep -v "(^#.*|^$)" filename.pairs | awk 'BEGIN {OFS = "\t"} {if ($2 == $4 & $5-$3 > = 200 & $5-$3 < = 2000 & (($6 =="+" & $7 == "+") || ($6 == "−" & $7 == "−"))) print}' -. Tandem orientations are considered to be theoretically interchangeable and are not separated[30]. The distances between ligation junctions in each pair were then determined and were plotted using the ggplot2 function geom_density().

**Genome tracks visualizations.** Genome track screen shots were generated with genome-coverage-normalized (RPGC) data using Integrative Genomics Viewer[99].

**Protein structure visualization.** Mononucleosome structures were built using Protein Data Bank (PDB) 5NL0 (ref. 100), and visualized using the PyMOL Molecular Graphics System, v.3.0 Schrödinger.

The nucleosome arrays were modelled using PDBs 6IPU and 6HKT[36,101] and EMD-2601 (ref. 37) in the open-source 3D computer graphics software, Blender[102].

Structure prediction of MYC/MAX–TFAP2C complex was performed using AlphaFold-Multimer run in the COSMIC[2] portal using the amino acid sequences of the TF-DBDs only[103]. The resulting complex was then aligned with the crystal structure of human TFAP2A in complex with DNA (PDB: 8J0K)[104], using the PyMOL align function. The electrostatic surface charge was calculated using the APBS plugin in PyMOL[105].

## Reporting summary

Further information on research design is available in the Nature Portfolio Reporting Summary linked to this article.

## Data availability

All next-generation sequencing data generated as part of this study have been deposited in the Gene Expression Omnibus (GEO) under series accession number GSE201852. Previously published H3K27ac ChIP–seq, RNA-seq and ATAC–seq data were obtained from GSE98124, GSE171127 and GSE70234 (ref. 7). Histone H1 ChIP–seq data were obtained from GSE156697 (ref. 33) and GSE46134 (ref. 78). OCT4 ChIP–seq data in MEFs OCT4 48 h were obtained from GSE168142 (ref. 13). CTCF, PolII, P300 and H3K4me1/3 ChIP–seq data were from GSE29184 and GSE29218 (ref. 106). H3K9me1/me2 ChIP–seq data were from GSE54412 (ref. 107). RAD21 ChIP–seq data were from GSE111820 and GSE115984 (ref. 108). BRN2 ChIP–seq data were from GSE35496 (ref. 24). HP1α, SUV39H1/2 and H3K9me3 ChIP–seq data are from GSE57092 (ref. 109). OCT4 and SOX2 ChIP–seq data from secondary OSKM reprogramming system were obtained from GSE101905 (ref. 10). OSKM ChIP–seq in *Mbd3*[f/−] secondary reprogramming system were obtained from GSE102518 (ref. 11). All data were aligned to mouse reference genome MGSCv37 (mm9) (PRJNA20689). The nucleosome structure was from PDBs 6IPU and 6HKT[36,101], and the TFAP2A–DNA structure was from PDB 8J0K[104].

## Code availability

All of the scripts with nucleosome positions and motifs have been deposited at GitHub (https://git.ecdf.ed.ac.uk/soufi_lab/motif_mononucleosome). All of the scripts with nucleosome array positions and motifs have been deposited at GitHub (https://git.ecdf.ed.ac.uk/soufi_lab/motif_nucleosome_arrays).

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

**Acknowledgements** We thank I. Chambers and R. Illingworth for comments on the manuscript and for discussions, and K. Kaji for providing the secondary reprogrammable TNG-KOSM-MEFs and for discussions. A.S. is supported by an MRC career development award (MR/N024028/1) and CRUK-OHSU Project Award (C65925/A26986); M.R.O. by a PhD scholarship from the Darwin Trust of Edinburgh; K.F. and S.M.P. by a CRUK Program Grant (DRCNPG-Nov21\100002); A.A. by a PhD scholarship (1078107040) from King Abdulaziz City for Science and Technology; M.H. by EPSRC DTP PhD; S.S. by Carnegie PhD Scholarship (PHD012744); Y.B. by research grants from the EMBO Young Investigator Programme (YIP), Israel Science Foundation (ISF, 161/23), Leo Foundation (LF-OC-24-001624), Israel Cancer Research Funds (ICRF, 1276940) and by gift from N. G. Biasini; H.P. and A.I.S. by NIH grant R01GM147165; and D.V.F. by NIH grant R01HD114814. Edinburgh Genomics is partly supported through core grants from NERC (R8/H10/56), MRC (MR/K001744/1) and BBSRC (BB/J004243/1).

**Author contributions** A.S. and Y.B. conceived the TF combination comparison project and designed the ChIP–seq, RNA-seq and ATAC–seq experiments. A.S. and M.R.O. conceived the nucleosome organization project and designed the MNase–seq and Micro-C experiments. A.S. conceived the guided search model and signpost elements. Y.B. conceived the competition model between ESRRB and EOMES on TFAP2C binding. A.S. constructed the PiggyBac dual reporter and built the 3D nucleosome array models. M.R.O. performed TF ChIP–seq, MNase–seq and Micro-C experiments with assistance from E.H.-P. Micro-C, ATAC–seq, western blots and chromatin acid extraction in H1-KD and H1-OE MEFs were carried out by K.F. with assistance from S.M.P. Reprogramming to iPS cells in H1-KD MEFs was performed by A.A. All bioinformatics were performed by M.R.O. with assistance from A.S., J.A., K.F. and E.P. Reprograming MEFs and H1-OE MEFs to iTS cells and iPS cells was performed by H.B., M.R. and M.A., who also established stable iPS cell and iTS cell clones. M.A. targeted ES cells with the signpost system and performed and characterized the reprogramming of the signpost experiments. M.J. performed co-IP experiments, reprogrammed cells for 72h for ATAC–seq and RNA-seq and generated and characterized TMR iPS cells. K.M. injected the signpost-targeted ES cells, TMR and OSKM iPS cells to blastocysts and generated chimeras. H.P. performed HPLC–MS and D.V.F. performed LC–MS, both with assistance by A.I.S. EMSAs were performed by M.R.O., K.F., M.H. and S.S. The stemness of iPS cells and iTS cells was examined by H.Y. and A.L. using immunofluorescence and transgene silencing.

**Competing interests** The authors declare no competing interests.

**Additional information**
**Correspondence and requests for materials** should be addressed to Yosef Buganim or Abdenour Soufi.

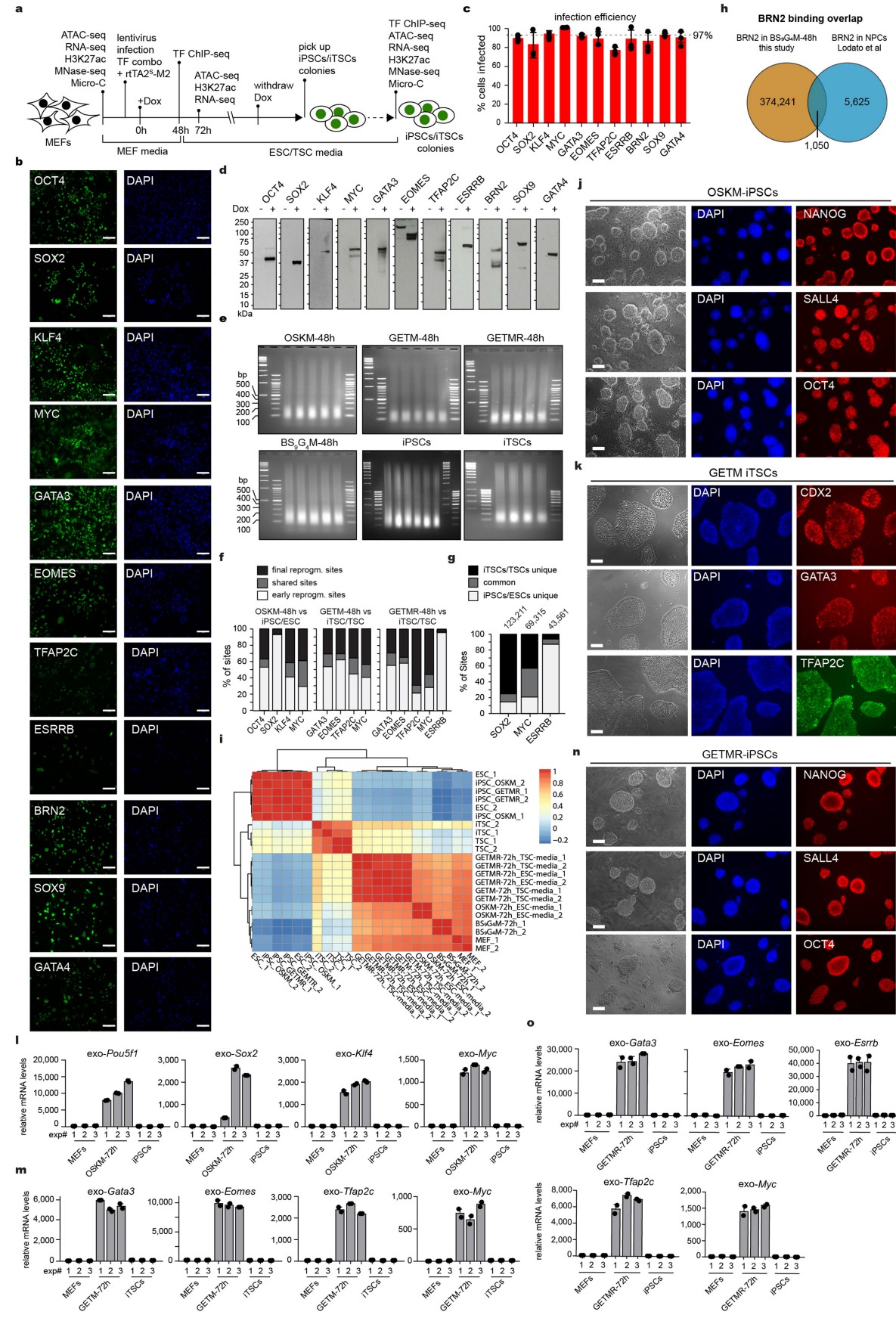

**Extended Data Fig. 1** | See next page for caption.

**Extended Data Fig. 1 | Pioneer TF off-targeting is a general feature in early reprogramming. a**, Experimental flowchart of reprogramming MEFs to iPSCs and iTSCs indicating timepoints of sample collection and experimental strategy carried out in this study. **b**, Immunofluorescence showing relatively homogenous ectopic expression of TFs in MEFs transduced with the corresponding lentivirus after doxycycline induction for 48 h. scale bar = 100 μm. **c**, Infection efficiency across the different TFs as measured by immunofluorescence shown in (b). Average biological replicates (n = 3) and error bars representing ± s.d. **d**, Western blot analysis showing the presence of the ectopic TFs running at the expected size in MEFs infected with the corresponding lentivirus only after doxycycline induction for 48 h. Raw blots are shown in Supplementary Fig. 1. **e**, Agarose gel electrophoresis showing equivalent chromatin fragmentation after sonication in all reprogramming contexts, which were used for ChIP-seq experiments. Each lane indicates an independent biological replicate. Unprocessed gels are shown in Supplementary Fig. 1. **f**, Overlap between TF binding sites at early and final stage reprogramming. Bars represent the percentage of the total number of sites identified between both conditions. **g**, Bar plots showing the extent of overlap between SOX2, MYC and ESRRB sites in iPSCs/ESCs and iTSCs/TSCs, indicating their cell-type-specific binding. Bars represent the percentage of the total number of sites identified in both conditions. **h**, Venn diagram showing the overlap between the binding of BRN2 in early reprograming (this study) and in NPCs. **i**, Pearson correlation heatmap of the top 500 most variable genes across all early and final reprogramming contexts as measured by RNA-seq. Correlation colour scale is indicated. **j,k,n**, Immunofluorescence of pluripotency (NANOG, SALL4, and OCT4) and trophoblast stem cell markers (CDX2, GATA3, and TFAP2C) in the fully reprogrammed iPSCs and iTSCs, respectively. The corresponding DAPI staining (blue), and brightfield images are also shown. Scale bar = 100 μm. **l,m,o**, Bar plots showing the silencing of exogenous reprogramming genes (indicated above) after the completion of reprogramming. Three biological replicates (exp.) of MEFs, 72 h after TF induction, and iPSC or iTSC clonal lines were used. Gene expression measured by qPCR and the mean values of technical replicates (n = 2) normalized against *Gapdh*. Error bars representing ± s.d.

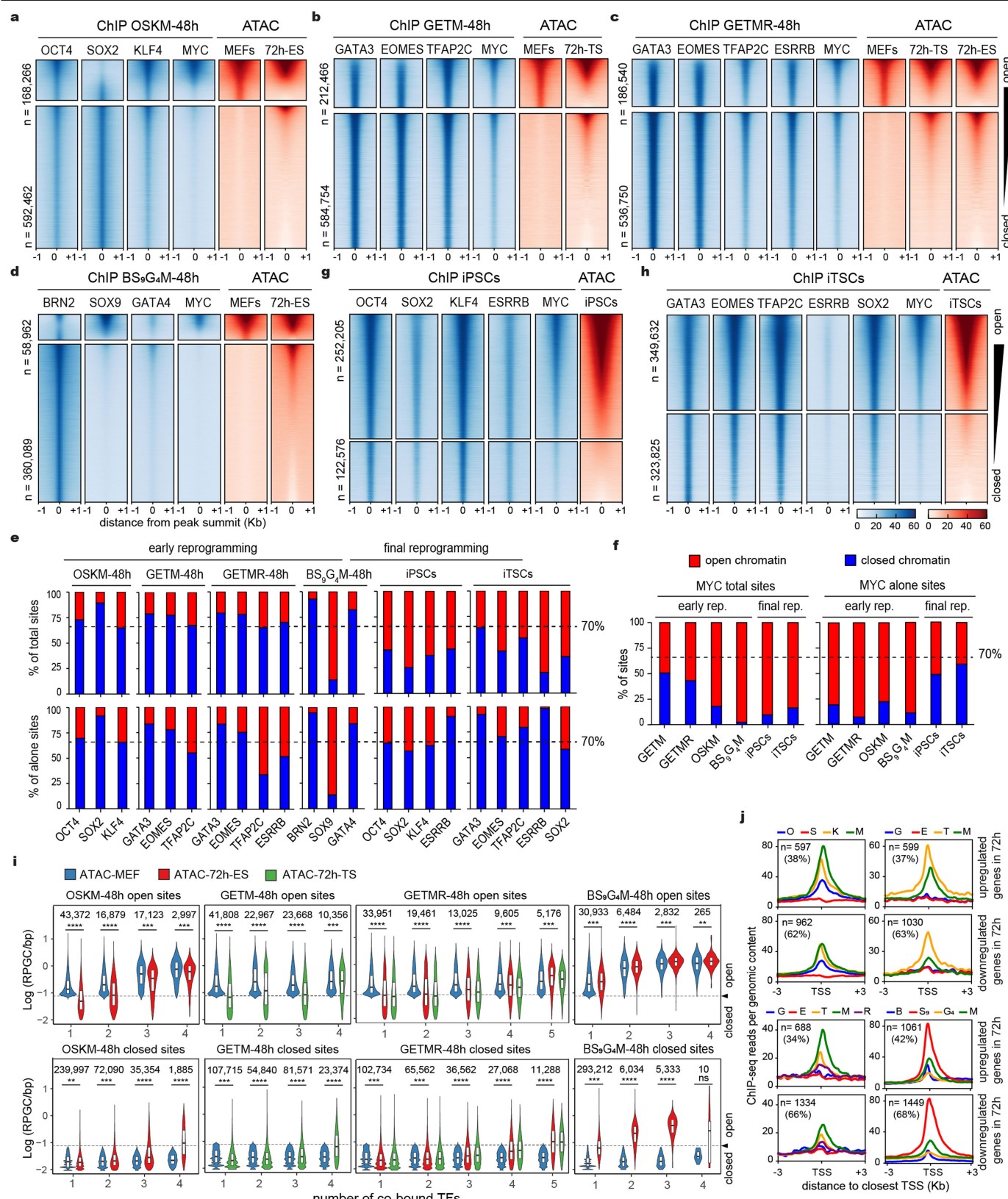

**Extended Data Fig. 2** | See next page for caption.

**Extended Data Fig. 2 | Pioneer TFs target closed chromatin individually and together during early reprogramming. a**, Read density heatmaps of O,S,K,M ChIP-seq signal (blue) in OSKM-48h spanning ±1 kb around the summits of O,S,K,M peaks pooled together. Heatmaps of ATAC-seq signal (red) showing changes of chromatin accessibility around TF binding sites from MEFs to OSKM-72h. Open and closed sites separated according ATAC-seq in MEF and rank ordered by ATAC-seq in OSKM-72h. The number of sites (n) is indicated. **b,c,d,g,h**, As in (a) but for G,E,T,M in GETM-48h, G,E,T,M,R in GETMR-48h, B,S$_9$,G$_4$,M in BS$_9$G$_4$M-48h, O,S,K,R,M in iPSCs/ESCs and G,E,T,R,S,M in iTSCs/TSCs, respectively. **e**, Bar plots showing the percentage of TF binding to closed sites (blue) versus open sites (red) in early and final reprogramming. Total sites are shown on top and unique sites where each TF is bound individually are shown at the bottom. **f**, Same as in (e) for MYC sites. **i**, Violin plots of chromatin accessibility changes in early reprogramming as a function of the number of TFs co-bound within open chromatin (top) and closed chromatin (bottom) for each TF combination. Open and closed chromatin threshold is indicated by dotted line. The violin shapes indicate the maxima, minima and data distribution. The bottom, top, and middle line of boxes indicate the first quartile (25th percentile), third quartile (75th percentile), and median (50th percentile), respectively. Statistical significance measured by paired t-test and P values are indicated by (****) for p <= 0.0001, (***) p <= 0.001, (**) p <= 0.01, and (ns) for p > 0.05. **j**, Profile plots of TF enrichment around ±3 kb from TSS of upregulated (top panels) and downregulated genes (bottom panels) in early reprogramming (72 h). TF enrichment profiles are colour coded in each reprogramming system as shown above. The number (n) and percentage of up and downregulated genes are indicated.

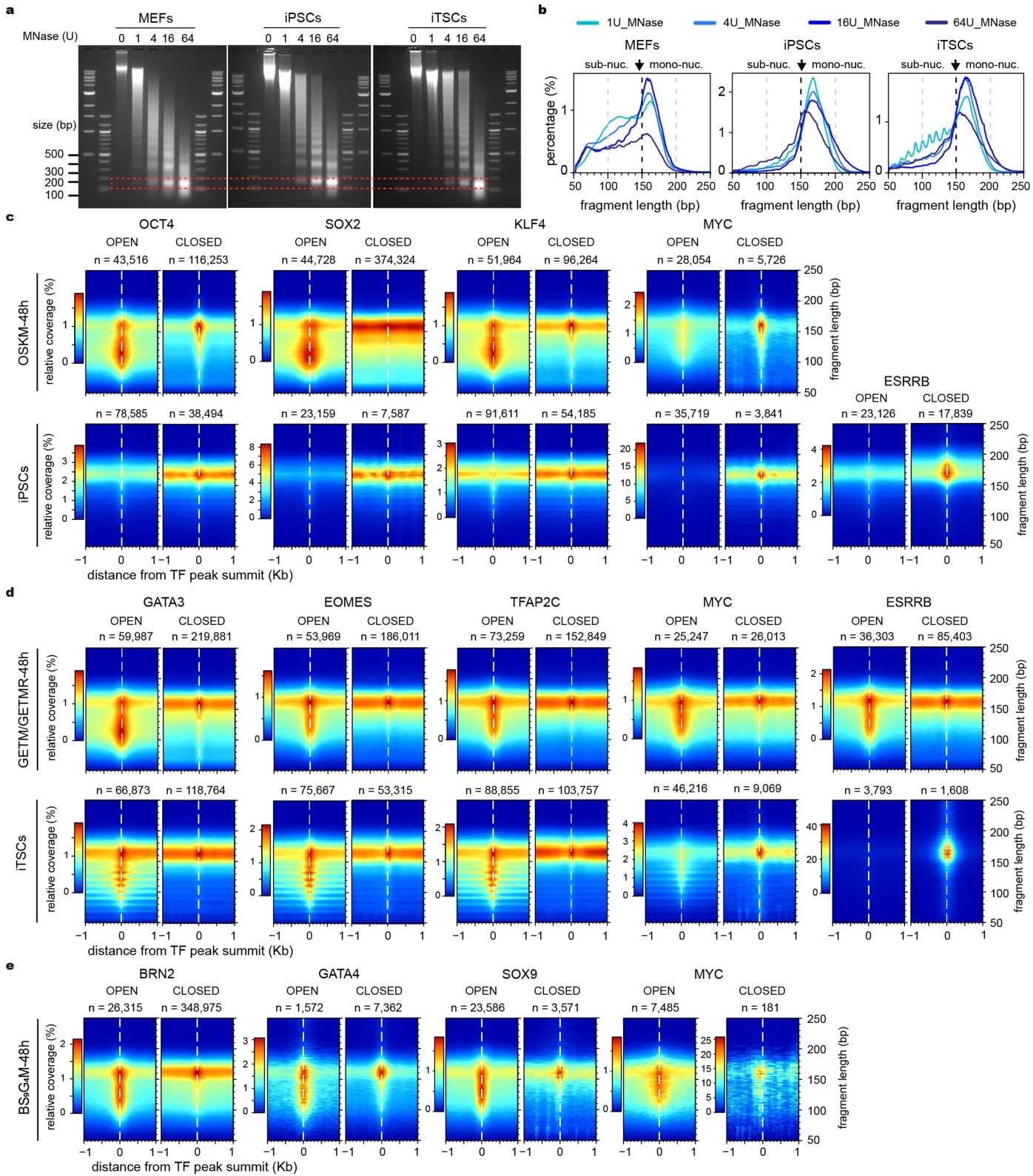

**Extended Data Fig. 3 |** See next page for caption.

**Extended Data Fig. 3 | TFs bind fragile or sub-nucleosomes in open chromatin and intact nucleosomes in closed chromatin. a**, Agarose gel electrophoresis showing gradual chromatin digestion with increasing amounts of MNase (1-64U). DNA fragments ranging from -90–200 bp (dotted red box) were used in MNase-seq. Representative images from at least n = 3 biological replicates, which were pooled together for sequencing. Unprocessed gels are shown in Supplementary Fig. 1. **b**, profile plots showing fragment size distributions obtained from MNase-seq in iPSCs, iTSCs, and MEFs using 1, 4, 16, and 64 U/mL MNase. Arrow on 150 bp indicates the separation between regions containing mainly sub-nucleosomes (< 150 bp) from those containing mainly canonical mono-nucleosomes (>150 bp). MNase amounts are colour coded as indicated on top. **c-e**, MNase-seq 2D heatmaps showing nucleosome enrichment against DNA fragment size around TF peak summits (±1 kb) within open (left) versus closed chromatin (right). The 2D heatmaps were generated using 1U MNase digestion to show fragile and sub-nucleosome species, which are not detected at higher MNase concentrations. Fragment sizes around -150–170 bp represent canonical nucleosome, and fragments <150 bp represent fragile or sub-nucleosomes. Heatmaps are auto-scaled according to closed chromatin signal on each set as indicated on the left. In (d) a 10 bp footprint pattern within open chromatin in iTSCs indicate well positioned fragile at low MNase concentrations.

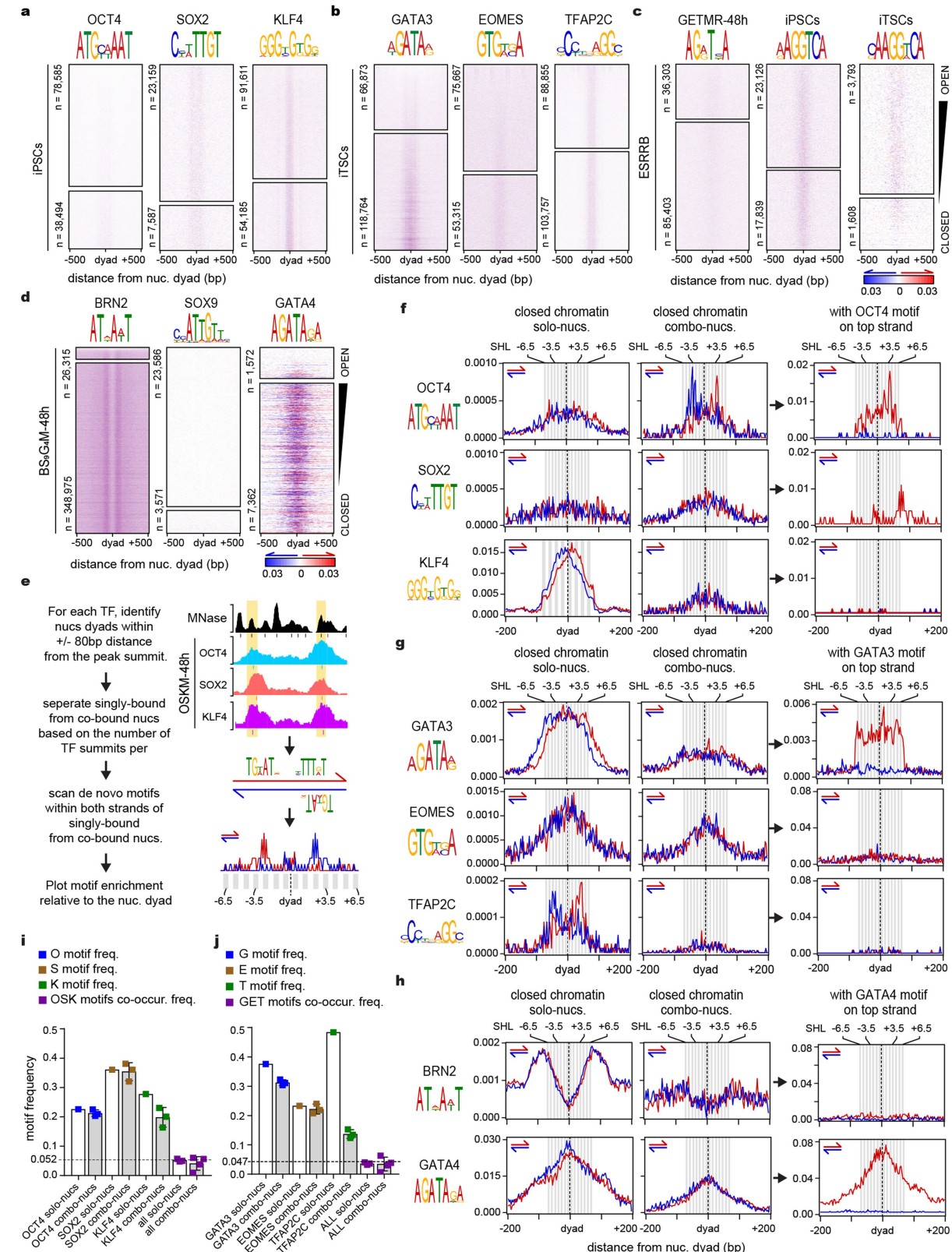

**Extended Data Fig. 4** | See next page for caption.

**Extended Data Fig. 4 | TFs display distinct motif readout on mono-nucleosomes when bound individually and together. a**, Motif density heatmaps showing the distribution of de novo motifs (logos on top) around nucleosome dyads (±500 bp) targeted by OSK in fully reprogrammed cells within open (top) and closed (bottom) chromatin. Motifs are scored on both DNA strands (blue and red) following the colour gradient scale at the bottom. The number (n) of nucleosomes targeted by each TF are indicated. **b**–**d**, Same as in (a) for GET in final reprogramming, ESRRB in early and fully reprogrammed cells, and $BS_9G_4$ in early reprogramming, respectively. **e**, Flowchart of assigning nucleosomes to TF solo-binding and combo-binding followed by motif scanning around the dyads. **f**, Line plots showing motif scores on the top (red) and bottom (blue) DNA strands around nucleosome dyads (±200 bp) targeted by OSK when bound individually (solo-nucs, left panels) or in combination (combo-nucs, middle panels) in fully reprogrammed cells. Nucleosomes bound by OSK together and contain OCT4 motif on the top strand (right panels) within closed chromatin. **g,h**, Same as in (f) for GET in iTSCs and Brn2 and Gata4 in $BS_9G_4M$-48h cells, respectively. **i**, Bar plots showing the frequency of OSK motif occurrence alone and together (co-occur. freq.) in solo-nuc. and combo-nucs. Average motif frequencies with error bars representing ± s.d. Average motif co-occurrence frequency by chance is indicated by dotted line. **j**, Same as (i) for GET motifs.

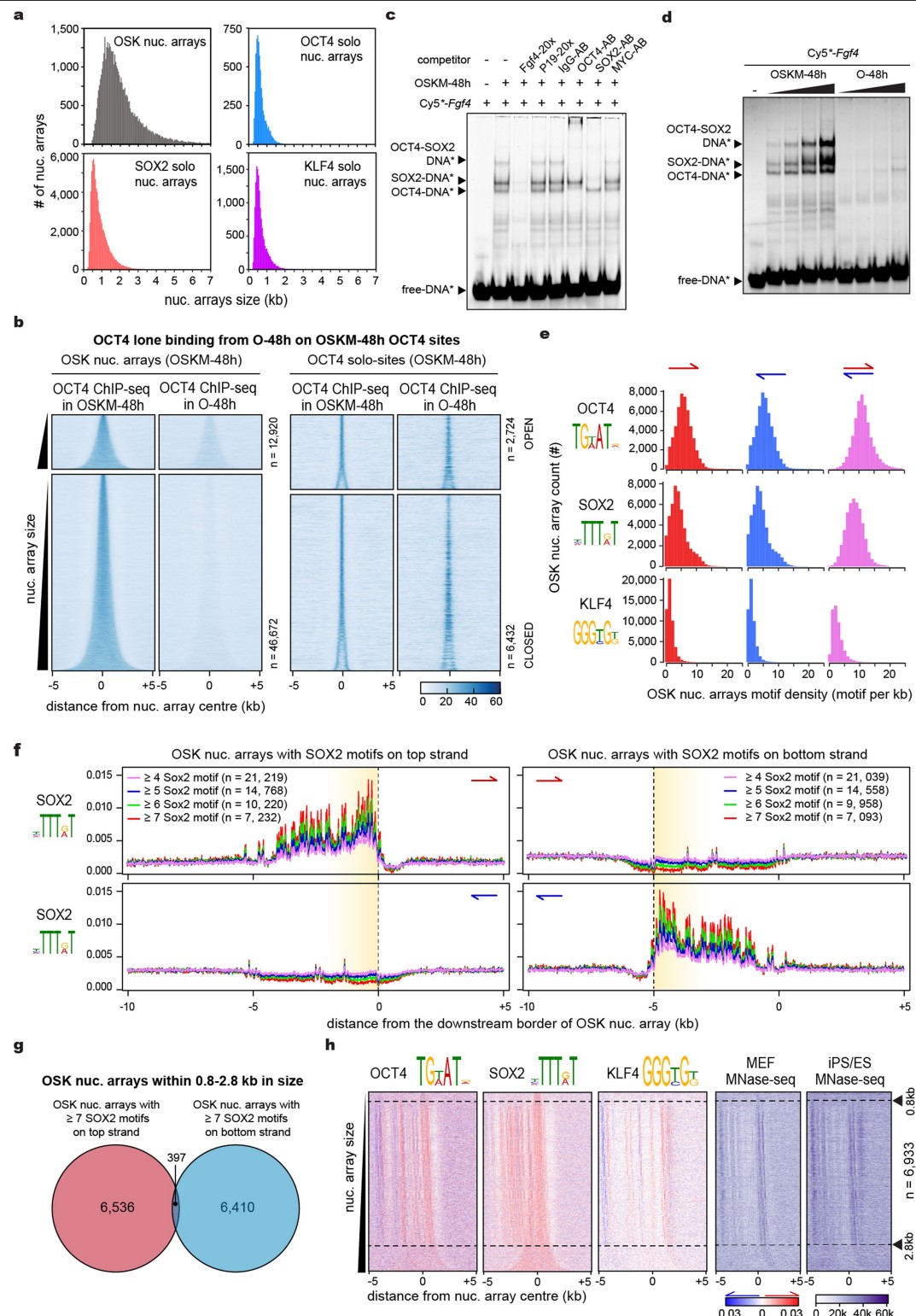

**Extended Data Fig. 5** | See next page for caption.

**Extended Data Fig. 5 | OSK motif readout on nucleosome arrays decipher combinatorial binding in early reprogramming. a**, Histograms showing size distribution of OSK nucleosome arrays (grey) compared to arrays targeted by O,S,K individually (blue, red, magenta, respectively). **b**, Read density heatmaps showing OCT4 ChIP-seq around OSK nucleosome arrays (left) or OCT4 lone-bound sites (right) in OSKM-48h and O-48h, spanning ±5 kb around the array midpoints. The nucleosome arrays were ranked ordered based on size and grouped into open (top panels) and closed (bottom panels) according to ATAC-seq in MEFs (not shown). The number of nucleosome arrays (n) are indicated. **c**, OCT4 and SOX2 bind specifically together to *Fgf4* enhancer containing OCT/SOX composite motif. Super-shift EMSA showing three retarded bands when OSKM-48h nuclear lysates were incubated with Cy5-labelled oligonucleotide from *Fgf4* enhancer. The bands correspond to OCT4-DNA, SOX2-DNA and OCT4/SOX2-DNA complexes. All three bands were diminished when excessive amounts of the specific (*Fgf4*) but not the non-specific competitor (*P19*) were added. Specific bands were also diminished when the corresponding antibodies were added. Representative image from (n = 3) biological replicates. Uncropped gels are shown in Supplementary Fig. 1. **d**, EMSA showing OCT4/SOX2-DNA complexes were formed only when OSKM-48h nuclear lysates were incubated with *Fgf4* enhancer but not O-48h lysates. Representative image from (n = 2) biological replicates. Uncropped gels are shown in Supplementary Fig. 1. **e**, Histograms showing OSK motif frequency distribution within OSK nucleosome arrays. **f**, Profile plots showing SOX2 motif distribution around OSK nucleosome array borders, ranging in motif density from ≥ 4-to-7 motif/kb. Motif enrichment is measured in both strands separately, showing motif unidirectional orientation. The number (n) of nucleosome arrays with the different SOX2 motif densities are indicated. **g**, Venn diagram showing the overlap between OSK nucleosome arrays containing SOX2 motifs on the top and bottom strands. Only arrays with motif density ≥ 7 motif/kb and 0.8-2.8 kb in size are shown here. **h**, Heatmaps showing the OSK motif distribution patterns (logos on top) around nucleosome array midpoint (±5 kb) bound by OSK in early reprogramming within closed chromatin and contain ≥ 7 SOX2 motif/kb on the top strand. Motifs are scored on the top (red) and bottom (blue) strands as indicated by the colour gradient scale at the bottom. MNase heatmaps (purple) within the same OSK nucleosome arrays are shown on the left. The arrays were rank ordered based on size and those within 0.8-2.8 kb are indicated by arrowheads and dashed lines. The number (n) of the OSK nucleosome arrays is indicated.

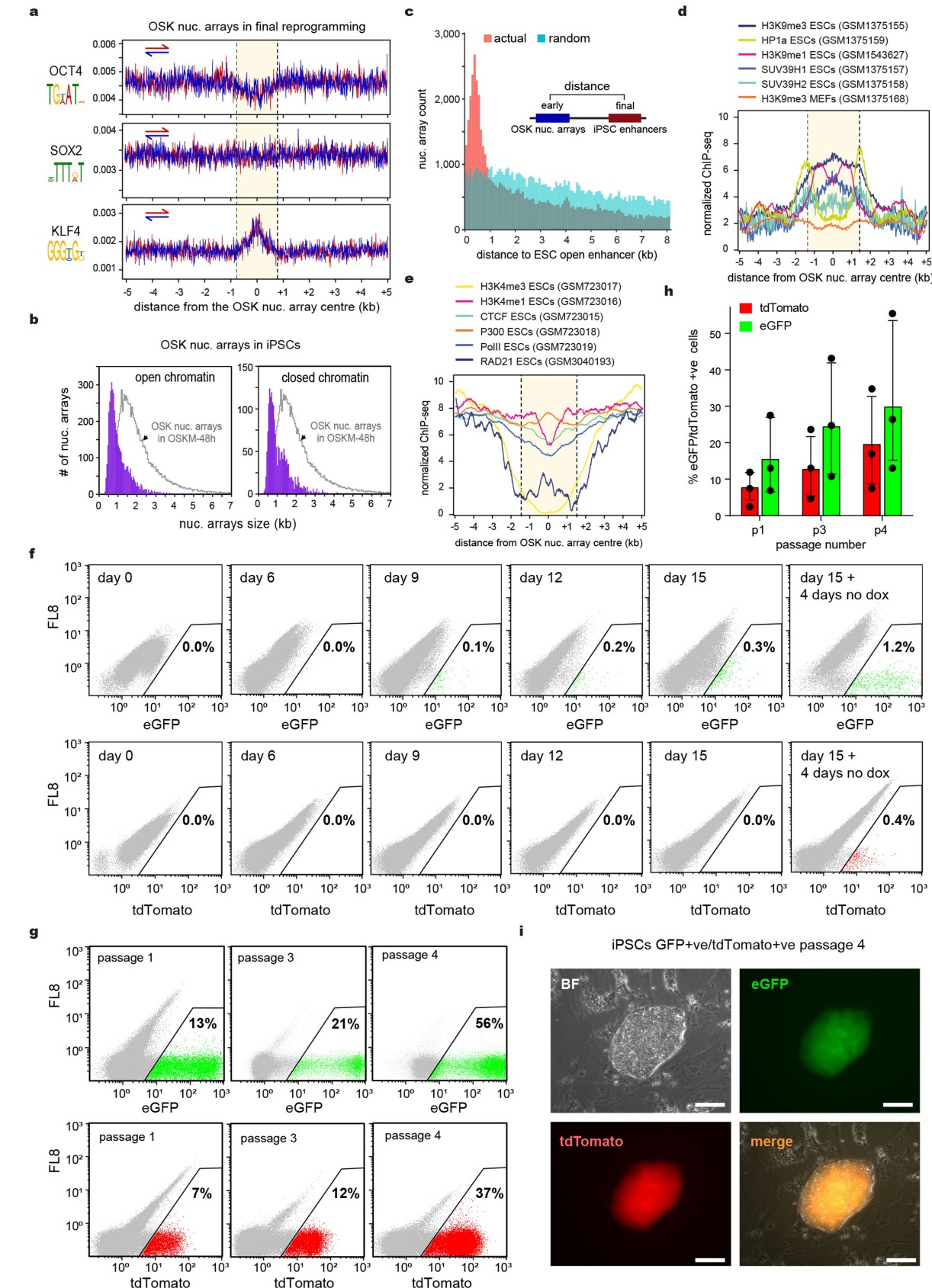

**Extended Data Fig. 6** | See next page for caption.

**Extended Data Fig. 6 | Combinatorial binding to signpost elements guides OSK to pluripotency enhancers during reprogramming. a**, Line plots showing O,S,K motifs enrichment on the top (red) and bottom (blue) DNA strands around the midpoints of OSK nucleosome arrays (±5 kb) in fully reprogrammed iPS cells. The average array size highlighted in yellow with dotted lines at the borders. **b**, Histograms showing size distribution of OSK nucleosome arrays bound in final reprogramming (purple) compared to early reprogramming (grey). **c**, The early reprogramming OSK nucleosome arrays are adjacent to enhancers in iPS cells. Bar plot showing the distance between OSK nucleosome arrays in early reprogramming and enhancers in iPSCs (represented by a schematic in the inset). The experimental distance (actual) was compared to random sequences. Two-sided Wilcoxon rank-sum test with continuity correction: $w = 10,696,640,768$ and $P = 2.286 \times 10^{-6}$. **d**, OSK nucleosome arrays in early reprogramming are silenced in iPS cells. Line plots showing that histone marks and co-factors associated with heterochromatin are enriched within the early OSK nucleosome arrays (highlighted in yellow) after the completion of reprogramming. The GEO access codes of the data used are indicated. **e**, Same as (d) for histone marks associated with active chromatin. **f**, The expression of eGFP and tdTomato were measured by flow cytometry during reprogramming, showing motif directionality in *Nanog* signpost element is important for gene reactivation. **g**, eGFP and tdTomato expression gradually increases in iPSCs after passaging. The expression of eGFP and tdTomato were measured by flow cytometry in iPSCs after different passages. **h**, Bar plot quantifying eGFP+ve and tdTomato+ve cells during reprogramming as measured in (g). Data are mean ± s.d. from biological replicates (n = 3). **i**, Fluorescence images of iPSC colony after passage four. Bright field (BF) and merged images are also shown. Representative image from n = 3 biological replicates. Scale bar = 100 µm.

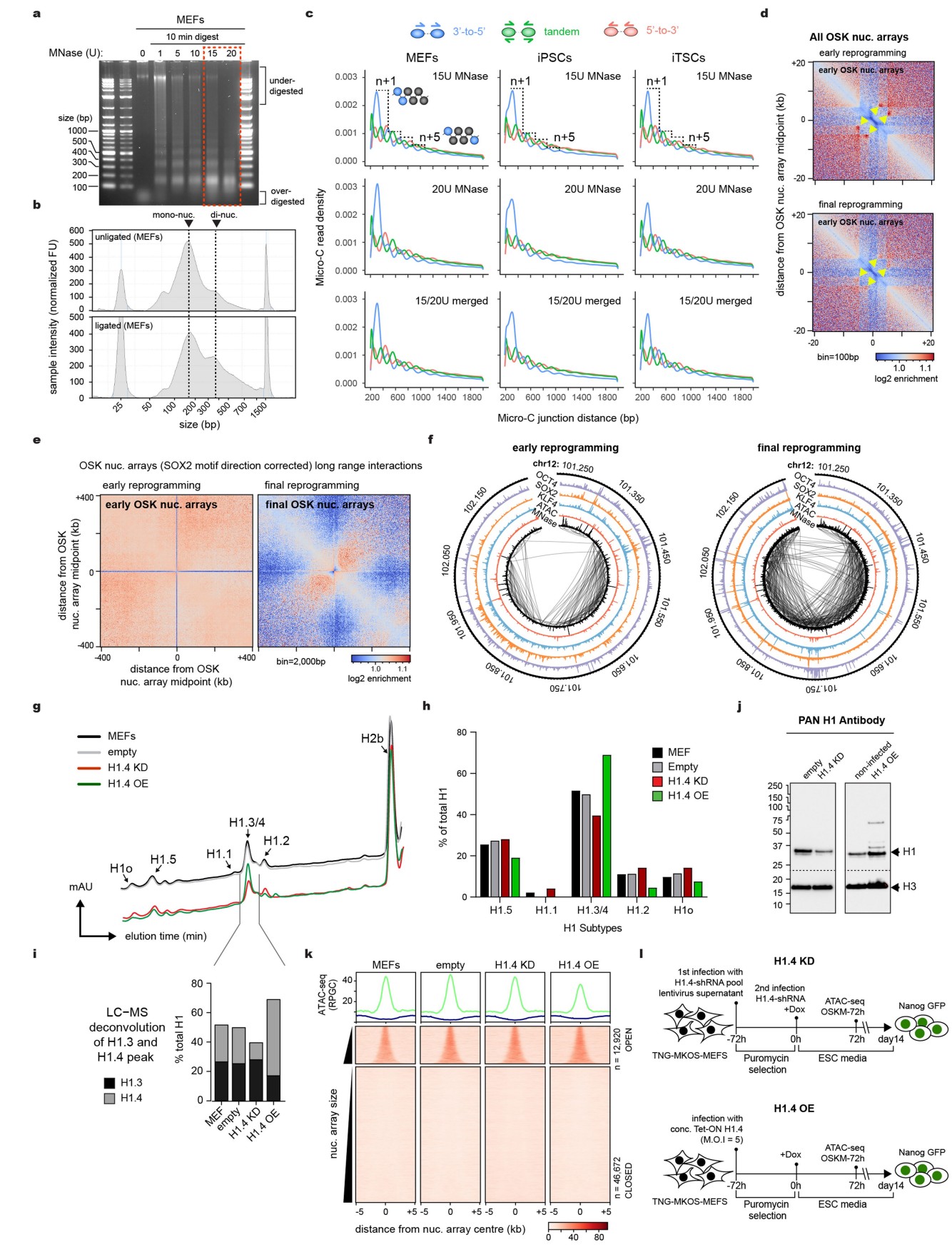

**Extended Data Fig. 7** | See next page for caption.

**Extended Data Fig. 7 | Micro-C reveals distinct spatial organization of nucleosome arrays targeted by OSK. a**, Agarose gel electrophoresis showing gradual chromatin digestion with increasing amounts of MNase concentrations. 15 and 20 U of MNase (dotted red box) were used for Micro-C experiments. Representative image from n = 4 biological replicates, which were pooled together for sequencing. **b**, Profile plot showing mono- and di-nucleosome DNA fragment sizes before (top) and after (bottom) proximity ligation. **c**, Decaying curves of inter-nucleosomal Micro-C contacts zoomed within 200 bp and 2 kb distance. Micro-C contact density normalized by sequencing depth. Three curves showing distinct read pair orientations relative to one another are colour coded as shown in the schematics above. Contacts of up to six nucleosomes can be resolved (dashed line). Schematics illustrating the inter-nucleosomal contacts between n/n + x in different orientations are indicated on top. Insets show an example of n/n + 1 and n/n + 5 (painted blue) in 3′-to-5′ orientation. **d**, Micro-C pileup heatmaps of OSK nucleosome arrays in early reprogramming (top) and fully reprogrammed cells (bottom). Maps are plotted at 100 bp resolution for fine-scale inter-nucleosome contacts and centred around the upstream near-border. Yellow arrowheads indicate strong interactions between the near and far-border, which disintegrate in final reprogramming. **e**, OSK bind to more interactive enhancers after reprogramming. Pileup Micro-C analysis showing long-range interactions at 2 kb resolution around nucleosome arrays midpoints targeted by OSK in early (left) and final reprogramming (right). **f**, Circos plots showing long range interactions linking OSK binding (ChIP-seq) along with chromatin accessibility (ATAC-seq) and nucleosome positions (MNase-seq). More long-range interactions are observed after reprogramming. **g**, Reverse-phase HPLC analysis of chromatin histone extracts purified from MEFs (black), MEF-empty (grey), MEF-H1.4KD (red), and MEF-H1.4OE (green), showing the abundance of H1 variants. Absorbance at 214 nm in milli-absorbance-units (mAU) was plotted as function of elution time (min). **h**, Bar plots of relative H1 amounts quantified by HPLC in (g) showing the compensation effects of H1 variant expression after H1.4KD and H1,4OE in MEFs. **i**, LC-MS successfully deconvoluted the H1.3 (black) and H1.4 (grey) amounts in MEFs, MEF-empty, MEF-H1.4KD, and MEF-H1.4OE, which were not resolved by HPLC in (g). **j**, Western blot analysis showing the amounts of total H1 (using pan-H1 antibody) in MEF-H1.4KD and MEF-H1.4OE compared to MEF-empty. Protein ladder sizes in KDa are indicated. representative image from n = 5 biological replicates. Raw blots are shown in Supplementary Fig. 1. **k**, Average profile plots (top) and read density heatmaps (bottom) of ATAC-seq signal around nucleosome arrays bound by OSK in MEFs, MEF-empty, MEF-H1.4KD and MEF-H1.4OE. The nucleosome arrays were ranked ordered based on size and grouped into open and closed according to ATAC-seq in MEFs. The number of nucleosome arrays (n) are indicated. Four biological replicates (n = 4) were sequenced and merged for analysis. **l**, Experimental flow chart of ATAC-seq to measure the effects of H1.4KD (top) and H1.4OE (bottom) on chromatin accessibility as a proxy for OSKM binding during reprogramming early of TNG-KOSM-MEFs. Fully reprogrammed KOSM-MEFS were assessed by the expression of GFP, which has been knocked-in to one of the *Nanog* alleles (TNG).

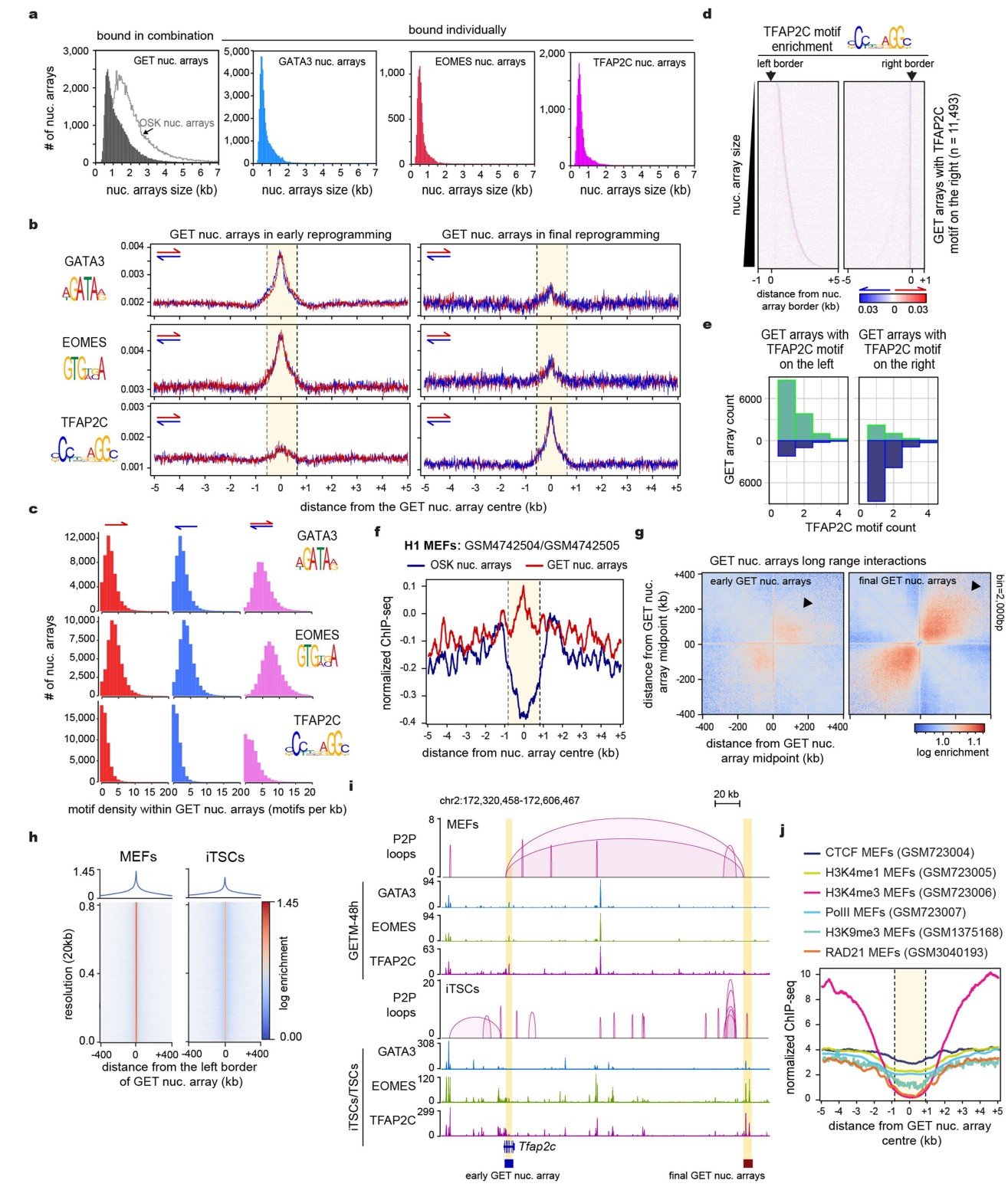

**Extended Data Fig. 8** | See next page for caption.

**Extended Data Fig. 8 | GET recognize motif-dense and highly inter-connected nucleosome arrays enriched for H1. a**, Histograms showing size distribution of GET nucleosome arrays (grey bars) compared to arrays targeted by GET individually (blue, red, magenta, respectively). OSK nucleosome arrays are also shown for comparison (grey line). **b**, Profile plots showing GET motif distribution on top (red) and bottom (blue) DNA strands around the centre of GET nucleosome arrays (highlighted in yellow), with borders indicated in dotted lines. **c**, Histograms showing GET motif frequency distribution within GET nucleosome arrays in early reprogramming. **d**, Motif density heatmaps on both DNA strands (red and blue) around the left border (left panel) and right border (right panel) of GET nucleosome arrays containing TFAP2C motif on the right border. The arrays were rank-ordered based on size and motifs scored by the colour gradient scale at the bottom. Number of arrays (n) is indicated on the side. **e**, TFAP2C motifs are enriched in either the left or right border of the GET nucleosome arrays. Bar plot showing the count of TFAP2C motifs within each border of GET nucleosome arrays. **f**, profile plot showing the enrichment of H1 within GET nucleosome arrays in early reprogramming (red) in contrast to OSK nucleosome arrays (blue). Average GET nucleosome array size highlighted in yellow. GEO access codes of H1 ChIP-seq is indicated. **g**, Micro-C pile-up heatmaps of nucleosome arrays targeted by GET in early reprogramming (left) and in iTSCs (right) showing the increase of long-range interactions (indicated by arrow) after the completion of reprogramming. Bins = 2,000 bp and log enrichment scale is indicated at the bottom. **h**, Corner stripe stackup profiles (top) and heatmaps (bottom) showing the diffusion of borders around GET nucleosome arrays from early (left panels) to full reprogramming (right panels) at 2 kb resolution. Only GET nucleosome arrays with TFAP2C motif on the left border are shown (n = 11,501). **i**, GET translocate across interconnected nucleosome arrays during reprogramming. Arch representations of peak to peak (P2P) loops (magenta) showing interactions of exemplar nucleosome array bound by GET in early (blue) and final reprogramming (red). Genome browser tracks of ChIP-seq corresponding to GET nucleosome arrays highlighted in yellow. P2P loops are called by FitHiChIP with Q<0.01 threshold. **j**, GET nucleosome arrays in early reprogramming are not enriched for Cohesin and CTCF in MEFs. Line plots showing that histone marks and co-factors associated with open chromatin are depleted within GET nucleosome arrays in MEFs. The GEO access codes of the data used are indicated.

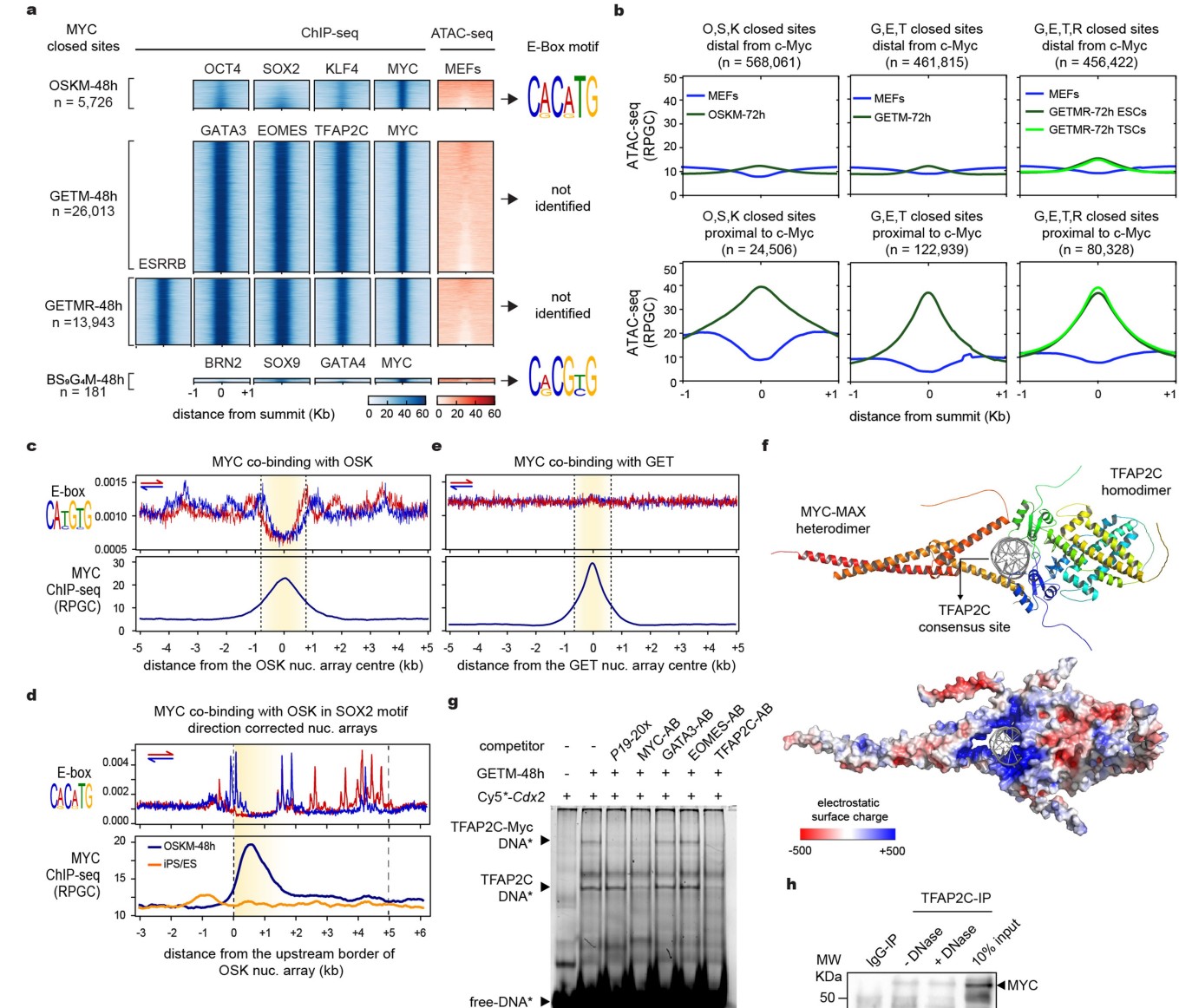

**Extended Data Fig. 9 | Non-pioneer MYC binding with OSK and GET to nucleosome arrays follow distinct mechanism. a**, MYC binding to closed chromatin is markedly different in the four TF combinations. Read density heatmaps showing TF ChIP-seq signal (blue) and ATAC-seq signal (red) spanning ±1 kb of MYC peak summits within closed chromatin in the indicated early reprogramming condition. The numbers (n) of MYC bound sites in each condition are indicated. Colour scale bars (RPGC) are indicated below. E-box motifs identified by de novo motif analysis from each group are indicated. **b**, Closed chromatin sites proximal to MYC display more opening in early reprogramming. Average chromatin accessibility (ATAC-seq) around ±1 kb of OSK, GET and GETR sites distal versus proximal to MYC sites before and after 72 h ectopic TF expression. The number of sites (n) is indicated. **c**, Profile plots (top panel) showing E-box motif enrichment on the top (red) and bottom (blue) DNA strands around the near-border (dotted line) of nucleosome arrays bound by OSK and MYC during early reprogramming. MYC ChIP-seq enrichment in early reprogramming is shown in the bottom panel. The average array size highlighted in yellow. **d**, Same as (c) for OSKM nucleosome arrays corrected for SOX2 motif orientation. **e**, Same as (c) for GET binding with MYC. **f**, Cartoon (top panel) and electrostatic surface (bottom panel) representations showing the interaction of MYC/MAX heterodimer with TFAP2C homodimer.

The protein surface is coloured according to its electrostatic potential from red (−500 kT, negatively charged) to blue (+500 kT, positively charged). The complex structure was predicted by Alpha-Multimer only considering DBDs of MYC/MAX and TFAP2C. Cartoon representation of DNA (grey) containing TFAP2C site is shown. **g**, TFAP2C and MYC bind specifically together to *Cdx2* enhancer site (TFAP2C target). Super-shift EMSA showing two retarded bands when GETM-48h nuclear lysates were incubated with Cy5-labelled oligonucleotide from *Cdx2* enhancer containing TFAP2C site. The two bands correspond to TFAP2C-DNA and MYC/TFAP2C-DNA complexes. The MYC/TFAP2C-DNA band diminished after adding excessive amount of *P19* (*Cdkn2d* promoter) oligonucleotide (MYC target) or MYC antibody as competitors. Both bands diminished after adding TFAP2C antibody, unlike GATA3 and EOMES antibodies. Representative image from n = 2 biological replicates. Uncropped gels are shown in Supplementary Fig. 1. **h**, Co-immunoprecipitation of TFAP2C and MYC indicating direct protein-protein interaction. Immunoprecipitation of TFAP2C, but not IgG, allows the detection of MYC by western blot in the presence of absence of DNase. Band representing MYC is indicated by an arrowhead. Molecular weight marker (KDa) is indicated. Representative image from n = 2 biological replicates. Raw blots are shown in Supplementary Fig. 1.

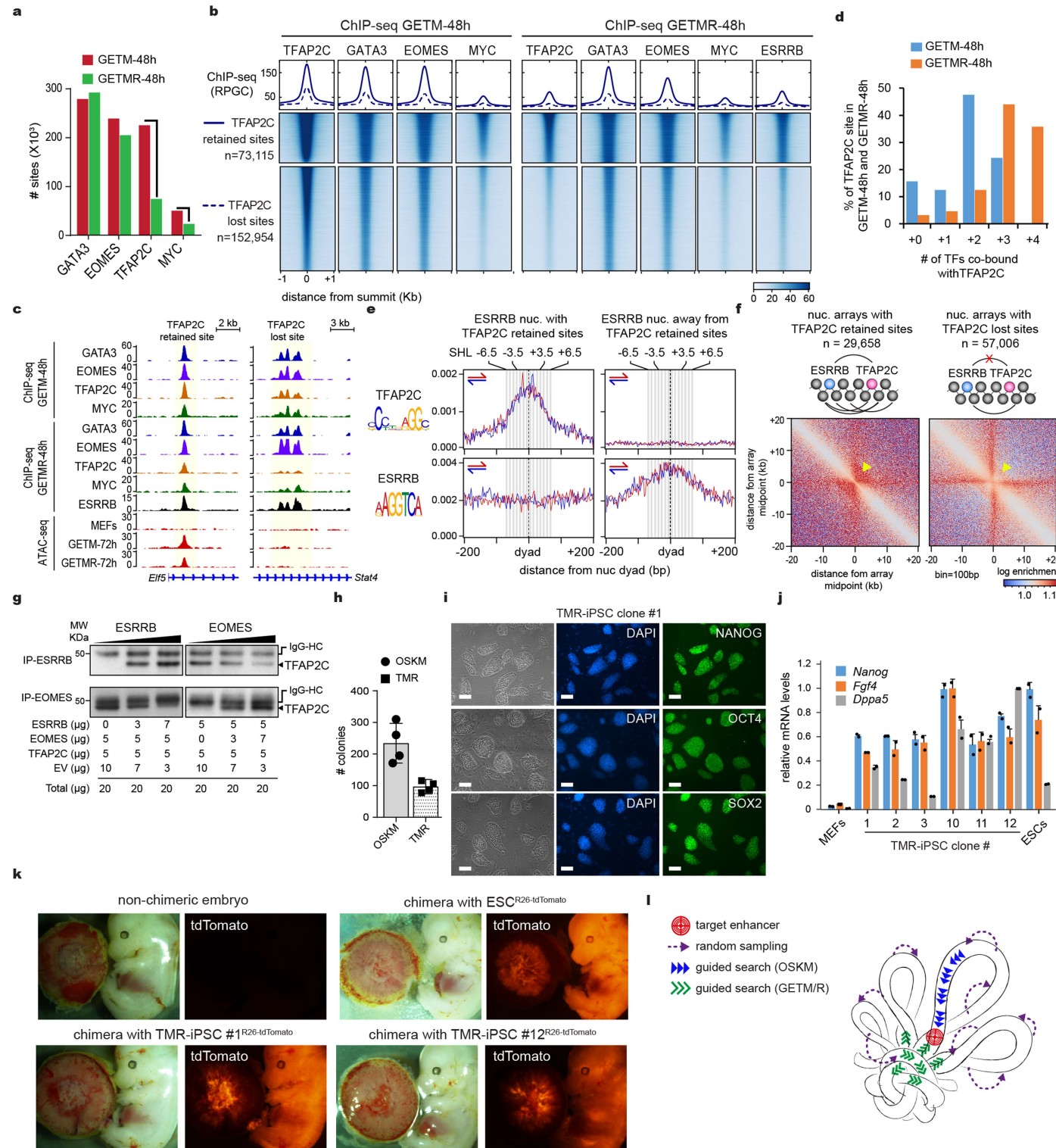

**Extended Data Fig. 10** | See next page for caption.

**Extended Data Fig. 10 | Competitive interaction between EOMES and ESRRB for TFAP2C/MYC on nucleosome arrays expands GETMR reprogramming.** **a**, Bar plot of GETM peaks identified by ChIP-seq in GETM-48h and GETMR-48h cells showing a significant loss of TFAP2C and MYC sites in the presence of ESRRB. **b**, ChIP-seq read density heatmaps (blue) of GETM in GETM-48h cells and GETMR in GETMR-48h cells spanning ±1 kb from TFAP2C summits of retained (top) versus lost sites (bottom) when comparing GETM-48h to GETMR-48h cells. Sites are sorted based on the central enrichment of TFAP2C in GETM-48h cells. The average enrichment of the corresponding TFs in retained (solid line) versus lost (dotted line) TFAP2C sites are shown above. The number (n) of sites are indicated in the left. Colour scale indicated indicate normalized ChIP-seq enrichment (RPGC). **c**, Genome browser tracks of representative loci containing a TFAP2C retained or lost site, showing GETM and GETMR enrichment (ChIP-seq) in GETM-48h and GETMR-48h cells, respectively, and chromatin accessibility (ATAC-seq) in MEFs, GETM-72h and GETMR-72h cells. **d**, Bar plots showing the percentage of TFAP2C sites bound individually or co-bound with other TFs. **e**, Profile plots of TFAP2C and ESRRB motif distribution around the dyad of nucleosomes bound by ESRRB overlapping with or away from TFAP2C retained sites (left and right panels, respectively) in GETMR-48h cells. **f**, Micro-C pile-up heatmaps around nucleosome arrays containing retained (left) or lost (right) TFAP2C sites after adding ESRRB to GETM during early reprogramming. Maps plotted at 100 bp resolution. Yellow arrowheads indicate local cross-interactions. Schematic on top showing the co-binding of ESRRB and TFAP2C mediated by inter-nucleosome interactions. **g**, Immunoprecipitation of EOMES or ESRRB and western blot for TFAP2C in the presence of constant EOMES and increasing ESRRB (left) or constant ESRRB

and increasing EOMES (right). The amount of each transfected plasmid encoding the corresponding TF is shown below. The bands representing TFAP2C, and antibody heavy chain (IgG-HC) are indicated. Representative image from n = 2 biological replicates. Molecular weight marker (KDa) is indicated. Raw blots are shown in Supplementary Fig. 1. **h**, Bar plot showing the reprogramming efficiency of MEFs to iPS cells using OSKM and TMR. Mean values of biological replicates (n = 4) with error bars representing ± s.d. **i**, Immunofluorescence of the indicated pluripotency markers (green fluorescence) in stable TMR iPS clonal lines. Bright field images showing typical morphology of the corresponding TMR-iPS cells. Nuclear DAPI staining (Blue) images are also shown. Representative image from n = 6 independent iPS clones shown in (j). Scale-bar = 100 μm. **j**, Gene expression of pluripotency markers by q-PCR from six independent TMR-iPS clones (n = 6) as compared to MEFs (negative control) and ESCs (positive control). Mean values of technical replicates (n = 2) with error bars representing ± s.d. **k**, Two stable TMR-iPS clonal lines carrying a tdTomato reporter in Rosa26 locus were used for chimera assays. Both lines display significant chimeric contribution as measured by tdTomato fluorescence across the whole embryo, which was equivalent the ES cells counterparts. Non-chimeric embryos were used as negative control. **l**, Sketch illustrating guided search model of TF combinatorial binding to signpost elements. In a standard random sampling, TFs transiently interact with low-affinity sites (dotted arrows), searching for gene regulatory targets (red bullseye). In guided search model, chromatin loops displaying oriented OSKM motifs (blue arrows) or loop junctions condensing GET/R motifs act as signposts that direct TF binding to their target enhancers.

# Reporting Summary

## Statistics

For all statistical analyses, confirm that the following items are present in the figure legend, table legend, main text, or Methods section.

| n/a | Confirmed | |
|---|---|---|
| ☐ | ☒ | The exact sample size (*n*) for each experimental group/condition, given as a discrete number and unit of measurement |
| ☐ | ☒ | A statement on whether measurements were taken from distinct samples or whether the same sample was measured repeatedly |
| ☐ | ☒ | The statistical test(s) used AND whether they are one- or two-sided<br>*Only common tests should be described solely by name; describe more complex techniques in the Methods section.* |
| ☐ | ☒ | A description of all covariates tested |
| ☐ | ☒ | A description of any assumptions or corrections, such as tests of normality and adjustment for multiple comparisons |
| ☐ | ☒ | A full description of the statistical parameters including central tendency (e.g. means) or other basic estimates (e.g. regression coefficient) AND variation (e.g. standard deviation) or associated estimates of uncertainty (e.g. confidence intervals) |
| ☐ | ☒ | For null hypothesis testing, the test statistic (e.g. *F*, *t*, *r*) with confidence intervals, effect sizes, degrees of freedom and *P* value noted<br>*Give P values as exact values whenever suitable.* |
| ☐ | ☒ | For Bayesian analysis, information on the choice of priors and Markov chain Monte Carlo settings |
| ☐ | ☒ | For hierarchical and complex designs, identification of the appropriate level for tests and full reporting of outcomes |
| ☐ | ☒ | Estimates of effect sizes (e.g. Cohen's *d*, Pearson's *r*), indicating how they were calculated |

*Our web collection on statistics for biologists contains articles on many of the points above.*

## Software and code

Policy information about availability of computer code

| Data collection | - Sequencing data was collected using the Illumina 2500, 4000 and NovaSeq platforms.<br>- Western blots and EMSAs were imaged using BioRAD ChemiDoc MP imaging system.<br>- Fluorescent immunostaining images were captured by Nikon Eclipse T! microscope and IRIS Digital Cell Imaging System.<br>- DNA fragment sizes were measure by Agilent 2200 Tapestation.<br>- Histones were analyzed by reversed-phase high pressure liquid chromatography using Waters 2695 system equipped with a Vydac 218TP C18 HPLC column, and Waters 996 Photodiode Array Detector.<br>- LCMS analyses were performed on a TripleTOF 5600+ mass spectrometer (AB SCIEX) coupled with M5 MicroLC system (AB SCIEX/Eksigent) and PAL3 autosampler.<br>- For colony counting whole wells were imaged at a resolution of 4 μm/pixel using a CELIGO image cytometer.<br>- For flow cytometry, a Beckman Coulter (Gallios) flow cytometer was used. Data acquisition and analysis were conducted using the Kaluza Software (Version 1.0.14029.14028). |
|---|---|
| Data analysis | data was analyzed using the following open source and commercial softwares: FASTQC v0.11.8, Bowtie v2.3, Bedtools V2.28, Picard v2.20, MACS v2.1.1, DeepTools V2, Qhttps://git.ecdf.ed.ac.uk/soufi_lab/motif_mononucleosomeualimap V2.2.1, SAMTool2 v1.3.1, BEDOPS V2.49, MEME v5.0.2, R v3.6 (various R packages as indicated in the methods), Cutadapt v3.3, STAR 2.7, DANPOS2, plot2DO v1.0, cLoops2, Coolpup.py v0.9.7, Pymol v3.0.3 with APBS plugin, IGV version 2.13.2 run with JAVA 11.0.13 (OpenJDK 64-bit). AlphaFold Multimer (AlphaFold2 version 2.3.1 on COSMIC2 cloud server), ImageJ (v1.54f) and Java 1.8.0_322 (64-bit), PeakView (version 2.2.0.11391, ABSciex), MicroApp (version 2.0.1.2133, ABSciex), UpSetR (version 1.4.0), FitHiChIP (version 11.0).<br><br>Custom scripts were deposited in UOE GitHub: |

<https://git.ecdf.ed.ac.uk/soufi_lab/motif_mononucleosome>
<https://git.ecdf.ed.ac.uk/soufi_lab/motif_nucleosome_arrays>

For manuscripts utilizing custom algorithms or software that are central to the research but not yet described in published literature, software must be made available to editors and reviewers. We strongly encourage code deposition in a community repository (e.g. GitHub). See the Nature Portfolio guidelines for submitting code & software for further information.

## Data

Policy information about availability of data

All manuscripts must include a data availability statement. This statement should provide the following information, where applicable:
- Accession codes, unique identifiers, or web links for publicly available datasets
- A description of any restrictions on data availability
- For clinical datasets or third party data, please ensure that the statement adheres to our policy

All next generation sequencing data generated as part of this study have been deposited in the Gene Expression Omnibus (GEO) under the series accession number GSE201852 (released upon publication, reviewers' access token: izkngkyynzutvmv). Previously published H3K27ac ChIP-seq, RNA-seq and ATAC-seq data were obtained from GSE98124, GSE171127 and GSE70234. Histone H1 ChIP-seq was obtained from GSE156697 and GSE46134. Oct4 ChIP-seq in MEFs-Oct4-48h were obtained from GSE168142. CTCF, PolII, P300, H3K4me1/3 ChIP-seq from GSE29184 and GSE29218. H3K9me1/me2 ChIP-seq from GSE54412. Rad21 ChIP-seq from GSE111820 and GSE115984. Brn2 ChIP-seq from GSE35496. HP1α, Suv39h1/2, and H3K9me3 ChIP-seq are from GSE57092. Oct4 and Sox2 ChIP-seq from secondary OSKM reprogramming system obtained from GSE101905. OSKM ChIP-seq in Mbd3f/- secondary reprogramming system are obtained from GSE102518. All data aligned to mouse reference genome MGSCv37 (mm9).

## Research involving human participants, their data, or biological material

Policy information about studies with human participants or human data. See also policy information about sex, gender (identity/presentation), and sexual orientation and race, ethnicity and racism.

| | |
|---|---|
| Reporting on sex and gender | N/A |
| Reporting on race, ethnicity, or other socially relevant groupings | N/A |
| Population characteristics | N/A |
| Recruitment | N/A |
| Ethics oversight | N/A |

Note that full information on the approval of the study protocol must also be provided in the manuscript.

# Field-specific reporting

Please select the one below that is the best fit for your research. If you are not sure, read the appropriate sections before making your selection.

☒ Life sciences    ☐ Behavioural & social sciences    ☐ Ecological, evolutionary & environmental sciences

For a reference copy of the document with all sections, see nature.com/documents/nr-reporting-summary-flat.pdf

# Life sciences study design

All studies must disclose on these points even when the disclosure is negative.

| | |
|---|---|
| Sample size | Sample sizes are indicated in each Figure legend. Generally, three to six biological replicates were used. This was determined according to established methods in the field and previous experience such as that in (Soufi et al; DOI:10.1016/j.cell.2012.09.045 and Soufi et al; DOI:10.1016/j.cell.2015.03.017), which allowed us to predetermine the number of sample size of each experiment. The number of ChIP-seq peaks were identified to be significantly enriched over input by MACS2 software (q = 0.01). ATAC-seq enrichment within was considered significant based on 20 normalized reads or more. All sequencing libraries were normalized to sequencing depth. |
| Data exclusions | All replicates and no data were excluded. All experiment included positive, negative or internal controls. To remove over-represented sequencing data, ENCODE black list were excluded from all next-generation sequencing data. Sequencing duplicated reads generated by PCR and optical density and non-uniques sequences were removed from all next-generation sequencing data to remove PCR bias. |
| Replication | All replication attempt were successful. Each Experiment was repeated as indicated in the figure legends and the Methods section. |
| Randomization | For pileup Micro-C, 10 random control regions located between 100 kb and 1Mb away per interval per averaged window were used. Random sequences with similar group number and size for nucleosome array were generated using Bedtools shuffle. Motif analysis used random sequences that are 1kb away from ChIP-seq peaks. |

| | |
|---|---|
| | All reprogramming experiments included non-reprogrammed controls or infected controls with empty vectors so so each experimental condition can be allocated into separate groups. |
| Blinding | H1-KO and H1-OE for ATAC-seq and Micro-C were blinded.<br>Sequencing data showed the same diversity and were normalized to the sequencing depth. No blinding was required when bias can be quantified and removed. |

# Reporting for specific materials, systems and methods

We require information from authors about some types of materials, experimental systems and methods used in many studies. Here, indicate whether each material, system or method listed is relevant to your study. If you are not sure if a list item applies to your research, read the appropriate section before selecting a response.

## Materials & experimental systems

| n/a | Involved in the study |
|---|---|
| ☐ | ☒ Antibodies |
| ☐ | ☒ Eukaryotic cell lines |
| ☒ | ☐ Palaeontology and archaeology |
| ☐ | ☒ Animals and other organisms |
| ☒ | ☐ Clinical data |
| ☒ | ☐ Dual use research of concern |
| ☒ | ☐ Plants |

## Methods

| n/a | Involved in the study |
|---|---|
| ☐ | ☒ ChIP-seq |
| ☐ | ☒ Flow cytometry |
| ☒ | ☐ MRI-based neuroimaging |

## Antibodies

| Antibodies used | All antibodies and amounts are listed in Table2 in the method section. |
|---|---|
| Validation | All antibodies were obtained from commercial sources and were validated by the company; refer to the company website for detailed validation analysis. The antibodies were also validated in our laboratory by ChIP-seq (specific peaks), western blots (one band corresponding to the expected size), and immuno-fluorescence (nuclear staining in ES cells). |

## Eukaryotic cell lines

Policy information about cell lines and Sex and Gender in Research

| Cell line source(s) | Mouse embryonic fibroblasts (MEFs) were derived from 129 strain mice kept at the University of Edinburgh animal facility. HEK 293T cell lines were used for lentivirus production (TAKARA #63218). These cells are isolated from human embryonic kidneys (HEK) and the 293T cells are transformed with large T antigen. HEK 293T cell line was originally created in Michele Calos's lab at Stanford (DuBridge et al; doi:10.1128/MCB.7.1.379).<br>Mouse iPSCs, iTSCs, ESCs, and TSCs were all generated in Yossi Buganim's laboratory in the Hebrew University and Hadassah Medical Center. |
|---|---|
| Authentication | MEFs were authenticated by the University of Edinburgh Animal facility by genotyping DNA extracted from the tail clip or ear notch using PCR. Mouse iPSCs, iTSCs, ESCs, and TSCs and their derivatives were authenticated by PCR and immunofluorescence in Yossi Buganim's laboratory as previously reported (Benchetrit et al; DOI: 10.1016/j.stem.2019.03.018).<br>The original HEK 293T cell line was authenticated previously (DuBridge et al; doi:10.1128/MCB.7.1.379). The commercial Lenti-X 293T Cell Line used in this study is a human embryonic kidney (HEK) cell line, transformed with adenovirus type 5 DNA, that also expresses the SV40 large T antigen. The cell line was subcloned for high transfectability and high-titer virus production by TAKARA (JAPAN). This lot of cells has been tested and found to be free of Mycoplasma contamination by TAKARA. |
| Mycoplasma contamination | All cells were routinely checked for Mycoplasma contamination and tested negative. |
| Commonly misidentified lines<br>(See ICLAC register) | No misidentified cell lines were used in this study. |

## Animals and other research organisms

Policy information about studies involving animals; ARRIVE guidelines recommended for reporting animal research, and Sex and Gender in Research

| Laboratory animals | Blastocyst were derived from mouse CB6F1 host females after mating with CB6F1 males. Injected blastocysts were transferred to 2.5dpc pseudo-pregnant CD1/ICR females. |
|---|---|
| Wild animals | no wild animals were used in the study. |

| Reporting on sex | Sex information has not been collected and no sex- or gender-based analysis have been carried out as it is not relevant to this study. |
|---|---|
| Field-collected samples | no field collected samples were used in the study. |
| Ethics oversight | All animal experiments for the iPSC and iTSC generation from mouse embryonic fibroblasts were approved by the University of Edinburgh Animal Welfare and Ethical Review Body, performed at the University of Edinburgh, and carried out according to regulations specified by the Home Office and Project License. All reprogramming experiments have been approved by the University of Edinburgh SBS ethics committee (asoufi-0001). The joint ethics committee (IACUC) of the Hebrew University and Hadassah Medical Center approved the study protocol for animal welfare. The Hebrew University is an AAALAC international accredited institute. This research was performed in compliance with the Ethic Committee of Shaare Zedek Medical Center, the joint ethics committee (IACUC) of the Hebrew University and Hadassah Medical Center and the National ethic committee (Israel health ministry) and NIH. |

Note that full information on the approval of the study protocol must also be provided in the manuscript.

# Plants

| Seed stocks | N/A |
|---|---|
| Novel plant genotypes | N/A |
| Authentication | N/A |

# ChIP-seq

## Data deposition

☒ Confirm that both raw and final processed data have been deposited in a public database such as GEO.

☒ Confirm that you have deposited or provided access to graph files (e.g. BED files) for the called peaks.

| Data access links
*May remain private before publication.* | To review GEO accession GSE201852:
Go to https://www.ncbi.nlm.nih.gov/geo/query/acc.cgi?acc=GSE201852
Enter token izkngkyynzutvmv into the box |
|---|---|
| Files in database submission | GSM6077123  Oct4-ChIP-seq, mESCs, biol rep 1
GSM6077124  Oct4-ChIP-seq, mESCs, biol rep 2
GSM6077125  Sox2-ChIP-seq, mESCs, biol rep 1
GSM6077126  Sox2-ChIP-seq, mESCs, biol rep 2
GSM6077127  Klf4-ChIP-seq, mESCs, biol rep 1
GSM6077128  Klf4-ChIP-seq, mESCs, biol rep 2
GSM6077129  Myc-ChIP-seq, mESCs, biol rep 1
GSM6077130  Myc-ChIP-seq, mESCs, biol rep 2
GSM6077131  Input-ChIP-seq, mESCs, biol rep 1
GSM6077132  Input-ChIP-seq, mESCs, biol rep 2
GSM6077133  Input-ChIP-seq, miTSCs, biol rep 1
GSM6077134  Input-ChIP-seq, miTSCs, biol rep 2
GSM6077135  Gata3-ChIP-seq, miTSCs, biol rep 1
GSM6077136  Gata3-ChIP-seq, miTSCs, biol rep 2
GSM6077137  Eomes-ChIP-seq, miTSCs, biol rep 1
GSM6077138  Eomes-ChIP-seq, miTSCs, biol rep 2
GSM6077139  Tfap2c-ChIP-seq, miTSCs, biol rep 1
GSM6077140  Tfap2c-ChIP-seq, miTSCs, biol rep 2
GSM6077141  Myc-ChIP-seq, miTSCs, biol rep 1
GSM6077142  Myc-ChIP-seq, miTSCs, biol rep 2
GSM6077143  Sox2-ChIP-seq, miTSCs, biol rep 1
GSM6077144  Sox2-ChIP-seq, miTSCs, biol rep 2
GSM6077145  Oct4-ChIP-seq, OSKM-48h, biol rep 1
GSM6077146  Oct4-ChIP-seq, OSKM-48h, biol rep 2
GSM6077147  Sox2-ChIP-seq, OSKM-48h, biol rep 1
GSM6077148  Sox2-ChIP-seq, OSKM-48h, biol rep 2
GSM6077149  Klf4-ChIP-seq, OSKM-48h, biol rep 1
GSM6077150  Klf4-ChIP-seq, OSKM-48h, biol rep 2
GSM6077151  Myc-ChIP-seq, OSKM-48h, biol rep 1
GSM6077152  Myc-ChIP-seq, OSKM-48h, biol rep 2
GSM6077153  Input-ChIP-seq, OSKM-48h, biol rep 1 |

```
GSM6077154  Input-ChIP-seq, OSKM-48h, biol rep 2
GSM6077155  Gata3-ChIP-seq, GETM-48h, biol rep 1
GSM6077156  Gata3-ChIP-seq, GETM-48h, biol rep 2
GSM6077157  Eomes-ChIP-seq, GETM-48h, biol rep 1
GSM6077158  Eomes-ChIP-seq, GETM-48h, biol rep 2
GSM6077159  Tfap2c-ChIP-seq, GETM-48h, biol rep 1
GSM6077160  Tfap2c-ChIP-seq, GETM-48h, biol rep 2
GSM6077161  Myc-ChIP-seq, GETM-48h, biol rep 1
GSM6077162  Myc-ChIP-seq, GETM-48h, biol rep 2
GSM6077163  Input-ChIP-seq, GETM-48h, biol rep 1
GSM6077164  Input-ChIP-seq, GETM-48h, biol rep 2
GSM6077165  Gata3-ChIP-seq, GETMR-48h, biol rep 1
GSM6077166  Gata3-ChIP-seq, GETMR-48h, biol rep 2
GSM6077167  Eomes-ChIP-seq, GETMR-48h, biol rep 1
GSM6077168  Eomes-ChIP-seq, GETMR-48h, biol rep 2
GSM6077169  Tfap2c-ChIP-seq, GETMR-48h, biol rep 1
GSM6077170  Tfap2c-ChIP-seq, GETMR-48h, biol rep 2
GSM6077171  Myc-ChIP-seq, GETMR-48h, biol rep 1
GSM6077172  Myc-ChIP-seq, GETMR-48h, biol rep 2
GSM6077173  Esrrb-ChIP-seq, GETMR-48h, biol rep 1
GSM6077174  Esrrb-ChIP-seq, GETMR-48h, biol rep 2
GSM6077175  Input-ChIP-seq, GETMR-48h, biol rep 1
GSM6077176  Input-ChIP-seq, GETMR-48h, biol rep 2
GSM6077177  Gata4-ChIP-seq, BS9G4M-48h, biol rep 1
GSM6077178  Gata4-ChIP-seq, BS9G4M-48h, biol rep 2
GSM6077179  Brn2-ChIP-seq, BS9G4M-48h, biol rep 1
GSM6077180  Brn2-ChIP-seq, BS9G4M-48h, biol rep 2
GSM6077181  Sox9-ChIP-seq-BS9G4M48h_rep1
GSM6077182  Sox9-ChIP-seq, BS9G4M-48h, biol rep 2
GSM6077183  Myc-ChIP-seq, BS9G4M-48h, biol rep 1
GSM6077184  Myc-ChIP-seq, BS9G4M-48h, biol rep 2
GSM6077185  Input-ChIP-seq, BS9G4M-48h, biol rep 1
GSM6077186  Input-ChIP-seq, BS9G4M-48h, biol rep 2
GSM6077187  Esrrb-ChIP-seq, mESCs
GSM6077188  Esrrb-ChIP-seq, miTSCs
GSM6077189  1U MNase, MEFs
GSM6077190  4U MNase, MEFs
GSM6077191  16U MNase, MEFs
GSM6077192  64U MNase, MEFs
GSM6077193  1U MNase, mESCs
GSM6077194  4U MNase, mESCs
GSM6077195  16U MNase, mESCs
GSM6077196  64U MNase, mESCs
GSM6077197  1U MNase, mTSCs
GSM6077198  4U MNase, mTSCs
GSM6077199  16U MNase, mTSCs
GSM6077200  64U MNase, mTSCs
GSM6077201  15U Mnase MicroC, MEFs
GSM6077202  20U Mnase MicroC, MEFs
GSM6077203  15U Mnase MicroC, mESCs
GSM6077204  20U Mnase MicroC, mESCs
GSM6077205  15U Mnase MicroC, mTSCs
GSM6077206  20U Mnase MicroC, mTSCs
GSM8351362  15U Mnase MicroC, MEF-H1OE
GSM8351363  20U Mnase MicroC, MEF-H1OE
GSM8351364  15U Mnase MicroC, MEF-H1KD
GSM8351365  20U Mnase MicroC, MEF-H1KD
GSM8354076  MEF, ATAC-seq, rep_1
GSM8354077  MEF, ATAC-seq, rep_2
GSM8354078  MEF, empty vector control, ATAC-seq, rep_1
GSM8354079  MEF, empty vector control, ATAC-seq, rep_2
GSM8354080  MEF, empty vector control, ATAC-seq, rep_3
GSM8354081  MEF, empty vector control, ATAC-seq, rep_4
GSM8354082  MEF, H1KD, ATAC-seq, rep_1
GSM8354083  MEF, H1KD, ATAC-seq, rep_2
GSM8354084  MEF, H1KD, ATAC-seq, rep_3
GSM8354085  MEF, H1KD, ATAC-seq, rep_4
GSM8354086  MEF, H1OE, ATAC-seq, rep_1
GSM8354087  MEF, H1OE, ATAC-seq, rep_2
GSM8354088  MEF, H1OE, ATAC-seq, rep_3
GSM8354089  MEF, H1OE, ATAC-seq, rep_4
GSM8354090  MKOS_MEF, 0h, ATAC-seq, rep_1
GSM8354091  MKOS_MEF, 0h, ATAC-seq, rep_2
GSM8354092  MKOS_MEF, 72h, ATAC-seq, rep_1
GSM8354093  MKOS_MEF, 72h, ATAC-seq, rep_2
GSM8354094  MKOS_MEF, 0h, H1KD, ATAC-seq, rep_1
```

GSM8354095  MKOS_MEF, 0h, H1KD, ATAC-seq, rep_2
GSM8354096  MKOS_MEF, 72h, H1KD, ATAC-seq, rep_1
GSM8354097  MKOS_MEF, 72h, H1KD, ATAC-seq, rep_2
GSM8354098  MKOS_MEF, 72h, H1OE, ATAC-seq, rep_1
GSM8354099  MKOS_MEF, 72h, H1OE, ATAC-seq, rep_2
GSM8354100  MKOS_MEF, 0h, empty_ctrl, ATAC-seq, rep_1
GSM8354101  MKOS_MEF, 0h, empty_ctrl, ATAC-seq, rep_2
GSM8354102  MKOS_MEF, 72h, empty_ctrl, ATAC-seq, rep_1
GSM8354103  MKOS_MEF, 72h, empty_ctrl, ATAC-seq, rep_2

**Genome browser session**
(e.g. UCSC)

Following normalized files have been submitted in GEO, which can be used in genome browser:

GSE201852_Brn2-ChIP-seq_BS9G4M48h.SeqDepthNorm.bw 331.8 Mb (http) BW
GSE201852_Eomes-ChIP-seq_GETM48h.SeqDepthNorm.bw 321.9 Mb (http) BW
GSE201852_Eomes-ChIP-seq_GETMR48h.SeqDepthNorm.bw 345.4 Mb (http) BW
GSE201852_Eomes-ChIP-seq_miTSC.SeqDepthNorm.bw 257.9 Mb (http) BW
GSE201852_Esrrb-ChIP-seq_GETMR48h.SeqDepthNorm.bw 374.5 Mb (http) BW
GSE201852_Esrrb-ChIP-seq_mESC.SeqDepthNorm.bw 259.6 Mb (http) BW
GSE201852_Esrrb-ChIP-seq_miTSC.SeqDepthNorm.bw 331.3 Mb (http) BW
GSE201852_Gata3-ChIP-seq_GETM48h.SeqDepthNorm.bw 310.4 Mb (http) BW
GSE201852_Gata3-ChIP-seq_GETMR48h.SeqDepthNorm.bw 306.6 Mb (http) BW
GSE201852_Gata3-ChIP-seq_miTSC.SeqDepthNorm.bw 287.2 Mb (http) BW
GSE201852_Gata4-ChIP-seq_BS9G4M48h.SeqDepthNorm.bw 249.1 Mb (http) BW
GSE201852_Input-ChIP-seq_BS9G4M48h.SeqDepthNorm.bw 356.4 Mb (http) BW
GSE201852_Input-ChIP-seq_GETM48h.SeqDepthNorm.bw 357.4 Mb (http) BW
GSE201852_Input-ChIP-seq_GETMR48h.SeqDepthNorm.bw 353.5 Mb (http) BW
GSE201852_Input-ChIP-seq_OSKM48h.SeqDepthNorm.bw 390.1 Mb (http) BW
GSE201852_Input-ChIP-seq_mESC.SeqDepthNorm.bw 491.4 Mb (http) BW
GSE201852_Input-ChIP-seq_miTSC.SeqDepthNorm.bw 507.2 Mb (http) BW
GSE201852_Klf4-ChIP-seq_OSKM48h.SeqDepthNorm.bw 321.6 Mb (http) BW
GSE201852_Klf4-ChIP-seq_mESC.SeqDepthNorm.bw 326.2 Mb (http) BW
GSE201852_MEF-H1KD_combined_MicroC.mm9.mapq_30.100.mcool 3.1 Gb (http) MCOOL
GSE201852_MEF-H1OE_combined_MicroC.mm9.mapq_30.100.mcool 2.4 Gb (http) MCOOL
GSE201852_MEF_H1KD_merged_SeqDepthNorm.bw 448.5 Mb (http) BW
GSE201852_MEF_H1OE_merged_SeqDepthNorm.bw 420.0 Mb (http) BW
GSE201852_MEF_combined_MicroC.mm9.mapq_30.100.mcool 2.2 Gb (http) MCOOL
GSE201852_MEF_empty_merged_SeqDepthNorm.bw 426.5 Mb (http) BW
GSE201852_MEF_merged_SeqDepthNorm.bw 235.0 Mb (http) BW
GSE201852_MKOS_MEF_0hr_merged_SeqDepthNorm.bw 266.1 Mb (http) BW
GSE201852_MKOS_MEF_72hr_merged_SeqDepthNorm.bw 283.7 Mb (http) BW
GSE201852_MKOS_MEF_H1KD_0hr_merged_SeqDepthNorm.bw 170.7 Mb (http) BW
GSE201852_MKOS_MEF_H1KD_72hr_merged_SeqDepthNorm.bw 156.2 Mb (http) BW
GSE201852_MKOS_MEF_H1OE_72hr_merged_SeqDepthNorm.bw 392.3 Mb (http) BW
GSE201852_MKOS_MEF_empty_0hr_merged_SeqDepthNorm.bw 181.4 Mb (http) BW
GSE201852_MKOS_MEF_empty_72hr_merged_SeqDepthNorm.bw 224.8 Mb (http) BW
GSE201852_Myc-ChIP-seq_BS9G4M48h.SeqDepthNorm.bw 233.3 Mb (http) BW
GSE201852_Myc-ChIP-seq_GETM48h.SeqDepthNorm.bw 292.1 Mb (http) BW
GSE201852_Myc-ChIP-seq_GETMR48h.SeqDepthNorm.bw 253.1 Mb (http) BW
GSE201852_Myc-ChIP-seq_OSKM48h.SeqDepthNorm.bw 286.3 Mb (http) BW
GSE201852_Myc-ChIP-seq_mESC.SeqDepthNorm.bw 318.1 Mb (http) BW
GSE201852_Myc-ChIP-seq_miTSC.SeqDepthNorm.bw 378.6 Mb (http) BW
GSE201852_Oct4-ChIP-seq_OSKM48h.SeqDepthNorm.bw 361.7 Mb (http) BW
GSE201852_Oct4-ChIP-seq_mESC.SeqDepthNorm.bw 338.9 Mb (http) BW
GSE201852_Sox2-ChIP-seq_OSKM48h.SeqDepthNorm.bw 432.3 Mb (http) BW
GSE201852_Sox2-ChIP-seq_mESC.SeqDepthNorm.bw 333.2 Mb (http) BW
GSE201852_Sox2-ChIP-seq_miTSC.SeqDepthNorm.bw 251.6 Mb (http) BW
GSE201852_Sox9-ChIP-seq_BS9G4M48h.SeqDepthNorm.bw 301.4 Mb (http) BW
GSE201852_Tfap2c-ChIP-seq_GETM48h.SeqDepthNorm.bw 311.3 Mb (http) BW
GSE201852_Tfap2c-ChIP-seq_GETMR48h.SeqDepthNorm.bw 376.4 Mb (http) BW
GSE201852_Tfap2c-ChIP-seq_miTSC.SeqDepthNorm.bw 228.4 Mb (http) BW
GSE201852_mESC_combined_MicroC.mm9.mapq_30.100.mcool 2.7 Gb (http) MCOOL
GSE201852_mTSC_combined_MicroC.mm9.mapq_30.100.mcool 2.4 Gb (http) MCOOL

## Methodology

**Replicates**

Three ChIP replicates were pooled to make a DNA library for each ChIP-seq experiment and two independent replicates were carried out. All other sequencing data were carried out in duplicates or a pool of triplicates. Different MNase concentrations were used to generate independent sequencing libraries.

**Sequencing depth**

Around 50-60 million pair-end reads were obtained on average from each ChIP-seq.

**Antibodies**

Listed in Table 2 of the method section.

**Peak calling parameters**

Duplicates were removed from the aligned pair-end BAM files using Picard prior to peak calling. TF peaks (sample files) showing significant enrichment over input DNA (control files) obtained from the same cells were called using MACS2 (version 2.1.1.20160309)

and a fragment size of 200 bp (--nomodel --extsize 200) and were controlled to q value (minimum FDR) cut-off of 0.01 (-q 0.01). The peaks that overlapped with the ENCODE mm9 blacklist were removed using the bedtools intersect function (flag –v).
To identify broadPeaks of TF binding, peaks were called as using MACS2 with the following flags: -B --broad-cutoff 0.1 --broad --nomodel --extsize 200. Regions that overlapped with the ENCODE blacklist were removed using the bedtools intersect function (flag –v).

| Data quality | Quality controls of DNA libraries were carried out by DNA fragment size distribution using Tapestation and Bioanalyzer (Agilent). Sequencing quality was assessed by mean quality scores using FASTQC and only Phred scores above 30 were considered. Sequence duplication and library complexity was assessed by MutiQC and Qualimap prior to further analysis. Duplicates were removed by Picard and adapters by Cutadapt. Libraries were normalized by squencing depth to 1X genome coverage using DeepTools. |
| --- | --- |
| Software | FASTQC v0.11.8, MultiQC v1.3, Bowtie v2.3, Bedtools V2.28, Picard v2.20, MACS v2.1.1, DeepTools V2, Qualimap V2.2.1, SAMTool2 v1.3.1, MEME v5.0.2, R v3.6 (various R packages as indicated in the methods), BEDOPS V2.49 and Cutadapt v3.3. |

# Flow Cytometry

## Plots

Confirm that:

☒ The axis labels state the marker and fluorochrome used (e.g. CD4-FITC).

☒ The axis scales are clearly visible. Include numbers along axes only for bottom left plot of group (a 'group' is an analysis of identical markers).

☒ All plots are contour plots with outliers or pseudocolor plots.

☒ A numerical value for number of cells or percentage (with statistics) is provided.

## Methodology

| Sample preparation | cells were first trypsinized and then neutralized with medium containing 10% fetal bovine serum (FBS). Following this, the cells were centrifuged and washed twice with phosphate-buffered saline (PBS) to ensure the removal of any residual trypsin and medium. The washed cells were then resuspended in PBS for subsequent analysis. |
| --- | --- |
| Instrument | Flow cytometric analysis was performed using a Beckman Coulter (Gallios) flow cytometer. |
| Software | Data acquisition and analysis were conducted using the Kaluza Software (Version 1.0.14029.14028). |
| Cell population abundance | The fluorescent markers eGFP and tdTomato were used to identify and quantify specific cell populations. |
| Gating strategy | To remove dead cells, all samples were initially gated using the FSC-A/SSC-A gating to identify the live cell population (below 200 FS Area). To remove cell doublets, single cells were selected by gating forward scatter height vs area. The positively fluorescent cells were gated based on the fluorescent intensity of positive control cells. |

☒ Tick this box to confirm that a figure exemplifying the gating strategy is provided in the Supplementary Information.

