## [Peer Review file · Nature]

Nucleosome fibre topology guides transcription factor binding to enhancers

Corresponding Author: Dr Abdenour Soufi

Version 0:

Reviewer comments:

Referee #1

(Remarks to the Author)

O'Dwyer et al. explore the role of the 3D chromatin context in TF-induced reprogramming using various TF-induced cell fate transitions, nucleosome spacing and TF binding, chromatin accessibility and 3D chromatin folding. They find that the binding of individual TFs to chromatin is well explained by the presence or absence of the respective TF motif on mononucleosomes. By contrast, the co-binding of cell fate-instructive TFs such as OKS or GET instead follows more complex rules that involve directional binding of TFs within and nearby nucleosomal arrays. These binding patterns dynamically change during reprogramming in a seemingly non-random manner. Moreover, the co-binding of TFs within and nearby nucleosomal arrays are different for different TF cocktails and correlate with 3D chromatin structure and H1 binding. Based on the comparison of TF co-binding patterns during early and late stages of reprogramming to either iPSCs or iTSCs, the authors propose that chromatin fibers serve as "signposts" for TFs to engage with closed chromatin before being able to access nearby cis regulatory elements (CREs).

This is a comprehensive study that offers intriguing new principles by which TF combinations may engage with chromatin to active CREs during (induced) cell fate transitions. As such, the manuscript could be of interest to a general readership. However, I have some concerns about the experimental setup and the conclusions drawn from the analyses. Moreover, experiments to validate some of the authors' model are missing. The impact of the study would be elevated if the authors could show that suppression of histone H1 expression during early OKS reprogramming facilitates acquisition of an ESC-like CRE state as one would predict. Similarly, it would help the authors' argument if they could show that an iPSC reprogramming system that is more efficient and homogeneous (via use of small molecules or perturbation of other factors) reduces the binding of TFs to signpost elements nearby ESC-like CREs.

Specific comments:

1. All reprogramming experiments appear to be based on the introduction of individual dox-inducible vectors into MEFs. However, it is unclear what fraction of cells actually express all TFs. This variable can obviously change the authors' interpretations in a major way. It would have been preferable to use polycistronic vectors for these experiments to avoid heterogeneity of TF expression. The authors need to either repeat some of the key experiments with a polycistronic vector or, at a minimum, demonstrate that the majority of cells infected with multiple vectors express all TFs.
2. A key conclusion of this manuscript is that the binding of single TFs to mononucleosomes (in MEFs that presumably express multiple TFs such as OKSM or GETM; see point 1) is explained by the presence or absence of the respective TF motif. Is it possible that this pattern was observed because of the heterogeneity of MEFs transduced with multiple TF vectors, i.e. are the authors looking at cells that only received one or two TF? To control for this possibility, the authors should perform some of the assays with MEFs expressing a single TF (in addition to the meta-analyses of published data).
3. The authors need to characterize the TF systems they are using in terms of efficiency of reprogramming to iPSCs (OKSM) or iTSCs (GETM), and the fraction of iPSCs vs iTSCs that are being generated with GETMR. Also, the authors need to do some basic characterization of the generated iPSCs and iTSCs using marker analysis after transgene silencing and differentiation potential.
4. What is the chromatin status and physiological role, if known, of TF motifs that serve as signposts? Are these elements

normally bound and active in other cell types?

5. The impact of this study would be elevated if the authors could connect the observation of TF binding to signpost elements to some functional readout. For example, they could show that depletion of H1 (in the OKSM system) or the use of a more efficient iPSC reprogramming system reduces the binding of TFs to signposts. Alternatively, they could assess TF co-binding patterns in reprogramming-proficient (SSEA1+) vs reprogramming-deficient (SSEA1-) intermediates when using a homogeneous OKSM system.

6. The GETMR data are interesting but complicated by the fact that two different cell fates, iPSCs and iTSCs, are being generated from the same starting population. It would be more informative to assess whether GTMR or TMR, which like OKS generates iPSCs, follow a similar or different mechanism of TF co-binding (i.e. more like OKSM or more like GETM?).

7. TF-induced reprogramming involves ectopic and often super-physiological expression of TFs in heterologous cell types, raising the question of whether the authors' observations extend to physiological cell fate transitions or whether they are an "artifact" of forced TF expression systems. For example, is it possible that forced expression of OKSM in established iPSCs or ESCs would also keep signpost elements engaged by TFs? The authors should at least discuss these possibilities in the manuscript.

8. A previous study by Plath et al found that in mouse, OKSM initially engage with open chromatin rather than heterochromatin (PMID: 28111071). How do the authors reconcile their data with this previous observation (using a more homogeneous polycistronic OKSM transgene)?

9. Why did the authors show ATAC-Seq data of the 72h time point of reprogramming even though most assays were performed at the 48h time point when cells were shown to be more homogeneous?

10. The authors' use of the terms sign-posts, search-guards, search-targets, etc. are a bit vague and ambiguous, especially since they are used to describe two seemingly different mechanisms (OKSM vs GETM reprogramming).

11. Experiments using BSG expression are not particularly helpful but rather distracting and the authors should consider removing the data to streamline the manuscript.

Referee #2

(Remarks to the Author)

Complex cell state transitions like reprogramming require the combined action of multiple TFs. How these factors engage which each other to associate with chromatin remains largely unexplored.

In this study, the authors have addressed this question using four different TF cocktails that each result in a different reprogramming outcome. By combining data on TF binding (ChIP-seq) with nucleosome organization (MNase-seq and Micro-C) at different stages of reprogramming, the authors report a number of new findings that provide important new insights into how TFs bind shared regions in the genome:

1. Co-binding mostly occurs over larger arrays of nucleosomes
2. Different TF cocktails show a different syntax of co-binding
3. The authors identify "signpost" elements, where TF cocktails bind at early stages of reprogramming. Signposts neighbor regulatory elements that are bound by the same TF cocktails at later stages, indicating that they help making these regulatory elements available.

This study presents an impressive and exciting set of data, and particularly the identification of signpost elements (and associated dynamics) provides an important new dimension to the dynamics of TF (co-)binding. As concluded by the authors, it's likely that these new findings expand to other complex cell state transitions (development, carcinogenesis) and possibly other cellular functions as well.

Unfortunately, the manuscript suffers from a number of issues that make it a sub-optimal choice for Nature (see below). Instead, it will be better suited, as a single submission or split up in multiple manuscripts, for a more specialized audience.

Issues:

1. **Biology:** whereas the authors suggest that the identified syntax for TF binding (as inferred through their binding motifs) must be essential to determine cooperative binding, a second explanation may be that this syntax is necessary for the correct functional and regulatory output of these TFs: relative TF motif orientation and spacing may be essential for correct protein-protein interactions (among members of the TF cocktails or when engaging with other factors) or creation of a correct higher-order nucleosome structure.

Without functional validations (e.g. by changing the orientation or relative position of TF motifs, followed by analysis of TF co-binding and transcriptional regulation), both explanations should be considered.

2. **Accessibility:** the manuscript tries to convey an immense amount of information, resulting both in very dense and complex text and figures. For the complete interpretation of the data, the supplemental data was often essential as well. The addition of more information to the main figures will therefore be required. Moreover, considerable prior knowledge on cellular reprogramming and nucleosome structure is a must. This is further complicated by the very high density of abbreviations in the text.

Combined, I therefore estimate that the current manuscript will be difficult to access for the wider audience of Nature. Instead, the results may profit from splitting in two separate manuscripts (temporal dynamics of TF binding and differences between TF cocktails).

3. Interpretation: the authors go to great lengths to interpret the data presented in their figures. While reviewing, in a number of cases it was either not obvious to me how the authors came to their conclusions or the interpretation appeared to contradict other findings.

- Line 53-55: I don't see how individual TF pioneer function and cooperative TF action are the most obvious aspects to raise questions about co-binding.

- Line 88-90: Why does this result suggest off-target binding? The function of these TFs could be different at different stages of reprogramming.

- Line 120-122: This is not a fully accurate description of the aim. The analysis does not show that motifs are differentially recognized, but rather that the relative position and orientation of motifs among each other is different.

- Line 126: And so is Klf4, as is mentioned in line 138.

- Line 134-138: See my first comment: the composite motif may not be essential to promoter co-binding but rather for correct functional/regulatory function.

- From line 152: I am confused about this section: in line 134 the co-occurrence of Oct4 and Sox2 motifs was discussed, but this section reports that Oct4 and Sox2 do not co-occur.

- Line 182-183: I do not see how a single TF could bind to an array of nucleosomes.

- Line 197-199: See my first comment: motif directionality may be essential for correct functional/regulatory function.

- Line 222-224: The fact that different TFs recognize different motifs is well known. This conclusion is a poor description of the previous paragraph.

- Line 344-345: It's difficult to imagine how the absence of its binding motif will instruct *Esrrb* to reduce *Tfap2c* binding. An alternative interpretation may be that an unknown downstream *Esrrb* target may be involved?

- Line 353: I'm not sure if this conclusion can be considered as competition (as mentioned in the abstract and figure title).

Referee #3

(Remarks to the Author)

The manuscript by O'Dwyer et al., entitled "Nucleosome fibre topology guides transcription factor co-binding to gene regulatory elements" investigated the mechanisms that mediate TF co-binding during cell fate transition. The authors compared four sets of transcription factor combinations during cellular reprogramming: OSKM (Oct4, Sox2, Klf4, and C-Myc), GETM (Gata3, Eomes, Tfap2c, and c-Myc), GETMR (Gata3, Eomes, Tfap2c, c-Myc, and *Esrrb*), and BS9G4M (Brn2, Sox9, Gata4, and c-Myc, control set) by integrating ChIP-seq, MNase-seq, ATAC-seq, and Micro-C data. The major conclusions include: (1) motif recognition on mono-nucleosomes does not explain TF co-binding; (2) TF co-binding is critical to cell reprogramming by comparing between early and final reprogramming cells; (3) TFs co-bindings occur in arrays of nucleosomes displaying distinct motifs in terms of distribution and orientation. These regions of nucleosome fibers serve as "signpost elements"; (4) they proposed that TF co-binding to signpost elements in early reprogramming can directly lead to tissue-specific genome occupancy in accordance with a guided search model; (5) two distinct guided-search modes were identified. one for OSK that follows motif orientation within H1-depleted chromatin environment and the other for GET that follows motif density in an H1-enriched environment.

Major comments, concerns, and questions:

TF co-binding has been recognized as a major mechanism that is critical to cellular fate transition and function. The authors obtained multiple genomics data to address this important question and performed sophisticated and comprehensive analysis of the data. I have several major concerns as described below:

(1) Some of the reported findings from the manuscript have been reported in previous studies. For example, it was reported that OSK co-occupancy drives reprogramming of somatic cells (Chronis et al. 2017). The specific motif distribution and orientation for TF co-binding was reported by Avsec, et al. (2021). They found that TF co-binding was guided by soft syntax rules, which followed clear intermotif distance-dependent relationships consistent with nucleosome-mediated cooperativity. The TF cooperativity associated with specific motif pairs was often directional and consistent with motifs mediating the role of pioneer TF with different strengths. Ibarra, et al. (2020) reported the DNA shape as driver for TF cooperativity.

(2) This study may be considered as a significant extension of the author's previous work (Soufi et al. 2015 PMID: 25892221) with more TFs combination sets and the Micro-C data. The authors found that OSK preferentially target silent sites enriched for nucleosomes in vivo.

(3) Regarding the proposed model of higher-order nucleosome topology mediating the TF co-binding, it is now just correlative. Much more evidence is required to substantiate the model.

(4) Reprogramming using multiple TFs is a dramatic non-physiological process. Most physiological cellular differentiation involves relatively mild or stepwise processes. How does the guided search model apply to those processes?

(5) Except for Figure 2H/Figure 3C and F, which presented explicit examples, most of this manuscript presented high-level abstract aggregation analyses. To facilitate the readers to understand the results, I suggest the authors to include specific examples for a majority of the aggregation analyses, which can also directly show the quality of sequencing data.

(6) The description of analysis methods was comprehensive. However, necessary codes and test data should be deposited at GitHub for readers to check reproducibility, especially for Figure 1D, Figure 2C, and other similar figures.

- (7) As the control set, what are the overlaps for BS9G4M with ESC or TSC (Extended Figure 1B)?
- (8) "To assess whether motif readout plays a role in TF co-binding to nucleosome arrays, we mapped motif enrichment across nucleosome arrays within close chromatin." What is the logic behind the selection to only study the close chromatin?
- (9) "Isolating OSK nucleosomes arrays containing 4 or more Sox2 motifs/kb on either strand revealed that these arrays contain..." Why Sox motif but not Oct4 motif?
- (10) "In fully reprogrammed cells, nucleosome arrays co-bound by OSK (CRES) contain almost no Oct4 and Sox2 motifs, no orientation-specific motif distribution, and smaller in size". Were these OSK co-bound peaks weak ChIP-seq peaks and therefore could not represent true co-binding of OSK? Same question to the fully reprogrammed GET nucleosome arrays.
- (11) "Interestingly, transitioning from early to fully reprogrammed cells was concurrent with a lateral shift of OSK co-binding across the upstream near-border of the nucleosome arrays following OSK motif direction (Fig. 2G,H)". What's the fraction of this shifting profile for co-binding events? I feel only if the majority of the nucleosome arrays show this shifting pattern, this observation is interesting. If indeed the majority of them show the shifting pattern, I accept the conclusion of "the early co-binding of both OSK and GET is not random and can directly lead to lineage-specific binding to CREs in fully reprogrammed cells.....guiding TF co-binding to CREs targets". However, globally GET do not show the same shifting pattern. How do the authors explain the discrepancy for a very basic biological process?
- (12) Regarding the conclusion of "long-range interactions mediated by both OSK and GET were significantly enhanced from early to final reprogramming transition, but GET arrays were located within more interactive environment reminiscent of multi-loop aggregates.": to be more rigorous, some background control regions of nucleosome arrays (non OSK or non GET bindings) should be added to avoid the observation is just part of the global changes showing no specificity of the OSK or GET bindings.
- (13) It is not clear to this reviewer that Figure 4F H1 shows significant higher signal level at Tfp2c binding sites. Any possibility to perform statistical test? Or any background region can be used as a control to show the signal is enriched?
- (14) Some analysis cutoffs were defined without explanation. For example, "For ATAC-seq sorted peaks, peaks were split based on RPGC to open (>20 RPGC) or close chromatin (<20 RPGC)". It seems the set was all ATAC-seq peaks, should them be all open sites as ATAC-seq peaks? Also, for the following "representing the value whereby no ATAC enrichment is observed within the central 301 bp peak over the flanking 350 bp either side". How were the 301 and 350 bps determined?
- (15) Another cutoff was also not clear to this reviewer: "we identified singly-bound and co-bound nucleosomes based on the presence of ChIP-seq summits within +/- 80bp from the dyad". How was the 80 bp determined?
- (17) It was not clear to this reviewer that "Fragment size enrichment heatmaps were drawn using plot2DO with ChIP-seq peak summits or TSS as a reference and the aligned raw bam files as the sample." Why did the authors need the ChIP-seq peak summits (and which TF's summits?) or TSS as a reference to check the MNase-seq reads fragment size?
- (18) Is Figure 2H presenting an intergenic region? Otherwise, gene information should be provided.

Minor:

- (1) Extended Figure 8E, Eomes was labeled as Eommes.
- (2) Figure 3B and similar analysis color bar indicates the signal are log2 enrichment, comparing to what kind of background?

Version 1:

Reviewer comments:

Referee #1

(Remarks to the Author)

The authors have addressed most of my concerns and as a result, the manuscript has been significantly improved. I am supportive of publication.

(Remarks on code availability)

Referee #2

(Remarks to the Author)

Overall, I find this revised manuscript a great improvement, both regarding the validation of the findings (particularly figure 3) and how the results are presented. The identification and convincing validation of signpost elements and the local redistribution of TF binding during differentiation/reprogramming are exciting discoveries that merit publication in Nature.

Yet, an obvious limitation of the study remains that it provides little direct evidence why such a mechanism is used during the multi-day process of differentiation/reprogramming. The authors focus on a hypothesis whereby signpost can help TF scanning/sampling, which could indeed be the case. Based on the presented evidence, I'm still not convinced that other models/hypotheses should not be considered as well. First, the scanning model hypothesis comes with the limitations that the observed TF binding reorganization during differentiation is in the order of days, whereas scanning and TF binding itself occurs at a time scale of seconds/minutes. Moreover, I don't see why the author's model requires a separation of enhancers and signposts, considering that these elements do not drastically differ in the fraction of the genome that would be available to TFs. Instead, I don't see why my previous comment about signposts as regulators of enhancer activity may not be equally plausible. Here, I imagine that signposts can act as rheostat or attenuator to either allow a gradual opening or an improved synchronization of opening (i.e. reduction of stochasticity/noise) of the nearby enhancers. Rather than focusing only on the current hypothesis/model, the authors should discuss/propose alternative explanations as well.

Minor comments (in order of their appearance in the text):

1. From line 118 and Fig. S3: the description in the figure legend how peaks/arrays were included in the analysis leaves space for ambiguity. The authors should be more explicit in their description of “summits of all identified OSKM peaks”. Does this mean that peaks for all factors should be present, or rather some factors, or is the presence of a peak for a single factor sufficient?
Also: figure panels S3g,h are mentioned prior to panels S3e,f in the text. The authors should reorganize their labeling to allow for the correct sequence of panels in the text.
2. From line 151 and Fig. S7: whereas the difference between solo-nucs and combo-nucs is interesting, I consider the dynamics between early and late differentiation equally relevant for the study. Fig. S7b,c should be added to the main figures.
3. From line 168: the authors mention differences between solo-nucs and combo-nucs. The differences for Gata3 in early replicating cells appear quite minimal though (Fig. 1f). How were these differences qualified?
4. Line 188: I’m not sure how these data strongly suggest independence. Should the claim be limited to a lack of preference?
5. From line 215: the y-axis in Fig. S8e varies considerably between factors. Isn’t this difference about motif spreading, rather than density?
6. Line 241: Fig. 8b should read Fig. 9b?
7. From line 254: the authors should verify the labels in Fig. S9e. According to the provided GSM entries, several data sets are from MEFs, instead of ESCs (H3K4me3, H3K4me1, Rad21; compare Fig. S13e).
8. Line 306 and further. I’m not convinced that the term “loop extrusions” should be used. I agree that the elements are enriched within loops (as compared to the presence at loop anchors). Yet, “extrusions” suggest an active process for their formation (as for instance is the case for the formation for long-range chromatin loops by SMC-proteins), but evidence for such an activity is not provided in this study. I urge the authors to use the word “loops” instead.
9. From line 314 and Fig. 4b-d: the identification of the reduced interactions within the OSK arrays and the contacts between the flanking borders are intriguing. I’m not convinced though that the current analysis allows the conclusion that the opposing Klf1 motifs are aligned towards one direction. At a distance of a few kbs, the chromatin chain will be quite flexible, thus allowing “flipping” as well. To confirm this claim, the authors will need to confirm within the Micro-C data that convergent or divergent orientations of the Klf4 motifs are enriched over tandem orientations on either side of the ligation junctions. Additionally: (i) Could the authors add more tick marks to panel 4b, both on the x- and y-axis? The size of the depletion and the length of the “jets” that emanate from the corners are difficult to estimate. (ii) Panel 4c is not the most intuitive. Why do the authors not show their result as in Fig. 5i (and move this panel to the supplement)? This will make it more clear that this is an example from a single locus and not a pile-up (and thus should not be directly compared to Fig. 4a). Moreover, could the authors synchronize the use of colors between Fig. 4a and 4c?

(Remarks on code availability)

Referee #3

(Remarks to the Author)

The authors have done a great job in extensively addressing most of reviewers' comments. My suggestions/concerns have been thoughtfully addressed. I have one remaining question.

Figure 4c shows the same genomic region as Figure 2g. There are ChIP-seq/ATAC-seq/MNase-seq signals within the borders embraced by the two dash lines, suggesting it was not a low-mappability region. But it appears that signals are missing within the borders for Micro-C data. Can the authors explain why the signals are missing? Or is there a better example?

(Remarks on code availability)

Version 2:

Reviewer comments:

Referee #2

(Remarks to the Author)

In this newly revised manuscript, the authors have sufficiently addressed all my minor comments.

Reading the replies to my questions about data interpretation, I’m left with the impression that the issue is more with ambiguity in the writing rather than a difference in opinion.

Specifically:

1. Rebuttal, page 1: Until their latest comment, I was under the impression that the authors envisioned a guided-search model that incorporates a gradual opening of the chromatin surrounding the signposts and enhancers (as shown in Fig. 2h), thereby expanding the search space over the space of several days. Instead, from what I gather, the authors envision that the kinetics of the guided search do not so much change, but rather the functional output of the combinatorial binding. This potential source of confusion, linked to different time scales represented by the experiments, merits a sentence in the

discussion.

2. Rebuttal, page 2: From the data presentation, I had not realized that signposts (~187 Mb) cover a much larger fraction of the genome as compared to OSK co-occupied enhancers (~ 5 Mb). I think these are important numbers, so why not add an explicit mention in the text?

Besides this, I'm fully supportive of the publication of this study.

(Remarks on code availability)

Referee #3

(Remarks to the Author)

The authors have fully addressed my concerns and, I believe, also addressed other reviewers' questions. I have no more concerns.

Keji Zhao

(Remarks on code availability)

Response to referees' comments:

We thank all three reviewers for their positive feedback and insightful comments. Please find below our point-by-point response (Bold text) to all the points raised (highlighted in yellow).

Referee #1 (Remarks to the Author):

O'Dwyer et al. explore the role of the 3D chromatin context in TF-induced reprogramming using various TF-induced cell fate transitions, nucleosome spacing and TF binding, chromatin accessibility and 3D chromatin folding. They find that the binding of individual TFs to chromatin is well explained by the presence or absence of the respective TF motif on mononucleosomes. By contrast, the co-binding of cell fate-instructive TFs such as OKS or GET instead follows more complex rules that involve directional binding of TFs within and nearby nucleosomal arrays. These binding patterns dynamically change during reprogramming in a seemingly non-random manner. Moreover, the co-binding of TFs within and nearby nucleosomal arrays are different for different TF cocktails and correlate with 3D chromatin structure and H1 binding. Based on the comparison of TF co-binding patterns during early and late stages of reprogramming to either iPSCs or iTSCs, the authors propose that chromatin fibers serve as "signposts" for TFs to engage with closed chromatin before being able to access nearby cis regulatory elements (CREs).

This is a comprehensive study that offers intriguing new principles by which TF combinations may engage with chromatin to active CREs during (induced) cell fate transitions. As such, the manuscript could be of interest to a general readership. However, I have some concerns about the experimental setup and the conclusions drawn from the analyses. Moreover, experiments to validate some of the authors' model are missing. The impact of the study would be elevated if the authors could show that suppression of histone H1 expression during early OKS reprogramming facilitates acquisition of an ESC-like CRE state as one would predict. Similarly, it would help the authors' argument if they could show that an iPSC reprogramming system that is more efficient and homogeneous (via use of small molecules or perturbation of other factors) reduces the binding of TFs to signpost elements nearby ESC-like CREs.

We thank the reviewer for reading our manuscript and appreciating the impact of our findings in the field of cell fate transition. We have addressed all the major issues raised by the reviewer, and thus, we believe that our paper is suitable for publication in Nature.

Specific comments:

1. All reprogramming experiments appear to be based on the introduction of individual dox-inducible vectors into MEFs. However, it is unclear what fraction of cells actually express all TFs. This variable can obviously change the authors' interpretations in a major way. It would have been preferable to use polycistronic vectors for these experiments to avoid heterogeneity of TF expression. The authors need to either repeat some of the key experiments with a polycistronic vector or, at a minimum, demonstrate that the majority of cells infected with multiple vectors express all TFs.

We thank the reviewer for pointing out the importance of infection efficiency. To carry out ChIP-seq of all TFs during reprogramming, large amount of chromatin was required (40ug of chromatin per ChIP reaction out of 3-4 replicates). Hence, large number of cells were needed to be infected for each reprogramming condition. To achieve the cell numbers for all reprogramming systems, we

have used ultracentrifugation to generate high virus titer preps ($\sim 7 \times 10^8$ infectious units per ml (IFU)). Such high viral titer cannot be achieved using polycistronic vectors, due to the large polycistronic vector size which significantly reduces the lentivirus packaging efficiency¹. Overall, our single factor lentivirus constructs are just below the 10kb packaging limit of lentivirus system, and thus having polycistronic vectors encoding 4 to 5 TFs will increase this size to more than 15 Kb, reducing the viral titer by three order of magnitude ($\sim 10^5$ IFU)¹. In the revised manuscript, we have included the quantification of our infection efficiency for each TF (Extended data Fig. 1b,c). Using TF-specific antibodies, immunostaining analysis shows that about 95% of cells can be induced with dox to ectopically express each TF. This suggests that around 80% of cells can induce the expression of all four TFs in OSKM, GETM and BG₄S₉M combination, and all five TFs in GETMR combination. Considering ChIP-seq is an enrichment-based method, the TF peaks identified in this study were generated largely from the cells that express all four/five factors.

OSKM Polycistronic vectors have been successfully used to generate secondary reprogramming models, in which, the resulting iPSC lines containing the integrated polycistronic cassette are used to generate MEFs either from mouse chimeras or engineered mouse lines. The integrated polycistronic cassette is then induced to reprogram MEFs back to iPSCs. However, this system is not appropriate for our other TF combinations (GETM, GETMR and BG₄S₉M) as not all of them generate iPSCs. Furthermore, it would be time consuming to generate enough MEFs from chimeras or engineer specific mouse lines for each TF system.

Nonetheless, we have analysed data from two well-defined OSKM secondary systems with good quality ChIP-seq data, which is now included in (Fig. 2h and 2i). Additionally, we have generated our own data from another OSKM secondary system, which is now included in (Fig. 4h):

- 1- We have analysed OSKM ChIP-seq data (GSE102518) generated by Zviran et al², who used a highly efficient (deterministic) secondary reprogramming system in which mouse embryonic fibroblast (MEF) carrying a single copy of doxycycline (DOX)-inducible polycistronic OSKM transgene. In this system, MEFs are genetically identical and depleted of MBD3 (Mbd3^{f/f}), containing m.Col1a-TetO-STEMCCA-OSKM and R26-M2RtTA. We have confirmed that in this secondary reprogramming system, OSK also binds to the nucleosome arrays we defined using our primary reprogramming system (Fig. 2h).
- 2- We have also analysed Oct4 and Sox2 ChIP-seq data (GSE101905) generated by Knaupp et al³, using another secondary reprogramming system of MEFs from the rtTA3-OSKM reprogrammable mouse model. Again, these MEFs are all genetically identical and hetero/hemizygous for the Oct4-GFP reporter, OSKM cassette and rtTA3. Interestingly, in this study the authors sorted cells for Thy1 or SSEA1 in early reprogramming, where SSEA1 positive cells are more likely to complete reprogramming compared to Thy1+ve cells that resist reprogramming. We revealed an intriguing difference in the way Oct4 and Sox2 spread across the nucleosome arrays in SSEA1+ve vs Thy1+ve cells (Fig. 2i). So, comparing ChIP-seq data from Thy1+ve and SSEA1+ve cells has not only addressed the heterogeneity of reprogramming but also reveal a striking difference in OSKM combinatorial binding that leads to inducing pluripotency.
- 3- We have also generated our own data using another reprogrammable MEF system constructed by our colleague Prof. Keisuke Kaji. These TNG-MKOS-MEFs contain a Nanog-GFP-IRES-Puro knock-in reporter, and Dox-inducible MKOS-IRES-mOrange polycistronic reprogramming cassette in the Sp3 locus⁴. Using ATAC-seq as a proxy of OSKM binding, we have confirmed that OSKM bind and open the same nucleosome array we mapped in our original primary reprogramming system (Fig. 4h).

Overall, data from these different reprogramming models are consistent with our conclusion, validating our lentivirus primary reprogramming system. More importantly, we clarify that OSKM

binding is highly consistent across the different reprogramming models if binding to nucleosome arrays rather than the local ChIP-seq peaks was considered. Therefore, we present a unified model that can settle a long-standing debate about the inconsistent OSKM binding in different reprogramming models.

2. A key conclusion of this manuscript is that the binding of single TFs to mononucleosomes (in MEFs that presumably express multiple TFs such as OSKM or GETM; see point 1) is explained by the presence or absence of the respective TF motif. Is it possible that this pattern was observed because of the heterogeneity of MEFs transduced with multiple TF vectors, i.e. are the authors looking at cells that only received one or two TF? To control for this possibility, the authors should perform some of the assays with MEFs expressing a single TF (in addition to the meta-analyses of published data).

We have now addressed the heterogeneity of MEFs in response to point 1 above. However, to support that we are measuring the combinatorial binding of multiple factors rather than an average of individual TF binding from a pool of cells expressing different factors or correlating the ChIP-seq peaks of separate TF binding events, we present the following evidence:

- 1- We have shown in (Extended data Fig. 8b) that Oct4 binding where Oct4 is expressed as a single factor in MEFs, does not overlap with the OSKM co-binding to nucleosome arrays, confirming that OSKM co-binding occurs exclusively in MEFs overexpressing all the OSKM factors. Oct4 lone-binding however, is very similar between cells that express Oct4 as a single factor or in combination with SKM, confirming that binding to nucleosome arrays is only targeted in the presence of all OSKM factors not factors binding individually in a mixed cell population.
- 2- We have added EMSA supershift experiments to confirm that Oct4 and Sox2 from OSKM-48h can physically bind together to oligonucleotides with Oct-Sox combined motif (Extended data Fig. 8c). Furthermore, we show that this Oct4-Sox2-DNA complex is not bound in MEFs overexpressing Oct4 alone (Extended data Fig. 8d). Similarly, we have added supershift-EMSA to validate the co-binding of c-Myc and Tfp2c in GETM-48h (extended data Fig. 13e).
- 3- As explained in point (1) above, we have analysed data from Zviran et al ², who used a polycistronic STEMCCA-OSKM expression cassette in a secondary reprogramming system (Fig. 2h). We clearly show that our OSK-nucleosome arrays are also enriched for OSK from this study, accumulating at the border of the array following the motif orientation as observed using our single factor primary system.
- 4- We have also analysed Oct4 and Sox2 ChIP-seq data from Knaupp et al ³, another secondary reprogramming system with polycistronic OSKM cassette (Fig. 2i). In his study, the authors have measured Oct4 and Sox2 binding from two cell populations in early reprogramming sorted by two cell surface markers SSEA1 and Thy1 to indicate their reprogramming potential. In cells that are destined to convert to iPSCs (SSEA1 +ve), we show that Oct4 and Sox2 are bound together to our nucleosome arrays and enriched at the near borders according to their motif orientation. Interestingly, in cells refractory to reprogramming (Thy1 +ve), Oct4 and Sox2 enrichment spread beyond the nucleosome array and reach all the way from the near to the far borders. This indicates that these cells may resist reprogramming by not directing TF binding towards the pluripotency enhancers.

Importantly, OSKM, GETM and GETMR are all functional combinations and together can convert cellular identity not when these factors are expressed individually. Interestingly, the non-functional control BG₄S₉M combination show very limited co-binding (Extended data Fig. 4b), suggesting that TF combinatorial binding is crucial for reprogramming.

3. The authors need to characterize the TF systems they are using in terms of efficiency of reprogramming to iPSCs (OSKM) or iTSCs (GETM), and the fraction of iPSCs vs iTSCs that are being generated with GETMR. Also, the authors need to do some basic characterization of the generated iPSCs and iTSCs using marker analysis after transgene silencing and differentiation potential.

We have now included a full characterization of OSKM/GETMR-iPSCs, and GETMR/GETM-iTSCs during reprogramming by RNA-seq (Fig. 1b and Extended Data Fig. 2a). We have shown that iPSCs and iTSCs are almost identical in gene expression of ESCs and TSCs, respectively. Additionally, we have characterized our GETM-iTSCs, OSKM-iPSCs and GETMR-iPSCs by immunofluorescence of key trophoblast stem cells and pluripotency markers, respectively (Extended Data Fig. 2b-d). To confirm full reprogramming, we have used qPCR to show that the GETM, OSKM, and GETMR transgenes have been silenced in three independent clonal iTSC/iPSC lines for the corresponding combination (Extended Data Fig. 2e-g).

Regarding the fraction of iPSCs vs iTSCs that are being generated with GETMR: we have previously established a four-fluorescent knock-in reporter system (BYKE-MEFs) to accurately track reprogramming to iTSCs and iPSCs⁵. The BYKE-MEFs contain: (1) Nanog-2A-EGFP, a cytoplasmic reporter that specifically marks iPSCs; (2) Elf5-2A-EYFP-NLS, a nuclear reporter that is specific for iTSCs; and (3) Utf1-2A-tdTomato and (4) Esrrb-2A-TagBFP cytoplasmic reporters that mark both cell types. Using FACS, we show that BYKE-MEFs infected with GETMR cultured in ESC medium produced mainly iPSCs and some differentiated trophoblasts (See below Figure 1D from Benchetrit et al⁵). GETMR-infected BYKE-MEFs grown in TSC medium formed mostly iTSCs, with few iPSC colonies (See below Figure 1E from Benchetrit et al⁵).

Redacted

Regarding the differentiation potential, our iTSC and iPSC reprogramming models have been functionally characterized extensively in our previous studies⁵⁻⁷. The full characterization of our new unpublished TMR reprogramming to iPSCs is now included in (Extended Data Figure 15e-h). This includes reprogramming efficiency, expression of pluripotency markers by immunofluorescence and qPCR, as well chimera contribution assays of two independent iPSC lines.

4. What is the chromatin status and physiological role, if known, of TF motifs that serve as signposts? Are these elements normally bound and active in other cell types?

We have now analysed a large number of ChIP-seq data including H3K9me1, H3K9me2, H3K9me3, Suv39h1, Suv39h2, HP1, H3K4me3, H3K4me2, PolIII, P300, CTCF, and RAD21 in different cell types. Although, we are not able to include all the analysis because of space limitation, we have included analysis in ESCs and MEFs as they are the most relevant to this study. No published data is currently available in mTSCs:

We have shown in (Extended Data Fig. 9d,e) that OSK arrays are mainly repressed by heterochromatin (H3K9me3/Suv39h1-2/HP1) and depleted from open chromatin marks in ESCs, consistent with H1 enrichment that we reported in the paper. Interestingly, HP1 and Suv39H2 are predominantly enriched at the OSK nucleosome array borders in ESCs, with a particular phased

enrichment of H3K9me1, H3K9me3, and Suv39H1 within the arrays. We have also observed a depletion of the structural proteins CTCF and Rad21 within OSK arrays. Altogether, this analysis indicated that these nucleosome arrays are tightly regulated in ESCs.

GET arrays were also depleted from open chromatin histone marks and PolII in MEFs as expected (Extended Data Fig. 13e). Surprisingly, however, GET arrays were depleted from any of CTCF and Rad21 (cohesin subunit) in MEFs, despite being located at the boundaries of TADs with a stripe Micro-C pattern. This pattern is usually associated with DNA being extruded through a loop across large distances by Cohesin and CTCF in open chromatin^{8,9}. Therefore, our data suggests that GET may translocate through closed chromatin fibres not by loop excursions but simply spread across the tightly connected chromatin loop junctions.

We believe that defining the physiological role of these elements in other chromatin associated process or other cell states is beyond the scope of this study, but our study will encourage future investigations to address these questions as we mention in our discussion.

5. The impact of this study would be elevated if the authors could connect the observation of TF binding to signpost elements to some functional readout. For example, they could show that depletion of H1 (in the OKSM system) or the use of a more efficient iPSC reprogramming system reduces the binding of TFs to signposts. Alternatively, they could assess TF co-binding patterns in reprogramming-proficient (SSEA1+) vs reprogramming-deficient (SSEA1-) intermediates when using a homogeneous OKSM system.

We have analysed Oct4 and Sox2 ChIP-seq data from Knaupp et al³, using secondary OSKM reprogramming system, in which SSEA1+ve and Thy1-ve cell populations have been sorted (Figure 2i), please see our reply to points #1 and #2. We confirm that TF co-binding in different secondary systems occur within the same OSK arrays we mapped in our primary MEF reprogramming system.

As suggested by the reviewer, we have both overexpressed and knocked down H1 in reprogrammable MEFs (secondary reprogramming system)⁴, to demonstrate that H1 protein levels negatively affect reprogramming (Fig. 4i). Using ATAC-seq as proxy for OSKM binding we also demonstrate that depletion and overexpression of H1 results in less chromatin accessibility during early reprogramming, consistent with less OSKM co-binding (Fig. 4h). Moreover, we have performed Micro-C experiments to study the effects of H1 levels on chromatin 3D organization. Interestingly, we reveal a remarkable twisting and untwisting of the chromatin fibre when H1 is overexpressed and knocked down, respectively (Fig. 4e,f). So, we provide more functional evidence that H1 levels affect chromatin organization and OSKM binding during reprogramming.

6. The GETMR data are interesting but complicated by the fact that two different cell fates, iPSCs and iTSCs, are being generated from the same starting population. It would be more informative to assess whether GTMR or TMR, which like OKS generates iPSCs, follow a similar or different mechanism of TF co-binding (i.e. more like OKSM or more like GETM?).

We agree with the reviewer that when studying GETMR reprogramming, we are averaging two separate reprogramming trajectories. This conclusion is supported by our RNA-seq data (Fig. 1b) and the fact that in ESCs media GETMR can give rise to iPSCs and some differentiated trophoblasts and in TSCs media GETMR can give rise to iTSCs and some iPSC colonies⁵. However, during early reprogramming GETMR whether cultured under ESC conditions (iPSC output) or TSC conditions (iTSC output) they both start from the same early transcriptional state (Fig. 1b). Furthermore, ATAC-seq of GETMR-72h under both ESC and TSC conditions induce the same chromatin changes in all TF bound sites (see figure below). Our main question was how GETMR (one TF combination) can lead to both iPSCs and iTSCs starting from the same chromatin state and the surprising answer

was that TF competitive binding bifurcate the iPSC and iTSC reprogramming trajectories. This is opposite to what we initially expected that cooperative TF binding expand the reprogramming capacity of GETMR.

We agree with the reviewer that studying TMR combination instead of GETMR will be more appropriate to reveal whether this combination follow a distinct or similar mechanism to OSKM during iPSC reprogramming, but we believe that this question is not the focus of this study.

7. TF-induced reprogramming involves ectopic and often super-physiological expression of TFs in heterologous cell types, raising the question of whether the authors' observations extend to physiological cell fate transitions or whether they are an "artifact" of forced TF expression systems. For example, is it possible that forced expression of OSKM in established iPSCs or ESCs would also keep signpost elements engaged by TFs? The authors should at least discuss these possibilities in the manuscript.

Please see our answer to (point #4). Overall, we have now included in (Extended Data Fig. 9d) the OSKM signpost elements are silenced by H3K9methylation and HP1 in ESCs. There is also a specific pattern of how these silencing marks are distributed within the arrays, indicating that these elements are under tight control. So, we don't envisage that the binding of TFs to these elements after the completion of reprogramming in iPSCs as we have previously shown that H3K9me3 blocks OSKM engagement with large parts of the genome despite their overexpression in early reprogramming¹⁰. In fact, we anticipate that the silencing of these signpost elements may help lock OSK binding to pluripotency enhancers in fully reprogrammed iPSCs.

8. A previous study by Plath et al found that in mouse, OSKM initially engage with open chromatin rather than heterochromatin (PMID: 28111071). How do the authors reconcile their data with this previous observation (using a more homogeneous polycistronic OSKM transgene)?

To reconcile these seemingly inconsistent observations about whether OSKM initially target open or closed chromatin, we have previously published a paper together with the Plath laboratory comparing our two reprogramming systems¹¹. As we used the human reprogramming system in our initial model¹⁰, we mainly concentrated on the difference between the human and mouse reprogramming models not the OSKM delivery system used in our studies. Therefore, to specifically address the reviewer's question about the difference between our current mouse primary reprogramming system and the Chronis et al mouse secondary system, we have reanalysed Oct4 ChIP-seq data from Chronis et al using our pipeline¹². To assess chromatin accessibility, we have also reanalysed ATAC-seq data from MEFs, OSKM-MEFs (NO-dox), and OSKM-48h also generated by Chronis et al. As you can see below, about half of the sites (52%) bound by Oct4 were within closed chromatin, which does not represent the majority within open chromatin as reported by Chronis et al¹². These sites only open-up after inducing OSKM for 48h (OSKM-48h). Interestingly, even without inducing OSKM by dox in the secondary reprogramming MEFs (NO-DOX), there is a slight opening of Oct4 sites, which is consistent with the known leaky expression of the secondary OSKM system. Therefore, the more accessible Oct4 sites (48%) using

Chronis et al data compared to our 30% of Oct4 sites may be attributed to the leaky expression of OSKM transgenes in their secondary OSKM MEFs. Indeed, the reprogrammable mice where these secondary MEFs are generated from (R26-M2rtTA and Col1A-OSKM) have been originally reported to establish undifferentiated teratomas due to the leaky OSKM transgene expression (Supplementary Figure 5a,b from Stadtfeld et al)¹³. Furthermore, two other independent studies using different secondary reprogramming systems (see our answer to point #1 above) have both reported similar observations to ours that OSK mainly target closed chromatin during early reprogramming^{2,3}. Together, these studies in human and mouse support that OSK target mainly closed chromatin during early reprogramming acting as pioneer factors.

We believe that the main source of this discrepancy that led Chronis et al to conclude that OSKM target open chromatin in early reprogramming is the way the authors have characterized open and closed chromatin. While we use the pre-existing chromatin state in MEFs (prior to OSKM induction) as a reference to characterize whether the sites are open or closed (please see our answer to point #14 of reviewer #3 below for more details), Chronis et al used the 48h time point (OSKM-48h) to assess chromatin accessibility of OSKM sites (please see below Figure 2C from Chronis et al¹²). It is not clear to us why the authors used OSKM-48h ATAC-seq instead of MEF or OSKM-MEF (NO-dox), which they have also generated but never used in their study (see our analysis above). We argue that chromatin accessibility in OSKM-48h is the effect of OSKM binding not the other way around.

Overall, we believe that comparing individual ChIP-seq peaks between different data sets will always result in inconsistencies due to the fact that TF binding occurs at a much larger genomic scale. Our study proposes a new way to look at TF combinatorial binding to larger nucleosome arrays, unifying the differences reported by using the various reprogramming models.

Redacted

9. Why did the authors show ATAC-Seq data of the 72h time point of reprogramming even though most assays were performed at the 48h time point when cells were shown to be more homogeneous?

We have carried out ATAC-seq at 72 hr time point for two reasons. First, we aimed to study the effects of TF binding on chromatin accessibility. Since we mapped TF sites at 48h, we assessed chromatin changes to these sites 24h later and therefore 72hr after TF induction. Second, at 72h the cells react homogeneously to TF induction despite being grown under different culture media for 24hr (Extended Data Fig. 1a). This is particularly important for GETMR combination where the infected cells are grown in either ESC or TSC media. Overall, we conclude that ESC or TSC culture conditions have very little impact on chromatin accessibility and gene expression at this early stage of reprogramming, confirming that the observed changes are driven directly by the binding of TFs which we measured in MEF media. We have now explained our rationale for using the 72h time point in the text (L120-121).

10. The authors' use of the terms sign-posts, search-guards, search-targets, etc. are a bit vague and ambiguous, especially since they are used to describe two seemingly different mechanisms (OSKM vs GETM reprogramming).

Although OSKM and GETM use different mechanisms to search for their enhancers, they both use what we call signpost elements to guide them to their sites. We have simplified the text in the discussion where we replaced the terms "search-guides" and "search-targets" by simply "guides" and "targets", respectively. We have also included a simpler diagram in figure 6e to explain the guided search mechanism.

11. Experiments using BSG expression are not particularly helpful but rather distracting and the authors should consider removing the data to streamline the manuscript.

We have made the control factors more relevant in the new revision. Firstly, we made it clear by RNA-seq data that these factors don't change MEF identity (Fig. 1b and Extended Data Fig. 2a). Secondly, we clarified that these factors exhibit very limited combinatorial binding within closed chromatin (Extended Data Fig. 3f, 4b, and 14a), thereby leading to the important conclusion that TF combinatorial binding is linked to function. Thirdly, as we explained in the text (L84-85), these factors belong to the same structural families as OSKM but still show different motif readouts on nucleosomes (Extended Data Fig. 6d, and 7f,g). The finding that Sox9 not acting as a pioneer factor

was also an important point that was revealed by this control combination (Extended Data Fig. 3d). In conclusion, this control TF combination is crucial to support our general finding that the DNA-binding domain structure is not sufficient to predict TF combinatorial binding or whether a TF can act as a pioneer factor.

Referee #2 (Remarks to the Author):

Complex cell state transitions like reprogramming require the combined action of multiple TFs. How these factors engage which each other to associate with chromatin remains largely unexplored.

In this study, the authors have addressed this question using four different TF cocktails that each result in a different reprogramming outcome. By combining data on TF binding (ChIP-seq) with nucleosome organization (MNase-seq and Micro-C) at different stages of reprogramming, the authors report a number of new findings that provide important new insights into how TFs bind shared regions in the genome:

1. Co-binding mostly occurs over larger arrays of nucleosomes
2. Different TF cocktails show a different syntax of co-binding
3. The authors identify “signpost” elements, where TF cocktails bind at early stages of reprogramming. Signposts neighbor regulatory elements that are bound by the same TF cocktails at later stages, indicating that they help making these regulatory elements available.

This study presents an impressive and exciting set of data, and particularly the identification of signpost elements (and associated dynamics) provides an important new dimension to the dynamics of TF (co-)binding. As concluded by the authors, it’s likely that these new findings expand to other complex cell state transitions (development, carcinogenesis) and possibly other cellular functions as well.

We thank the reviewer for reading our manuscript and appreciating the impact of our findings in the field of cell fate transition.

Unfortunately, the manuscript suffers from a number of issues that make it a sub-optimal choice for Nature (see below). Instead, it will be better suited, as a single submission or split up in multiple manuscripts, for a more specialized audience.

We believe that we addressed all the major issues raised by the reviewer (see below), and we therefore regard our revised manuscript to be a more coherent story supported by rigorous experimentation and suited for publishing in Nature.

Issues:

1. Biology: whereas the authors suggest that the identified syntax for TF binding (as inferred through their binding motifs) must be essential to determine cooperative binding, a second explanation may be that this syntax is necessary for the correct functional and regulatory output of these TFs: relative TF motif orientation and spacing may be essential for correct protein-protein interactions (among members of the TF cocktails or when engaging with other factors) or creation of a correct higher-order nucleosome structure.

Without functional validations (e.g. by changing the orientation or relative position of TF motifs, followed by analysis of TF co-binding and transcriptional regulation), both explanations should be considered.

We have now added an elegant experiment to functionally validate that motif syntax directly affect OSKM binding to signpost elements during reprogramming (Fig. 3). Using a piggybac system

we have constructed a dual reporter vector where the signpost element of the Nanog gene has been flipped to change the direction of OSK motifs (Fig. 3a and 3b). The first eGFP reporter is driven by the intact Nanog promoter-signpost-enhancer element (~5kb). While the second tdTomato reporter is under the Nanog promoter and enhancer separated by a flipped signpost element (Fig. 3b). We inserted an insulator between the two reporter cassettes to eliminate transcriptional interference, and flanked two insulators at both ends to minimize integration position effects from the neighbouring chromosomal environment. The resulting piggyBac construct was successfully integrated into ESCs, generating ESCs expressing both eGFP and tdTomato at similar efficiency (Fig. 3d). The dual eGFP/tdTomato+ve ESCs were then injected into host blastocyst (eGFP/tdTomato -ve), which efficiently contributed to chimeric mouse embryos at E13.5 (Fig. 3c). Importantly, both eGFP and tdTomato reporters were equally silenced in all tissues apart from the gonad where they remained active at E13.5, reflecting the precise expression of Nanog at this embryonic stage (Fig. 3e). We have therefore ectopically expressed OSKM in MEFs isolated from chimeric E13.5 embryos where the eGFP/tdTomato reporters were silenced. In support of our conclusions, during reprogramming of these MEFs with OSKM, eGFP +ve cells started to gradually increase from day 9 (Fig. 3f-h). Whereas tdTomato was not activated until after the completion of reprogramming (Fig. 3f-h and Extended Data Fig. 9f-h). We believe that this data clearly demonstrates that motif orientation within signpost elements plays a crucial role to effectively reactivate pluripotency enhancers such as Nanog during reprogramming.

2. Accessibility: the manuscript tries to convey an immense amount of information, resulting both in very dense and complex text and figures. For the complete interpretation of the data, the supplemental data was often essential as well. The addition of more information to the main figures will therefore be required. Moreover, considerable prior knowledge on cellular reprogramming and nucleosome structure is a must. This is further complicated by the very high density of abbreviations in the text.

Combined, I therefore estimate that the current manuscript will be difficult to access for the wider audience of Nature. Instead, the results may profit from splitting in two separate manuscripts (temporal dynamics of TF binding and differences between TF cocktails).

We have changed the manuscript significantly, simplifying the language and making it more accessible to a wider audience. Our two senior collaborators Prof Arthur Skoultchi and Prof Steven Pollard have both read the manuscript in detail and helped improve the accessibility of the manuscript. Although, our data may seem dense and our experiments may seem very technical, we made it clearer that we are addressing a simple and fundamental question in biology, which is how TFs find their enhancers to change cellular identity. This is reflected in the abstract and throughout the text. We have also resorted to simple diagrams in every figure to explain our conclusions. Overall, we strongly believe that our study provides an answer to a long-standing question in the field, which will interest the wider audience of Nature.

3. Interpretation: the authors go to great lengths to interpret the data presented in their figures. While reviewing, in a number of cases it was either not obvious to me how the authors came to their conclusions or the interpretation appeared to contradict other findings.

- Line 53-55: I don't see how individual TF pioneer function and cooperative TF action are the most obvious aspects to raise questions about co-binding.

We have rephrased this statement and removed the question about individual TF pioneer function and cooperative TF action (line 65-67). To make our rationale clearer, we started by focusing on the question of how TFs transition from their early sites to their cell-type-specific sites during

reprogramming. We raised the question of individual binding at the end of the introduction (line 75-76).

- Line 88-90: Why does this result suggest off-target binding? The function of these TFs could be different at different stages of reprogramming.

We fully agree with the reviewer that the term "OFF-target" does not reflect whether these sites are functional or not during reprogramming. The term "OFF-target" was coined by Wapinski et al to differentiate between OSK pioneer activity and the Ascl1 pioneer activity which seems to be "ON-target" during neuronal reprogramming¹⁴. But our question is how OSK transition from this early binding to pluripotency enhancers. We have made this point clearer by explicitly saying that OSK do not initially bind to pluripotency enhancers (line 68-69).

- Line 120-122: This is not a fully accurate description of the aim. The analysis does not show that motifs are differentially recognized, but rather that the relative position and orientation of motifs among each other is different.

We have now rephrased this statement to make the aim clearer:

- Line 151-155: "To define the motif grammar that may dictate whether TFs bind alone or together to nucleosomes,....".

- Line 126: And so is Klf4, as is mentioned in line 138.

We have now addressed this by saying:

Line 157: "More interestingly however, motif distribution of each OSK factor...."

- Line 134-138: See my first comment: the composite motif may not be essential to promoter co-binding but rather for correct functional/regulatory function.

We have completely removed the statement about the Oct/Sox composite motif as it is not important in this study. But please refer to our answer to the first comment above about the functionality of OSK binding to Nanog expression during reprogramming. We have now experimentally validated that flipping the orientation of OSK motifs outside the Nanog enhancer (signpost element), impedes the guiding of OSK binding to the enhancer during reprogramming. It is important to note that our conclusions are not describing the function of motifs within the enhancers which has been the subject of many previous studies.

- From line 152: I am confused about this section: in line 134 the co-occurrence of Oct4 and Sox2 motifs was discussed, but this section reports that Oct4 and Sox2 do not co-occur.

We apologise for the confusion; we have significantly changed the wording of this section to make it clearer. We started by adding the following statement:

- Line 179-181: "As the average enrichment of different motifs on combo-nucs does not necessarily represent their co-occurrence on the same nucleosomes, we assessed the interdependence of motif occurrence after fixing one motif arrangement criteria."

The main aim of this analysis was to examine whether the average enrichment of multiple motifs within a group of nucleosomes equate to their co-occurrence on the same nucleosomes. It is important to note that these nucleosomes are co-bound by multiple TFs as defined by the presence of their ChIP-seq peaks. Thus, by fixing the incidence of one TF motif and verifying the other, we have been able to examine motif co-occurrence on the same nucleosomes. In conclusion, the nucleosomes co-bound by three TFs can contain a motif of single factor but not necessarily all three together, however on average all three motifs are present.

- Line 182-183: I do not see how a single TF could bind to an array of nucleosomes.

We agree with the reviewer and that's what we report in the paper. For example, when expressed individually, Oct4 binding peaks are much smaller and can be contained within mono-nucleosomes. We added the following statement to make this clearer.

- Line 201-202: "When bound individually, O,S,K engaged much smaller sites containing one nucleosome on average."

However, this does not exclude other TFs (self-oligomerizing) that may spread across large nucleosome arrays individually. But none of the factors we studied in this paper behave this way.

- Line 197-199: See my first comment: motif directionality may be essential for correct functional/regulatory function.

See our answer to the first comment above. We have added the following conclusion to make this point clearer.

- Line 232-235: "In conclusion, specific motif grammar at the scale of chromatin fibre may direct OSK to accumulate near one border. The critical reprogramming activities may therefore lie within the context of broader nucleosome arrays and associated motif orientations, rather than a specific local motif grammar at enhancers."

- Line 222-224: The fact that different TFs recognize different motifs is well known. This conclusion is a poor description of the previous paragraph.

We have now restructured this section and discussed the OSK and GET nucleosome arrays separately. The revised structure of the manuscript provides a more cohesive narrative, directing the reader through commonality and differences between the various TF combinations we studied in this manuscript.

- Line 344-345: It's difficult to imagine how the absence of its binding motif will instruct Esrrb to reduce Tfap2c binding. An alternative interpretation may be that an unknown downstream Esrrb target may be involved?

We have indeed discovered a much simpler explanation than our original conclusion. So, we added the following statement.

- Line 433-442: " Interestingly, Esrrb nucleosomes where Tfap2c is retained were only enriched for Tfap2c motifs, and nucleosomes bound by Esrrb but don't overlap with Tfap2c sites were only enriched for Esrrb motifs, suggesting that their co-binding may occur at a nucleosome array level

(Fig. 6c). We have therefore identified nucleosome arrays that contained Tfp2c retained or lost sites in GETMR-48h. Remarkably, Micro-C pileup analysis revealed that Tfp2c binding was retained in arrays with more inter-nucleosome contacts that can mediate the combinatorial binding of Esrrb-Tfp2c (compare yellow arrowheads in Fig. 6d and diagrams above). In conclusion, the addition of Esrrb restricts GETM combinatorial binding, by retaining Tfp2c in nucleosome fibres with discrete topology."

In conclusion, Esrrb and Tfp2c are associated with their respective motifs that are present in different nucleosomes, but brought together by the nucleosome array structure, not motif co-occurrence. We have added a small diagram to Fig. 6d to clarify this conclusion. We agree with the reviewer that other cellular factors may also contribute to bringing Tfp2c and Esrrb together but this needs to be investigated in future studies.

- Line 353: I'm not sure if this conclusion can be considered as competition (as mentioned in the abstract and figure title).

Our conclusion was based on the fact that Tfp2c enrichment was depleted when adding Esrrb to the GETM combination (Extended Data Fig. 15a,b). Importantly, the sites where Tfp2c remained bound were also co-occupied with Esrrb, Gata3 and Eomes (Extended Data Fig. 15b). This combined with our previous observation that that changing the stoichiometry between Eomes and Esrrb in GEMTR can change the outcome of iPSC and iTSC reprogramming, led us to hypothesize that Eomes and Esrrb compete to interact with Tfp2c in GETMR. Our co-IP data (Extended Data Fig. 15d) and reprogramming with TMR to iPSCs (Extended Data Fig. 15e-h) support that GETMR act in competition.

Referee #3 (Remarks to the Author):

The manuscript by O'Dwyer et al., entitled "Nucleosome fibre topology guides transcription factor co-binding to gene regulatory elements" investigated the mechanisms that mediate TF co-binding during cell fate transition. The authors compared four sets of transcription factor combinations during cellular reprogramming: OSKM (Oct4, Sox2, Klf4, and C-Myc), GETM (Gata3, Eomes, Tfp2c, and c-Myc), GETMR (Gata3, Eomes, Tfp2c, c-Myc, and Esrrb), and BS9G4M (Brn2, Sox9, Gata4, and c-Myc, control set) by integrating ChIP-seq, MNase-seq, ATAC-seq, and Micro-C data. The major conclusions include: (1) motif recognition on mono-nucleosomes does not explain TF co-binding; (2) TF co-binding is critical to cell reprogramming by comparing between early and final reprogramming cells; (3) TFs co-bindings occur in arrays of nucleosomes displaying distinct motifs in terms of distribution and orientation. These regions of nucleosome fibers serve as "signpost elements"; (4) they proposed that TF co-binding to signpost elements in early reprogramming can directly lead to tissue-specific genome occupancy in accordance with a guided search model; (5) two distinct guided-search modes were identified. one for OSK that follows motif orientation within H1-depleted chromatin environment and the other for GET that follows motif density in an H1-enriched environment.

We thank the reviewer for their time and careful assessment of our manuscript.

Major comments, concerns, and questions:

TF co-binding has been recognized as a major mechanism that is critical to cellular fate transition

and function. The authors obtained multiple genomics data to address this important question and performed sophisticated and comprehensive analysis of the data. I have several major concerns as described below:

(1) Some of the reported findings from the manuscript have been reported in previous studies. For example, it was reported that OSK co-occupancy drives reprogramming of somatic cells (Chronis et al. 2017). The specific motif distribution and orientation for TF co-binding was reported by Avsec, et al. (2021). They found that TF co-binding was guided by soft syntax rules, which followed clear intermotif distance-dependent relationships consistent with nucleosome-mediated cooperativity. The TF cooperativity associated with specific motif pairs was often directional and consistent with motifs mediating the role of pioneer TF with different strengths. Ibarra, et al. (2020) reported the DNA shape as driver for TF cooperativity.

Although Chronis et al has reported about OSKM cooperative function during iPSC reprogramming, but as alluded to by reviewer #1 in comment #8 (see above), the combinatorial binding reported in that study mainly occurs in open chromatin within cis regulatory elements of somatic genes to mainly silence the somatic program by displacing somatic TFs. Our findings on the other hand represent previously unknown mechanism of how OSKM and other TF combinatorial binding to nucleosome arrays within closed chromatin can lead to cell-type specific binding to enhancers (not somatic enhancers). More importantly we reveal for the first time that motif grammar used by different TF combinations at a nucleosome array scale with specific 3D organization.

Many studies including Avsec et al. mentioned by the reviewer ¹⁵, have been able build algorithms that can predict motif syntax used by TFs to recognize enhancers of already established cell types. Thus, these studies don't provide any insights on how these TFs find and activate these enhancers prior to the establishment of a specific cell type. Our experimental study however has measured the binding of different TF combinations during cell fate transition. Hence, we show how TFs find their cell-type enhancers starting from a different cell type. We reveal what other studies have not been able to show that motif organization spreading beyond the enhancers guide TFs to enhancers. Importantly, we show that that motif organization become only apparent when nucleosome positioning and 3D organization are considered. So, rather than concentrating on the 2D DNA sequence within enhancers, we propose that new algorithms should consider the 3D organisation of nucleosome DNA at a much larger scale. Interestingly, Avsec et al report that Nanog binding has a strong ~10.5-bp periodic pattern, which is typical footprint for a nucleosome. However, this study does not consider nucleosome positioning as a key element that may help to display these motifs to Nanog. We have now added this discussion to our revised manuscript to steer future research directions (line 484-486).

DNA shape has been reported to influence TF cooperative binding as reported by Ibarra et al.¹⁶. But again, this study characterized cooperative binding at single site resolution i.e. less than 20bp. Our study however, as we mentioned above, demonstrate that the cooperative and competitive binding of TFs occurs at a much larger nucleosome array scale (few kb in size). Nucleosomes are well known to change DNA shape. Thus, we have added to the discussion that it would be interesting to study DNA shape at nucleosome fibre resolution and reveal how that influence TF binding. We also propose that future studies should consider the DNA shape induced by nucleosomes rather than using the common B-DNA conformation as a reference.

Altogether, our study reveals previously unknown mechanisms of TF co-binding to chromatin at a higher-order chromatin scale and within a cell fate transition context.

(2) This study may be considered as a significant extension of the author's previous work (Soufi et al. 2015 PMID: 25892221) with more TFs combination sets and the Micro-C data. The authors found that OSK preferentially target silent sites enriched for nucleosomes *in vivo*.

Our previous study by Soufi et al ¹⁷ have indeed studied the interaction of OSKM with nucleosomes to explain their pioneer activity. However, in that study we went to great length to study the individual binding of each of the OSKM factors to nucleosomes. The main aim was to demonstrate that OSK but not c-Myc act as pioneer factors and can target closed chromatin by binding to nucleosomes on their own, i.e. without any assistance from other TFs. At that time, it was still not recognized that TFs such as OSK can bind to nucleosomes and nucleosomes were largely regarded as barriers to TF binding. The dominating theory at the time was nucleosome mediated cooperativity in which a group of TFs (like OSKM) must compete with histones for DNA (evicting nucleosomes) to access chromatin ¹⁸.

The current study however, rather than an extension of our previous study, we have completely shifted our focus from individual factor binding to nucleosomes to multiple TF combinatorial binding so we can address a different question. The results were also surprising in that TF combinatorial binding does not occur at a mono-nucleosome level as was expected, but at multiple nucleosomes with distinct topology and motif distribution. Importantly, we studied not just OSKM, but various other TF combinations in different reprogramming contexts. So, we present an unprecedented view of how TFs recognize chromatin when working as a group.

(3) Regarding the proposed model of higher-order nucleosome topology mediating the TF co-binding, it is now just correlative. Much more evidence is required to substantiate the model.

We have now experimentally validated that OSK motif orientation within the signpost element of the pluripotency gene *Nanog* play an important role in activating *Nanog* enhancer during reprogramming (please see Fig. 3).

More specifically, we have demonstrated that by changing H1 protein amounts we can change chromatin fibre topology, which allowed to the study the effect on OSKM and GETM binding and reprogramming.

- We have been able to successfully overexpress and down regulate H1, which has been a very challenging task due to the high redundancy of the many H1 variants and lack of variant specific antibodies. By collaborating with The Skoultchi lab, a world leader in H1, we have been able to accurately measure the amounts of global H1 levels by targeting H1.4 variant using HPLC and LC-MS (Extended data Fig. 11 a-c).
- Using Micro-C, we present a previously unreported insight of how H1 levels change the twisting and untwisting of the chromatin fibre (Fig. 4e,f). We have also shown that changing H1 levels shortens the spacing between nucleosomes within OSK arrays.
- We have carried out ATAC-seq to show that both H1 overexpression and knockdown block OSKM from targeting and opening chromatin within the nucleosome arrays (Fig. 4h).
- This is also translated into an impaired reprogramming activity (Fig. 4i).
- We have also shown that H1 overexpression and knockdown significantly affect GET nucleosome array topology and reprogramming (Fig. 5g,k).

Thus, we believe that the new data support our original findings and substantiate our model.

(4) Reprogramming using multiple TFs is a dramatic non-physiological process. Most physiological

cellular differentiation involves relatively mild or stepwise processes. How does the guided search model apply to those processes?

Although TF mediated reprogramming is regarded as non-physiological, it is a stepwise process and was only discovered by drawing parallels with embryonic development¹⁹. Similarly, we believe that what we learn from cellular reprogramming can be applied back to cell fate decisions during differentiation. Furthermore, the non-physiological overexpression of TFs is known to occur in cancer. For example, the overexpression of Sox2 in glioblastoma is known to drive the neuron stem cell reprogramming²⁰. So, our guided search model may also apply to study cell fate decisions during disease progression.

Nevertheless, our study addresses a much more fundamental question that has been raised since the 1970s when Riggs et al reported that the lac repressor binding to λ DNA seemed to find its target (operator) site in vitro at a rate 1000-fold faster than estimated by a diffusion-controlled process (random)²¹. For this reason, the classical facilitated diffusion model has been proposed, explaining that TFs binding to low-affinity sites (non-specific) facilitate the search for the target sites by a combination of 1D sliding and 3D hoping mechanism²². Since then, the facilitated diffusion model has been validated by in vivo imaging in bacteria²³. However, the binding of TFs to the eukaryotic genome is much more complex and take place at multi-level chromatin organization structure, which cannot be explained by the classical facilitated diffusion model²⁴. Other models such as random sampling have been proposed, but again this cannot explain the dynamics by which TFs find their sites in mammalian cells. For example, single-molecule fluorescence tracking has shown that Sox2 finds its target sites in fewer than 100 binding attempts²⁵, suggesting that it only samples a miniscule fraction of the genome (< 1% of 2.5 Gbp)²⁶. By considering the many layers of complexity present in the eukaryotic genome including chromatin accessibility, nucleosome positioning and chromatin 3D organization in the context of cell fate transition, we discovered that TFs in combinations recognize large nucleosome assemblies. Therefore, by reducing the dimensionality to be explored in the eukaryotic genome, TFs can effectively search and find their sites.

In conclusion, our guided-search model provides an answer to what could not be explained by previous models. We make this general point clearer in our revised discussion and (Fig. 6e).

(5) Except for Figure 2H/Figure 3C and F, which presented explicit examples, most of this manuscript presented high-level abstract aggregation analyses. To facilitate the readers to understand the results, I suggest the authors to include specific examples for a majority of the aggregation analyses, which can also directly show the quality of sequencing data.

We have now included the Nanog locus in Fig. 3a, which we validated experimentally, and added a specific example for GET co-binding in Fig. 5i. We have also added examples of larger genomic regions in (Extended Data Fig. 10f and 13d). The combination of these specific examples and the higher-level analysis can illustrate to the reader our findings.

(6) The description of analysis methods was comprehensive. However, necessary codes and test data should be deposited at GitHub for readers to check reproducibility, especially for Figure 1D, Figure 2C, and other similar figures.

We have now deposited all our codes to GitHub (will be released upon publication). For reviewers access please use the following access tokens:

- Repository: https://git.ecdf.ed.ac.uk/soufi_lab/motif_mononucleosome. (Access token: ZVe-i1ntWN2ykaMuaqCk)
- Repository: https://git.ecdf.ed.ac.uk/soufi_lab/motif_nucleosome_arrays. (Access token: g8sz_cA9AYmWFzukMM2B)

(7) As the control set, what are the overlaps for BS9G4M with ESC or TSC (Extended Figure 1B)?

Brn2, Sox9 and Gata4 are not expressed in ESCs and TSCs, and therefore the overlap suggested by the reviewer is not possible. But we have now included the overlap between Brn2 in BS₉G₄M-48h with neuron progenitor cells (NPCs), where it is endogenously expressed (Extended Data Fig. 1h, see below). Interestingly, Brn2 does not target neuronal sites when combined with BS₉G₄M unlike in the BAM combination when it is recruited by the ON-target pioneer factor Ascl1.

(8) "To assess whether motif readout plays a role in TF co-binding to nucleosome arrays, we mapped motif enrichment across nucleosome arrays within close chromatin." What is the logic behind the selection to only study the close chromatin?

There are three main reasons for specifically studying closed chromatin:

1. Conceptually, the pluripotency and trophoblast stem cell genes are silenced and their enhancers are within closed chromatin in fibroblasts. Therefore, TFs must be access closed chromatin during reprogramming to activate gene expression and convert cellular identity.
2. Most sites (~70% of sites) targeted by pioneer TFs in early reprogramming are within closed chromatin (extended data Fig. 3c,d,e, and i).
3. Pioneer TFs target intact nucleosomes within closed chromatin, while open sites contain fragile nucleosomes or depleted from nucleosomes altogether (Extended Data Fig. 5c-e).

We have added the following statement to make our logic clearer: (line 136-138) " Considering that pioneer TFs engage closed chromatin by recognizing their cognate sites on nucleosomes, we hypothesized that the arrangement of multiple motifs on a single or mono-nucleosome would be sufficient to drive combinatorial TF binding."

(9) "Isolating OSK nucleosomes arrays containing 4 or more Sox2 motifs/kb on either strand revealed that these arrays contain..." Why Sox motif but not Oct4 motif?

The reason we used Sox2 motifs not Oct4 because it is more frequent, so we can include most of arrays in our analysis. We have now added the following statement to make this point clearer: (L220-224) " We focused on Sox2 motifs as it was the most prevalent within and outside the OSK arrays. Almost all (~90%) of OSK nucleosomes arrays contained 4 or more Sox2 motif/kb arranged

in a unique direction, allowing us to split the arrays into two distinct groups, with limited overlap, based on the strandness of Sox2 motifs (Extended Data Fig. 8f,g)".

(10) "In fully reprogrammed cells, nucleosome arrays co-bound by OSK (CRES) contain almost no Oct4 and Sox2 motifs, no orientation-specific motif distribution, and smaller in size". Were these OSK co-bound peaks weak ChIP-seq peaks and therefore could not represent true co-binding of OSK? Same question to the fully reprogrammed GET nucleosome arrays.

In fully reprogrammed iPSCs, OSK were bound together at pluripotency-specific enhancers. Therefore, OSK peaks in iPSCs represent the highest affinity peaks such as Nanog enhancer, which we have now included in (Fig. 3a). The same conclusion for GET binding in fully reprogrammed iTSCs. Because we mapped motif enrichment across larger nucleosome arrays (the whole enhancer) rather than single peaks, motif enrichment seemed not as substantial as previously shown. But even when mapping motif enrichment across single OSK sites on mono-nucleosomes, it is clear that Oct4 and Sox2 motifs were not as enriched as Klf4 motifs (Extended data Fig. 6a,b). This illustrates our main conclusion, that TFs don't necessarily transition from low affinity sites with low motif enrichment to high-affinity sites with more motif enrichment, which is the main premise of the classical facilitated diffusion model. We argue that OSK use motifs distributed on nucleosome arrays as guides not as their final targets, otherwise they will stay associated with the initial perfect motifs they encounter in early reprogramming. In conclusion, our guided search model provided a new insight into how TFs may use motif recognition to search for enhancers.

(11) "Interestingly, transitioning from early to fully reprogrammed cells was concurrent with a lateral shift of OSK co-binding across the upstream near-border of the nucleosome arrays following OSK motif direction (Fig. 2G,H)". What's the fraction of this shifting profile for co-binding events? I feel only if the majority of the nucleosome arrays show this shifting pattern, this observation is interesting. If indeed the majority of them show the shifting pattern, I accept the conclusion of "the early co-binding of both OSK and GET is not random and can directly lead to lineage-specific binding to CREs in fully reprogrammed cells.....guiding TF co-binding to CREs targets". However, globally GET do not show the same shifting pattern. How do the authors explain the discrepancy for a very basic biological process?

Out of the 9,176 nucleosome arrays we defined to be co-bound by OSK within open chromatin in fully reprogrammed iPSCs, and therefore most likely to be enhancers, 5,592 were located within 500 bp across the border from OSK nucleosome arrays in early reprogramming. This means that ~61% of OSK co-bound enhancers were guided during early reprogramming by binding to adjacent nucleosome arrays. However, this number is arbitrary as it may increase by increasing the 500bp threshold. Moreover, pluripotency enhancers contain more factors than OSK and thus defining their borders by OSK combinatorial binding may not be accurate. Therefore, we have included an unbiased analysis in (extended data Fig. 9c) to examine the significance of our findings. We have measured the distance of OSK nucleosome arrays in early reprogramming to the nearest enhancer as defined by ATAC-seq (open in iPSCs) compared to random genomic regions. This measurement has revealed that OSK nucleosome arrays are in close proximity to enhancers (~500bp average), while the random sequences are equally distributed across ~8kb distance. We ran the two-sided Wilcoxon rank-sum test, which resulted in: $w = 10,696,640,768$ and $P = 2.286 \times 10^{-6}$. This confirms that the lateral proximity of OSK nucleosome arrays to pluripotency enhancers is not random.

In GETM reprogramming system, we found that GET combinatorial binding is directly linked to enhancers in iTSCs through long distance interactions (loops), which are mediated by H1-enriched chromatin structure. We have now used cLoop2²⁷ and called all loops that link GET nucleosome

arrays in early and open enhancers in iTSCs, compared to random sequences. As shown in Fig. 5j, GET arrays are significantly more connected to iTSC enhancers than random sequences, supporting our conclusion that GET initial binding can guide their cell-type specific association with enhancers. ($n = 315,352$, $P < 5.752 \times 10^{-6}$, two-sided Wilcoxon rank-sum test).

Although, OSK co-binding during early reprogramming can spread relatively across short distances to enhancers next door in iPSCs and GET translocate across large distances to enhancers in iTSC. These measurements are in 2D, i.e. without considering the physical distance in 3D in the context of nucleosome array structure. So, what may appear to be far away in 2D, can be spatially nearby in 3D. We have illustrated our model in (Fig. 6e) to show that both OSK and GET may spread across similar distances in chromatin but follow different guiding mechanism. This difference should be interpreted as OSK and GET recognizing distinct chromatin folds and should not be considered as an inconsistency of a basic biological process. To illustrate this point, one can argue that TFs containing different DNA-binding domains bind completely different DNA sequences, but the sequence-specific TF-DNA interaction remains a basic biological process. Similarly, different groups of TFs bind together to different nucleosome array structures but TF combinatorial binding to nucleosome arrays is a basic biological process.

(12) Regarding the conclusion of “long-range interactions mediated by both OSK and GET were significantly enhanced from early to final reprogramming transition, but GET arrays were located within more interactive environment reminiscent of multi-loop aggregates. ”: to be more rigorous, some background control regions of nucleosome arrays (non OSK or non GET bindings) should be added to avoid the observation is just part of the global changes showing no specificity of the OSK or GET bindings.

All our pileup analysis that measures the overall Micro-C contacts (short and long range distances) across OSK and GET arrays are normalized against random sequences generated by shifting each target sequence within 0.1 - 1 Mbp distance. An average of 10 control regions per averaged window are used to generate the heatmaps. Furthermore, all Micro-C matrices are normalized against sequencing depth and coverage. We have made this clearer in our Materials and Methods section. Furthermore, we have now added circos plots (extended data figure 10f) to show long range contacts within an exemplar region of 1Mbp in chr12, including OSKM binding, ATAC-seq and MNase-seq. This will help to show how much of the global changes involve OSK binding. To show the same point for GET binding, we have now included Micro-C Contact heatmaps of an exemplar region of 3Mbp region on chr12 (Fig. 5i and Extended Data 13d).

(13) It is not clear to this reviewer that Figure 4F H1 shows significant higher signal level at Tfap2c binding sites. Any possibility to perform statistical test? Or any background region can be used as a control to show the signal is enriched?

We have now removed this analysis from the revised manuscript as it is not essential to our conclusions about GETMR competitive binding. We have concentrated on the main results that nucleosome arrays with retained Tfap2 sites contain more inter-nucleosome contacts than those where Tfap2c binding is lost after adding Esrrb (Fig. 6d). We propose that this nucleosome connectivity promote Tfap2c-Esrrb cooperativity in the arrays. Although, we still believe that this high inter-nucleosome connectivity is driven by H1 binding, it would be challenging, as the reviewer rightly pointed out, to accurately measure H1 levels at this resolution using the current CHIP-seq data. We believe that further biochemical studies are required to reconstitute nucleosome arrays with different amounts of H1 and see the effects on Tfap2c-Esrrb co-binding. Unfortunately, these are very technically challenging and time-consuming experiments and thus beyond the scope of this study.

(14) Some analysis cutoffs were defined without explanation. For example, “For ATAC-seq sorted peaks, peaks were split based on RPGC to open (>20 RPGC) or close chromatin (<20 RPGC)”. It seems the set was all ATAC-seq peaks, should they be all open sites as ATAC-seq peaks? Also, for the following “representing the value whereby no ATAC enrichment is observed within the central 301 bp peak over the flanking 350 bp either side”. How were the 301 and 350 bps determined?

We have now included more details in the Material and Method section to explain our rationale. However, due to limited space we cannot show all the bioinformatic analysis behind our decisions, which we list below.

1- The main question was to define whether TFs bind open or closed chromatin as measured by ATAC-seq. Using the Model-based Analysis of ChIP-seq (MACS2), we built empirical peak models for each TF by mapping the highly enriched paired tag density (see panel (a) in figure below). The distance between the forward and reverse tags (bandwidth) corresponds to the experimental size of DNA fragments bound by TFs in our ChIP-seq experiment, which is about 270bp for the Oct4 example shown in the figure shown below. Additionally, after calling TF ChIP-seq peaks with MACS2, the average peak size was defined as 280bp (panel (b) in the figure below). Therefore, we decided to fix the peak size to 300bp as it represents the average peak and fragment size for all TFs, concentrating on the most enriched central part of the peak (summit) where TFs are most likely associated with DNA. It is important to note that some peaks called by MACS2 contain multiple summits, so we considered all summits individually. To be symmetrical, 150bp was added to each side of the peak summit resulting in 301bp.

2- As there is no input DNA for ATAC-seq, we compared the enrichment of ATAC-seq within the peak to the immediate flanking region (local chromatin). We considered local chromatin accessibility within a total region of 1 Kb (standard practice in the field). After removing the 300bp enriched peak, the remaining 700bp was considered as flanking regions and was therefore divided by two resulting in 350bp. Then, ATAC-seq enrichment was measured as a ratio of central peak over flanking region.

3- To assess whether the TF site is within open or closed chromatin, we had to identify a threshold to eliminate any ATAC-enriched sites within closed chromatin (regions containing sparse ATAC-seq reads but appear as enriched). By plotting ATAC-seq enrichment of TF sites as function of number of reads (sequence coverage normalised in RPGC or reads per genome coverage), we identified the baseline of 20 RPGC (vertical line in panel c of the figure below). Therefore, any site below this threshold (< 20 RPGC) was defined to be closed or inaccessible. This is the most stringent baseline to make sure all closed sites used in this study are indeed within inaccessible chromatin regions.

(15) Another cutoff was also not clear to this reviewer: “we identified singly-bound and co-bound

nucleosomes based on the presence of ChIP-seq summits within +/- 80bp from the dyad". How was the 80 bp determined?

At the atomic resolution, the DNA wrapped around the histone octamer to form the core nucleosome particle was determined to be 146 bp²⁸. However, MNase-seq usually map nucleosomes ranging from ~140 to 200bp in size due to fuzziness, nucleosome wrapping and unwrapping dynamics, and association with linker histone²⁹⁻³¹. According to our MNase-seq data shown in (Extended Data Fig. 5b), intact nucleosomes are 160bp on average. We have made this clearer by the statement below: (line 142) "Intact mono-nucleosomes were identified as ~160bp fragments on average in all cell types..."

Based on the average nucleosome size of 160bp, we considered any TF site within +/- 80bp from the dyad (the centre of nucleosome) to be directly associated with a nucleosome. We added the following to the text to make this point clearer:

Line 152-154: " we identified nucleosomes bound by TFs individually (solo-nucs) and in combination (combo-nucs) based on the presence of ChIP-seq summits within +/- 80bp from the dyad (160bp nucleosome size), ..."

(17) It was not clear to this reviewer that "Fragment size enrichment heatmaps were drawn using plot2DO with ChIP-seq peak summits or TSS as a reference and the aligned raw bam files as the sample." Why did the authors need the ChIP-seq peak summits (and which TF's summits?) or TSS as a reference to check the MNase-seq reads fragment size?

All Plot2D heatmaps were drawn using TF peak summits as a reference to identify whether TF sites are enriched for nucleosomes. We have made this clearer by adding a label in each panel of Extended Data Fig. 5 c-e. Plot2D use raw bam file to quantify nucleosome enrichment as well as fragment size. Using raw reads will identify the experimental nucleosomal DNA size rather than inferring the size from the dyad. This is explained in more details in the Materials and Methods section.

(18) Is Figure 2H presenting an intergenic region? Otherwise, gene information should be provided.

Genome browser screenshot in Fig. 2h represents an intergenic region. The closest TSS is for Gphn gene, which is ~300 kb away, or 230 kb away from a hypothetical gene that is 95% identical to human HDAC1 and HDAC2.

Minor:

(1) Extended Figure 8E, Eomes was labeled as Eommes.

corrected.

(2) Figure 3B and similar analysis color bar indicates the signal are log2 enrichment, comparing to what kind of background?

All pileup heatmaps are drawn using random sequences as controls and were normalized around the total Micro-C signal over the whole window displayed at the resolution specified underneath each heatmap. So, for Fig. 4b this would be +/- 20kb flanking the OSK arrays using a 100 bp

resolution to measure fine scale inter-nucleosome contacts. We have made this point clearer in the figure legends. The example shown in Fig. 4c, directly compares the intensity of PETs across such a scale of 20kb.

REFERENCES:

1. Kalidasan, V., Ng, W.H., Ishola, O.A., Ravichantar, N., Tan, J.J., and Das, K.T. (2021). A guide in lentiviral vector production for hard-to-transfect cells, using cardiac-derived c-kit expressing cells as a model system. *Scientific Reports* *11*, 19265. [10.1038/s41598-021-98657-7](https://doi.org/10.1038/s41598-021-98657-7).
2. Zviran, A., Mor, N., Rais, Y., Gingold, H., Peles, S., Chomsky, E., Viukov, S., Buenrostro, J.D., Scognamiglio, R., Weinberger, L., et al. (2019). Deterministic Somatic Cell Reprogramming Involves Continuous Transcriptional Changes Governed by Myc and Epigenetic-Driven Modules. *Cell Stem Cell* *24*, 328-341 e329. [10.1016/j.stem.2018.11.014](https://doi.org/10.1016/j.stem.2018.11.014).
3. Knaupp, A.S., Buckberry, S., Pflueger, J., Lim, S.M., Ford, E., Larcombe, M.R., Rossello, F.J., de Mendoza, A., Alaei, S., Firas, J., et al. (2017). Transient and Permanent Reconfiguration of Chromatin and Transcription Factor Occupancy Drive Reprogramming. *Cell Stem Cell* *21*, 834-845 e836. [10.1016/j.stem.2017.11.007](https://doi.org/10.1016/j.stem.2017.11.007).
4. Ruetz, T., Pfisterer, U., Di Stefano, B., Ashmore, J., Beniazza, M., Tian, T.V., Kaemena, D.F., Tosti, L., Tan, W., Manning, J.R., et al. (2017). Constitutively Active SMAD2/3 Are Broad-Scope Potentiators of Transcription-Factor-Mediated Cellular Reprogramming. *Cell Stem Cell* *21*, 791-805 e799. [10.1016/j.stem.2017.10.013](https://doi.org/10.1016/j.stem.2017.10.013).
5. Benchetrit, H., Jaber, M., Zayat, V., Sebban, S., Pushett, A., Makedonski, K., Zakheim, Z., Radwan, A., Maoz, N., Lasry, R., et al. (2019). Direct Induction of the Three Pre-implantation Blastocyst Cell Types from Fibroblasts. *Cell Stem Cell* *24*, 983-994 e987. [10.1016/j.stem.2019.03.018](https://doi.org/10.1016/j.stem.2019.03.018).
6. Benchetrit, H., Herman, S., van Wietmarschen, N., Wu, T., Makedonski, K., Maoz, N., Yom Tov, N., Stave, D., Lasry, R., Zayat, V., et al. (2015). Extensive Nuclear Reprogramming Underlies Lineage Conversion into Functional Trophoblast Stem-like Cells. *Cell Stem Cell* *17*, 543-556. [10.1016/j.stem.2015.08.006](https://doi.org/10.1016/j.stem.2015.08.006).
7. Jaber, M., Radwan, A., Loyfer, N., Abdeen, M., Sebban, S., Khatib, A., Yassen, H., Kolb, T., Zapatka, M., Makedonski, K., et al. (2022). Comparative parallel multi-omics analysis during the induction of pluripotent and trophectoderm states. *Nature Communications* *13*, 3475. [10.1038/s41467-022-31131-8](https://doi.org/10.1038/s41467-022-31131-8).
8. Davidson, I.F., Bauer, B., Goetz, D., Tang, W., Wutz, G., and Peters, J.M. (2019). DNA loop extrusion by human cohesin. *Science* *366*, 1338-1345. [10.1126/science.aaz3418](https://doi.org/10.1126/science.aaz3418).
9. Grubert, F., Srivas, R., Spacek, D.V., Kasowski, M., Ruiz-Velasco, M., Sinnott-Armstrong, N., Greenside, P., Narasimha, A., Liu, Q., Geller, B., et al. (2020). Landscape of cohesin-mediated chromatin loops in the human genome. *Nature* *583*, 737-743. [10.1038/s41586-020-2151-x](https://doi.org/10.1038/s41586-020-2151-x).
10. Soufi, A., Donahue, G., and Zaret, K.S. (2012). Facilitators and impediments of the pluripotency reprogramming factors' initial engagement with the genome. *Cell* *151*, 994-1004. [10.1016/j.cell.2012.09.045](https://doi.org/10.1016/j.cell.2012.09.045).
11. Fu, K., Chronis, C., Soufi, A., Bonora, G., Edwards, M., Smale, S.T., Zaret, K.S., Plath, K., and Pellegrini, M. (2018). Comparison of reprogramming factor targets reveals both species-specific and conserved mechanisms in early iPSC reprogramming. *BMC Genomics* *19*, 956. [10.1186/s12864-018-5326-1](https://doi.org/10.1186/s12864-018-5326-1).
12. Chronis, C., Fiziev, P., Papp, B., Butz, S., Bonora, G., Sabri, S., Ernst, J., and Plath, K. (2017). Cooperative Binding of Transcription Factors Orchestrates Reprogramming. *Cell* *168*, 442-459 e420. [10.1016/j.cell.2016.12.016](https://doi.org/10.1016/j.cell.2016.12.016).
13. Stadtfeld, M., Maherali, N., Borkent, M., and Hochedlinger, K. (2010). A reprogrammable mouse strain from gene-targeted embryonic stem cells. *Nature Methods* *7*, 53-55. [10.1038/nmeth.1409](https://doi.org/10.1038/nmeth.1409).

14. Wapinski, O.L., Vierbuchen, T., Qu, K., Lee, Q.Y., Chanda, S., Fuentes, D.R., Giresi, P.G., Ng, Y.H., Marro, S., Neff, N.F., et al. (2013). Hierarchical mechanisms for direct reprogramming of fibroblasts to neurons. *Cell* *155*, 621-635. 10.1016/j.cell.2013.09.028.
15. Avsec, Ž., Weilert, M., Shrikumar, A., Krueger, S., Alexandari, A., Dalal, K., Fropp, R., McAnany, C., Gagneur, J., Kundaje, A., and Zeitlinger, J. (2021). Base-resolution models of transcription-factor binding reveal soft motif syntax. *Nature Genetics* *53*, 354-366. 10.1038/s41588-021-00782-6.
16. Ibarra, I.L., Hollmann, N.M., Klaus, B., Augsten, S., Velten, B., Hennig, J., and Zaugg, J.B. (2020). Mechanistic insights into transcription factor cooperativity and its impact on protein-phenotype interactions. *Nature Communications* *11*, 124. 10.1038/s41467-019-13888-7.
17. Soufi, A., Garcia, M.F., Jaroszewicz, A., Osman, N., Pellegrini, M., and Zaret, K.S. (2015). Pioneer transcription factors target partial DNA motifs on nucleosomes to initiate reprogramming. *Cell* *161*, 555-568. 10.1016/j.cell.2015.03.017.
18. Mirny, L.A. (2010). Nucleosome-mediated cooperativity between transcription factors. *Proceedings of the National Academy of Sciences of the United States of America* *107*, 22534-22539. 10.1073/pnas.0913805107.
19. Takahashi, K., and Yamanaka, S. (2006). Induction of pluripotent stem cells from mouse embryonic and adult fibroblast cultures by defined factors. *Cell* *126*, 663-676. 10.1016/j.cell.2006.07.024.
20. Bulstrode, H., Johnstone, E., Marques-Torrejón, M.A., Ferguson, K.M., Bressan, R.B., Blin, C., Grant, V., Gogolok, S., Gangoso, E., Gargica, S., et al. (2017). Elevated FOXG1 and SOX2 in glioblastoma enforces neural stem cell identity through transcriptional control of cell cycle and epigenetic regulators. *Genes Dev* *31*, 757-773. 10.1101/gad.293027.116.
21. Riggs, A.D., Bourgeois, S., and Cohn, M. (1970). The lac repressor-operator interaction: III. Kinetic studies. *Journal of Molecular Biology* *53*, 401-417. [https://doi.org/10.1016/0022-2836\(70\)90074-4](https://doi.org/10.1016/0022-2836(70)90074-4).
22. Berg, O.G., Winter, R.B., and von Hippel, P.H. (1981). Diffusion-driven mechanisms of protein translocation on nucleic acids. 1. Models and theory. *Biochemistry* *20*, 6929-6948. 10.1021/bi00527a028.
23. Elf, J., Li, G.W., and Xie, X.S. (2007). Probing transcription factor dynamics at the single-molecule level in a living cell. *Science* *316*, 1191-1194. 10.1126/science.1141967.
24. Badis, G., Berger, M.F., Philippakis, A.A., Talukder, S., Gehrke, A.R., Jaeger, S.A., Chan, E.T., Metzler, G., Vedenko, A., Chen, X., et al. (2009). Diversity and Complexity in DNA Recognition by Transcription Factors. *Science* *324*, 1720-1723. 10.1126/science.1162327.
25. Chen, J., Zhang, Z., Li, L., Chen, B.C., Revyakin, A., Hajj, B., Legant, W., Dahan, M., Lionnet, T., Betzig, E., et al. (2014). Single-molecule dynamics of enhanceosome assembly in embryonic stem cells. *Cell* *156*, 1274-1285. 10.1016/j.cell.2014.01.062.
26. Natarajan, A., Yardimci, G.G., Sheffield, N.C., Crawford, G.E., and Ohler, U. (2012). Predicting cell-type-specific gene expression from regions of open chromatin. *Genome Res* *22*, 1711-1722. 10.1101/gr.135129.111.
27. Cao, Y., Liu, S., Ren, G., Tang, Q., and Zhao, K. (2021). cLoops2: a full-stack comprehensive analytical tool for chromatin interactions. *Nucleic Acids Research* *50*, 57-71. 10.1093/nar/gkab1233.
28. Luger, K., Mader, A.W., Richmond, R.K., Sargent, D.F., and Richmond, T.J. (1997). Crystal structure of the nucleosome core particle at 2.8 Å resolution. *Nature* *389*, 251-260. 10.1038/38444.
29. Kornberg, R.D. (1974). Chromatin structure: a repeating unit of histones and DNA. *Science (New York, N.Y.)* *184*, 868-871.
30. Noll, M. (1974). Subunit structure of chromatin. *Nature* *251*, 249-251. 10.1038/251249a0.

31. Voong, L.N., Xi, L., Wang, J.-P., and Wang, X. (2017). Genome-wide Mapping of the Nucleosome Landscape by Micrococcal Nuclease and Chemical Mapping. *Trends in Genetics* 33, 495-507. <https://doi.org/10.1016/j.tig.2017.05.007>.

R1: The authors have addressed most of my concerns and as a result, the manuscript has been significantly improved. I am supportive of publication.

We thank the reviewer for their support and feedback that greatly helped to improve our manuscript.

R2: Overall, I find this revised manuscript a great improvement, both regarding the validation of the findings (particularly figure 3) and how the results are presented. The identification and convincing validation of signpost elements and the local redistribution of TF binding during differentiation/reprogramming are exciting discoveries that merit publication in Nature.

We really appreciate the positive feedback in support of the publication of our manuscript. We also like to thank the reviewer for their insightful comments that helped us to strengthen our findings.

Yet, an obvious limitation of the study remains that it provides little direct evidence why such a mechanism is used during the multi-day process of differentiation/reprogramming. The authors focus on a hypothesis whereby signpost can help TF scanning/sampling, which could indeed be the case. Based on the presented evidence, I'm still not convinced that other models/hypotheses should not be considered as well. First, the scanning model hypothesis comes with the limitations that the observed TF binding reorganization during differentiation is in the order of days, whereas scanning and TF binding itself occurs at a time scale of seconds/minutes.

We agree with the reviewer that the dynamics of TF binding to the genome (seconds/minutes) is much faster than the reprogramming process (days/weeks). But our ChIP-seq approach is capturing TF binding events during a 10 min period (fixing time by cross-linking) not during the whole reprogramming process. So, we are taking a 10 min snapshot during early reprogramming and another after the completion of reprogramming and thus have an average picture of all scanning events that occur during these two defined stages of reprogramming. We also agree with the reviewer that our study does not reveal the intricate details that drive the transition of TF-binding from early (signpost) to full reprogramming (enhancers). Future studies using higher resolution techniques such as single-molecule imaging at individual loci are required to address this question.

Moreover, I don't see why the author's model requires a separation of enhancers and signposts, considering that these elements do not drastically differ in the fraction of the genome that would be available to TFs.

Our data supports that there is a clear distinction between enhancers and signpost elements. For example, OSKM signpost elements are only occupied during early reprogramming and become enriched for heterochromatin factors in final reprogramming, suggesting that they are no longer accessible to OSKM in iPSCs. The opposite is true for enhancers, supporting that the two groups may serve distinct functions. Furthermore, OSK co-occupy open enhancers spanning total of 5,158,284bp (~0.2% of the genome). However, in early reprogramming OSK co-occupy signpost elements spanning a total of 96,927,091bp plus the rest of the unbound signpost element (from near- to far-border) spanning 186,688,000bp; together spanning 283,615,091bp (~10% of the genome). So guided by 10% of the genome, one can easily explain how OSK can effectively find 0.2% of the genome (enhancers). This is consistent with single molecule imaging showing that it takes less than 100 attempts for SOX2 and OCT4 to find their specific sites ¹.

Instead, I don't see why my previous comment about signposts as regulators of enhancer activity may not be equally plausible. Here, I imagine that signposts can act as rheostat or attenuator to either allow a gradual opening or an improved synchronization of opening (i.e. reduction of stochasticity/noise) of the nearby enhancers. Rather than focusing only on the current hypothesis/model, the authors should discuss/propose alternative explanations as well.

We agree with the reviewer that we cannot rule out that signpost elements can act as rheostat controlling the intensity of enhancer activity by controlling their chromatin opening. Nonetheless, we believe that the rheostat/attenuator model provided by the reviewer and our model are not mutually exclusive, i.e. the signpost elements can act as guides as well as rheostat or attenuators. Unfortunately, we don't have any experimental evidence that support the rheostat/attenuator model as of yet. Due to limited space, we are not able to discuss other possible models in more detail. But we have added the following statement: "It will be interesting to investigate whether the signpost elements can also act as rheostat or attenuator to fine-tune or synchronize enhancer activity".

Minor comments (in order of their appearance in the text):

1. From line 118 and Fig. S3: the description in the figure legend how peaks/arrays were include in the analysis leaves space for ambiguity. The authors should be more explicit in their description of "summits of all identified OSKM peaks". Does this mean that peaks for all factors should be present, or rather some factors, or is the presence of a peak for a single factor sufficient

We rephrased the description to the following: "Read density heatmaps of O,S,K,M ChIP-seq signal (blue) in OSKM-48h spanning ± 1 kb around the summits of O,S,K,M peaks pooled together". We have separated OSKM by

commas to indicate that the summits of the peaks from all four factors that were called individually were pooled together.

Also: figure panels S3g,h are mentioned prior to panels S3e,f in the text. The authors should reorganize their labeling to allow for the correct sequence of panels in the text.

This has now been fixed.

2. From line 151 and Fig. S7: whereas the difference between solo-nucs and combo-nucs is interesting, I consider the dynamics between early and late differentiation equally relevant for the study. Fig. S7b,c should be added to the main figures.

Unfortunately, we cannot include these figures in the main figure as it will not fit the size restriction that complies with NATURE format. We had to compress all the main figures considerably, making this task very challenging. Moreover, both early and final reprogramming don't show major changes in motif readout.

3. From line 168: the authors mention differences between solo-nucs and combo-nucs. The differences for Gata3 in early replicating cells appear quite minimal though (Fig. 1f). How were these differences qualified?

These differences have been qualified visually by inspecting the directionality and intensity of GATA3 motifs around the nucleosome dyads. It is apparent that GATA3 motifs on solo-nucs are distributed in more direction-specific orientation than combo-nucs.

4. Line 188: I'm not sure how these data strongly suggest independence. Should the claim be limited to a lack of preference?

We provided more explanation supporting our conclusion: "Importantly, the observed frequency of O,S,K motifs co-occurrence on the combo-nucs is almost identical to their expected independent probabilities (P), i.e. $P(OSK) = P(O)P(S)P(K)$ "

We have also removed the word "strongly" and rephrased to: "This suggests that OSK combo-binding and motif co-occurrence on mono-nucleosomes are independent events".

5. From line 215: the y-axis in Fig. S8e varies considerably between factors. Isn't this difference about motif spreading, rather than density?

We agree with the reviewer, and changed our conclusion to: "On average, OCT4 and SOX2 showed more motif spreading compared to KLF4 motifs".

6. Line 241: Fig. 8b should read Fig. 9b?

Fixed.

7. From line 254: the authors should verify the labels in Fig. S9e. According to the provided GSM entries, several data sets are from MEFs, instead of ESCs (H3K4me3, H3K4me1, Rad21; compare Fig. S13e).

We apologize for this error, we have corrected all the labels and also made sure that no mistakes were made in our original code.

8. Line 306 and further. I'm not convinced that the term "loop extrusions" should be used. I agree that the elements are enriched within loops (as compared to the presence at loop anchors). Yet, "extrusions" suggest an active process for their formation (as for instance is the case for the formation for long-range chromatin loops by SMC-proteins), but evidence for such an activity is not provided in this study. I urge the authors to use the word "loops" instead.

We agree with the reviewer, and to eliminate any confusion with active loop extrusion we have replaced "loop extrusions" with "loops" as suggested by the reviewer.

9. From line 314 and Fig. 4b-d: the identification of the reduced interactions within the OSK arrays and the contacts between the flanking borders are intriguing. I'm not convinced though that the current analysis allows the conclusion that the opposing Klf1 motifs are aligned towards one direction. At a distance of a few kbs, the chromatin chain will be quite flexible, thus allowing "flipping" as well. To confirm this claim, the authors will need to confirm within the Micro-C data that convergent or divergent orientations of the Klf4 motifs are enriched over tandem orientations on either side of the ligation junctions.

We agree with the reviewer that the distance of 1kb of chromatin will allow a single Klf4 motif to be flexible and orientated in different orientation. But according to our data, there is a cluster of Klf4 motifs not only one motif, together spanning ~ 1kb upstream the near border and another cluster of Klf4 motifs at the far border (Fig. 2d). Thus, if the near- and far-border clusters are linked together as shown by micro-C (Fig. 4b), then on average the two Klf4 motif clusters will point in the same direction on average. We appreciate that these conclusions are based on averages, but our experimental approaches (Micro-C and ChIP-seq) are average measurements. So, further bioinformatic analysis will not resolve this issue. Further biochemical and structural studies of individual loop junctions are required to reveal the role of tandem orientation of KLF4 motifs.

Additionally: (i) Could the authors add more tick marks to panel 4b, both on the x- and y-axis? The size of the depletion and the length of the “jets” that emanate from the corners are difficult to estimate.

Fixed

(ii) Panel 4c is not the most intuitive. Why do the authors not show their result as in Fig. 5i (and move this panel to the supplement)? This will make it more clear that this is an example from a single locus and not a pile-up (and thus should not be directly compared to Fig. 4a). Moreover, could the authors synchronize the use of colors between Fig. 4a and 4c?

We have changed Fig. 4c to be like Fig. 5i as requested by the reviewer. We have also called loops by FitHiChIP ($Q0.01$)², to only show significant loops rather than showing all PET raw data as arches which made the figure busy and confusing. We have synchronized the use of colors between Fig. 4a and 4c, and also kept the TF ChIP-seq colors consistent in all figures.

R3: The authors have done a great job in extensively addressing most of reviewers' comments. My suggestions/concerns have been thoughtfully addressed. I have one remaining question.

We thank the reviewer for taking the time to read our manuscript and all their instructive comments.

Figure 4c shows the same genomic region as Figure 2g. There are ChIP-seq/ATAC-seq/MNase-seq signals within the borders embraced by the two dash lines, suggesting it was not a low-mappability region. But it appears that signals are missing within the borders for Micro-C data. Can the authors explain why the signals are missing? Or is there a better example?

This is a very important point that we addressed in two ways. First, we changed the gene locus in Fig. 4c to *Nanog* (more relevant to pluripotency), which we validated experimentally in Fig. 3. We have also changed the locus to *Tfap2c* (more relevant to trophoblast stem cells) in the revised Extended data Fig. 8i. Second, rather than showing all PET raw data as arches, we have called loops by FitHiChIP², using coverage bias normalization and $QVALUE = 0.01$. Accordingly, we have re-quantified the loops that connect GET arrays to enhancers in Fig. 5j, using peak-to-peak loops called by FitHiChIP ($Q0.01$). Altogether, this will address any coverage bias in Micro-C that are not considered when showing raw data.

References:

- 1 Chen, J. *et al.* Single-molecule dynamics of enhanceosome assembly in embryonic stem cells. *Cell* **156**, 1274-1285 (2014). <https://doi.org/10.1016/j.cell.2014.01.062>
- 2 Bhattacharyya, S., Chandra, V., Vijayanand, P. & Ay, F. Identification of significant chromatin contacts from HiChIP data by FitHiChIP. *Nature Communications* **10**, 4221 (2019). <https://doi.org/10.1038/s41467-019-11950-y>